# Understanding the Statistical Accuracy-Communication Trade-off in Personalized Federated Learning with Minimax Guarantees

Xin Yu [* 1]   Zelin He [* 1]   Ying Sun [2]   Lingzhou Xue [1]   Runze Li [1]

## Abstract

Personalized federated learning (PFL) offers a flexible framework for aggregating information across distributed clients with heterogeneous data. This work considers a personalized federated learning setting that simultaneously learns global and local models. While purely local training has no communication cost, collaborative learning among the clients can leverage shared knowledge to improve statistical accuracy, presenting an accuracy-communication trade-off in personalized federated learning. However, the theoretical analysis of how personalization quantitatively influences sample and algorithmic efficiency and their inherent trade-off is largely unexplored. This paper makes a contribution towards filling this gap, by providing a quantitative characterization of the personalization degree on the tradeoff. The results further offers theoretical insights for choosing the personalization degree. As a side contribution, we establish the minimax optimality in terms of statistical accuracy for a widely studied PFL formulation. The theoretical result is validated on both synthetic and real-world datasets and its generalizability is verified in a non-convex setting.

## 1. Introduction

Federated Learning (FL) (McMahan et al., 2017) has emerged as a promising learning framework for aggregating information from distributed data, allowing clients to collaboratively train a shared global model in a communication-efficient manner. However, a critical challenge arising in the presence of various data across clients is data heterogeneity. In such cases, a single global model may fail to generalize well to all clients, motivating the need for model personalization. PFL enhances FL by learning personalized models tailored to individual clients, and has demonstrated strong empirical performance in various applications, such as driver monitoring (Yuan et al., 2023), and mobile computing (Zhang et al., 2024).

An important question in PFL is determining the degree of personalization, which controls the transition between fully collaborative training and pure local training. A higher degree of collaboration (less personalization) typically requires more frequent information exchange, potentially improving learning accuracy when client data distributions are similar. Conversely, increasing personalization reduces communication costs by prioritizing localized training, but may lead to higher generalization errors due to the limited size of client datasets (Paragliola, 2022). Understanding this trade-off is essential for optimizing model performance under communication constraints.

Most existing works focus purely on the algorithmic perspective of PFL (Lin et al., 2022; Li et al., 2024b; Wang et al., 2022; 2024b). However, the statistical accuracy of the solutions obtained in PFL remains largely unexplored. As a result, the connections between statistical accuracy, communication efficiency, and their trade-offs are not well understood, leaving a theoretical gap in understanding how to select the optimal personalization degree. In this paper, we fill in this gap by providing a fine-grained theoretical analysis of a widely adopted PFL problem formulation given by (2) that trains simultaneously local and global models. We quantitatively analyze the influence of personalization on both statistical and optimization convergence rates for *each of the local models*. Specifically, our contribution can be summarized as follows:

- **Statistical Accuracy with Optimal Guarantee**. We provide a non-asymptotic statistical convergence rate of the solution of Problem (2), revealing how personalization degree influences the statistical accuracy of each local model. In particular, as the personalization degree increases, the rate approaches to that of pure local training, $\mathcal{O}(1/n)$, where $n$ is the sample size per client. Conversely, decreas-

---

[*]Equal contribution  [1]Department of Statistics, The Pennsylvania State University, University Park, PA 16802, USA  [2]School of Electrical Engineering and Computer Science, The Pennsylvania State University, University Park, PA 16802, USA. Correspondence to: Ying Sun <ybs5190@psu.edu>, Lingzhou Xue <lzxue@psu.edu>.

*Proceedings of the $42^{nd}$ International Conference on Machine Learning*, Vancouver, Canada. PMLR 267, 2025. Copyright 2025 by the author(s).

ing the personalization degree allows the models to utilize information across all clients, achieving a rate closer to $\mathcal{O}(1/(mn) + R^2)$, where $mn$ is the total sample size of all clients and $R$ quantifies the statistical heterogeneity of the local datasets. We further establish the minimax optimality of the derived statistical rates, demonstrating the tightness of our analysis. To the best of our knowledge, this is the first work to achieve such an optimality.

• **Communication Efficiency under Personalization**. On the optimization aspect, we treat Problem (2) as a bi-level problem and propose (stochastic) algorithms with an explicit characterization of computation and communication complexity. We establish that the communication cost of finding an $\varepsilon$-solution is $\mathcal{O}(\kappa \frac{\lambda+\mu}{\lambda+L} \log \frac{1}{\varepsilon})$ and the computation cost in terms of gradient evaluations is $\mathcal{O}(\kappa \log \frac{1}{\varepsilon})$, where $\lambda$ is a parameter determining the personalization degree. This demonstrates that a smaller $\lambda$, corresponding to a higher degree of personalization, reduces communication complexity without incurring additional computational overhead.

• **Accuracy-Communication Tradeoff with Empirical Validation**. Building on our theoretical results for statistical and optimization convergence, we quantitatively characterize the trade-off between statistical accuracy and communication efficiency in PFL and discuss practical insights for selecting the personalization degree. We then conduct numerical studies on logistic regression with synthetic data and real-world data under our assumptions. The results corroborate our theoretical findings. We further test the numerical performance with CNN models, showing that the theoretical results, though established under convexity assumptions, generalize to non-convex settings.

## 2. Related Works

In recent years, FL has become an attractive solution for training models locally on distributed clients, rather than transferring data to a single node for centralized processing (Mammen, 2021; Wen et al., 2023; Beltrán et al., 2023). As each client generates its local data, statistical heterogeneity naturally arises with data being non-identically distributed between clients (Li et al., 2020; Ye et al., 2023). Given the variability of data in a network, model personalization is an appealing strategy used to improve statistical accuracy for each client. Formulations enabling model personalization have been studied independently from multiple fields. For example, meta-learning (Chen et al., 2018; Jiang et al., 2019; Khodak et al., 2019; Fallah et al., 2020) assumes all local models follow a common distribution. By minimizing the average validation loss, these methods aim to learn a meta-model that generalizes well to unseen new tasks. Representation learning (Zhou et al., 2020; Wang et al., 2024a) focuses on a setting where the local models can be represented as the composition of two parts, with one common

to all clients and the other specific to each client. Our work, however, differs in that we focus on a personalized FL setting with a mixture of global and local models. Closely related are multi-task learning methods (Liang et al., 2020) and transfer learning methods (Li et al., 2022; He et al., 2024a;b), but the statistical rate is established for either the average of all models or only the target model. Therefore, it remains unclear how personalization influences the statistical accuracy of each individual local model in these settings.

In the FL community, personalized FL methods can be broadly divided into two main strands. Different from the works mentioned previously, studies here primarily focus on the properties of the iterates generated by the algorithms. One line of work (Arivazhagan et al., 2019; Liang et al., 2020; Singhal et al., 2021; Collins et al., 2021) is based on the representation learning formulation. Another line of work (Smith et al., 2017; Li et al., 2020; Hanzely & Richtárik, 2020; Hanzely et al., 2020; Li & Richtárik, 2024) achieves personalization by relaxing the requirement of learning a common global model through regularization techniques. In particular, algorithms and complexity lower bounds specific to Problem (2) were studied in (Li et al., 2020; Hanzely et al., 2020; T Dinh et al., 2020; Hanzely & Richtárik, 2020; Li et al., 2021), see Table 1 for a detailed comparison. These works study Problem (2) from a pure optimization perspective and have not provided how personalization influences statistical accuracy and, consequently, the trade-off between statistical accuracy and communication efficiency.

There are only a few recent works we are aware of that study the statistical accuracy of regularization-based PFL. Specifically, Cheng et al. (2023) investigates the asymptotic behavior of the (personalized) federated learning under an over-parameterized linear regression model; neither a finite-sample rate nor algorithms are provided. Chen et al. (2023c) studies Problem (2) and establishes a non-asymptotic sample complexity. However, even with some overly strong assumptions, their analysis cannot match the statistical lower bound in most cases. Furthermore, neither of these works reveals the trade-off. While (Bietti et al., 2022) studies the influence of personalization in PFL, their focus is on the trade-off between privacy and optimization error, leaving the accuracy-communication trade-off unexplored.

**Notation.** Denote $[n] := \{1, 2, \cdots, n\}$. $\| \cdot \|_2$ denotes the $\ell_2$ norm for vectors and the Frobenius norm for matrices. For two non-negative sequences $\{a_n\}, \{b_n\}$, we denote $a_n \lesssim b_n$ if $a_n \leq C b_n$ for some constant $C > 0$ when $n$ is sufficiently large. We also use $a_n = \mathcal{O}(b_n)$, whose meaning is the same as $a_n \lesssim b_n$. $\tilde{\mathcal{O}}(\cdot)$ hides logarithmic factors. For two real numbers $a, b$, we let $a \wedge b = \min\{a, b\}$ and $a \vee b = \max\{a, b\}$.

## 3. Preliminaries

We consider an FL setting with $m$ clients. Each client $i$ has a collection of data $S_i = \{z_{ij}\}_{j \in [n_i]}$, where $n_i$ is the sample size of client $i$, with elements i.i.d. drawn from distribution $\mathcal{D}_i$. The data distributions among the clients are possibly heterogeneous. For each client $i$, the ultimate goal is to find a model that minimizes its local risk at the population level, defined as:

$$\boldsymbol{w}_\star^{(i)} \in \arg\min_{\boldsymbol{w}} \mathbb{E}_{z \sim \mathcal{D}_i} \ell(\boldsymbol{w}, z), \tag{1}$$

where $\ell(\boldsymbol{w}, z)$ is a loss function measuring the fitness of $\boldsymbol{w}$ to data point $z$. We call $\boldsymbol{w}_\star^{(i)} \in \mathbb{R}^d$ the ground truth model for client $i$. To understand the influence of model personalization, we study the following widely adopted PFL problem (Hanzely & Richtárik, 2020; Mishchenko et al., 2023; Li et al., 2020):

$$\min_{\substack{\boldsymbol{w}^{(g)} \\ \{\boldsymbol{w}^{(i)}\}_{i=1}^m}} \sum_{i \in [m]} p_i \left( L_i(\boldsymbol{w}^{(i)}, S_i) + \frac{\lambda}{2} \|\boldsymbol{w}^{(g)} - \boldsymbol{w}^{(i)}\|^2 \right), \tag{2}$$

where $\boldsymbol{w}^{(g)}$ and $\boldsymbol{w}^{(i)}$ are respectively the global and $i$-th local model to be learned, $L_i(\boldsymbol{w}, S_i) := \sum_{j=1}^{n_i} \ell(\boldsymbol{w}, z_{ij})/n_i$ is the empirical risk of client $i$ on its local data $S_i$, and the last term is a regularization term with parameter $\lambda$ controlling the personalization degree. The set $\{p_i\}_{i \in [m]}$ is a collection of nonnegative weights with $\sum_{i=1}^m p_i = 1$.

In Problem (2), each client $i$ learns a personalized model $\boldsymbol{w}^{(i)}$ by fitting to its local data $S_i$, while collaboration is achieved by shrinking the local models $\boldsymbol{w}^{(i)}$ towards a common global model $\boldsymbol{w}^{(g)}$. As $\lambda \to 0$, the influence of the global model diminishes, and Problem (2) increasingly behaves like the LocalTrain problem:

$$\text{LocalTrain:} \quad \min_{\boldsymbol{w}^{(i)}} L_i(\boldsymbol{w}^{(i)}, S_i), \quad \forall i \in [m], \tag{3}$$

where each client independently trains its model using only local data. In this case, Problem (2) is fully decoupled into $m$ separate problems, achieving the maximum degree of personalization. On the other hand, as $\lambda \to \infty$, the regularization term shrinks all local models $\boldsymbol{w}^{(i)}$ towards the global model $\boldsymbol{w}^{(g)}$. This corresponds to reducing the personalization degree and eventually pushes Problem (2) to another extreme given by:

$$\text{GlobalTrain:} \quad \min_{\substack{\boldsymbol{w}^{(i)} = \boldsymbol{w}^{(g)}, \\ \forall i \in [m]}} \sum_{i=1}^m p_i L_i(\boldsymbol{w}^{(i)}, S_i), \tag{4}$$

where the knowledge from all clients is pooled to train a single global model. Adjusting $\lambda$, therefore, controls the degree of personalization among clients and thus affects the statistical accuracy of the resulting solution. Noticeably,

the choice of $\lambda$ also potentially affects solving Problem (2) algorithmically in the FL setting. Intuitively, for large $\lambda$, we anticipate more frequent communications among the clients to facilitate collaboration as the local models are more tightly coupled to the global model. In contrast, a smaller $\lambda$ encourages more independent updates and reduces the need for communication, with LocalTrain (cf.(3)) being an edge case where all clients train independently without any communication.

We quantify both the statistical and optimization error of a PFL algorithm applied to Problem (2) by evaluating the Euclidean distance between the algorithm's output after $t$ communication rounds, $\boldsymbol{w}_t^{(i)}$, and the ground truth local model $\boldsymbol{w}_\star^{(i)}$ for each client $i \in [m]$. Let $\widetilde{\boldsymbol{w}}^{(i)}$ denote the solution of $i$-th local model in Problem (2). We have the following decomposition:

$$\|\boldsymbol{w}_t^{(i)} - \boldsymbol{w}_\star^{(i)}\|^2 \le 2\underbrace{\|\boldsymbol{w}_t^{(i)} - \widetilde{\boldsymbol{w}}^{(i)}\|^2}_{\text{optimization error}} + 2\underbrace{\|\widetilde{\boldsymbol{w}}^{(i)} - \boldsymbol{w}_\star^{(i)}\|^2}_{\text{statistical error}}.$$

In the rest of the paper, we first establish the statistical and optimization convergence rates independently. We then integrate these results to formally provide the trade-off between statistical accuracy and communication efficiency under different levels of personalization.

## 4. Convergence Rate Analysis

### 4.1. Effect of Personalization on Statistical Accuracy

In this section, we analyze how the choice of $\lambda$ influences the statistical accuracy of the solution to Problem (2). Recall the ground truth local model $\boldsymbol{w}_\star^{(i)} \in \mathbb{R}^d$ defined in (1). Next, we introduce the measure of statistical heterogeneity and define the parameter space for the statistical estimation problem. Specifically, we consider estimating $\boldsymbol{w}_\star^{(i)}$ from the following parameter space:

$$\mathcal{P}(R) = \left\{ \{\boldsymbol{w}_\star^{(i)}\}_{i=1}^m : \left\| \boldsymbol{w}_\star^{(i)} - \sum_{i=1}^m p_i \boldsymbol{w}_\star^{(i)} \right\|^2 \le R^2, \forall i \in [m] \right\}. \tag{5}$$

In (5), the statistical heterogeneity is measured as the Euclidean distance between the ground truth local models and their weighted average. A larger $R$ indicates a larger difference among the clients' local models, hence stronger statistical heterogeneity. Such a measure is commonly imposed in the existing literature (Li et al., 2023; Chen et al., 2023b;c; Duan & Wang, 2023). More discussion about the parameter space $\mathcal{P}(R)$ can be found in Appendix A.1. We assume the model dimension $d$ is finite.

Under such a parameter space, we aim to investigate the statistical accuracy of the solution to Problem (2) measured

by $\mathbb{E}\|\widetilde{\boldsymbol{w}}^{(i)} - \boldsymbol{w}_\star^{(i)}\|^2$, where the expectation is taken with respect to the joint distribution.

**Remark 1.** *Note that some existing literature adopts alternative metrics, such as individual excess risk (Chen et al., 2023c) to measure the statistical error. These two metrics are equivalent under the strong convexity and smoothness conditions on the loss functions.*

We study Problem (2) under the following regularity conditions.

**Assumption 1** (Smoothness). *The loss function $\ell(\cdot, z)$ is $L$-smooth, i.e. for any $x, y \in \mathbb{R}^d$:*

$$\|\nabla\ell(x, z) - \nabla\ell(y, z)\| \le L\|x - y\|, \ \forall z. \qquad (6)$$

**Assumption 2** (Strong Convexity). *The empirical loss $L_i(\cdot, S_i)$ is $\mu$-strongly convex, i.e., for any $x, y \in \mathbb{R}^d$ and $i \in [m]$:*

$$L_i(x, S_i) \ge L_i(y, S_i) + \langle\nabla L_i(y, S_i), x - y\rangle + \frac{\mu}{2}\|x - y\|^2. \qquad (7)$$

**Assumption 3** (Bound Gradient Variance at Optimum). *There exists a nonnegative constant $\rho$ such that $\mathbb{E}_{z\sim\mathcal{D}_i}\|\nabla\ell(\boldsymbol{w}_\star^{(i)}, z)\|^2 \le \rho^2$ for all $i \in [m]$.*

The strong convexity and smoothness assumptions are used to establish the estimation error and are widely adopted in the theoretical analysis of regularization-based PFL (T Dinh et al., 2020; Deng et al., 2020; Hanzely & Richtárik, 2020; Hanzely et al., 2020) and FL (Chen et al., 2023c; Cheng et al., 2023). Assumption 3 is used to quantify the observation noise, and is also standard in the literature (Duan & Wang, 2023; Chen et al., 2023c).

For simplicity, we assume $p_i = 1/m$ and $n_i = n$ for all $i \in [m]$. The following theorem provides an upperbound of the estimation error $\mathbb{E}\|\widetilde{\boldsymbol{w}}^{(i)} - \boldsymbol{w}_\star^{(i)}\|^2$ as a function of $\lambda$.

**Theorem 1.** *Suppose Assumption 1, 2 and 3 hold and consider the parameter space $\mathcal{P}(R)$ given by (5). The local models $\{\widetilde{\boldsymbol{w}}^{(i)}\}_{i=1}^m$ obtained by solving Problem (2) satisfy*

$$\mathbb{E}\left\|\widetilde{\boldsymbol{w}}^{(i)} - \boldsymbol{w}_\star^{(i)}\right\|^2$$
$$\le \min\left\{\frac{48\rho^2}{\mu^2}\frac{1}{N} + \left(\frac{48L^2}{\mu^2} + 3\right)R^2 + \frac{1}{q_1(\lambda)}, \qquad (8)\right.$$
$$\left.\frac{4\rho^2}{\mu^2}\frac{1}{n} + \left[\left(\frac{4\rho^2}{\mu^3}\frac{1}{n} + \frac{4}{\mu}R^2\right)\lambda + \frac{8}{\mu^2}R^2\lambda^2\right]\right\},$$

*where $N = mn$ and $q_1(\lambda)$, defined in (70), is a monotonically increasing function of $\lambda$ with $\lim_{\lambda\to\infty} q_1(\lambda) = \infty$.*

See Appendix A.3 for the proof. The bound given by (8) consists of two terms. The first term decreases as $\lambda$ increases. When $\lambda \to \infty$, the term $[q_1(\lambda)]^{-1} \to 0$ and the

first term approaches $\mathcal{O}(1/N + R^2)$. Here, $\mathcal{O}(1/N)$ corresponds to the sample complexity one could obtain if the data distributions of the $m$ clients are homogeneous, reflecting the benefit of collaborative training. Term $\mathcal{O}(R^2)$ reflects the negative impact due to the bias introduced by statistical heterogeneity. In contrast, the second term in (8) increases with $\lambda$, showing that a smaller $\lambda$ leads to a rate closer to $\mathcal{O}(1/n)$ corresponding to that of pure local training. This rate is independent of $R$, making it robust to high data heterogeneity but at the expense of reduced sample efficiency. Together, these results provide a continuous and quantitative characterization of how the personalization degree, governed by $\lambda$, determines the statistical accuracy of the solution $\widetilde{\boldsymbol{w}}^{(i)}$.

**Minimax-optimal Statistical Accuracy.** To demonstrate the tightness of our analysis, we now show that as a direct implication of Theorem 1, the local models $\widetilde{\boldsymbol{w}}^{(i)}$ are rate-optimal. Denoting $\kappa = L/\mu$ as the conditional number, we have the following corollary.

**Corollary 1.** *Suppose Assumption 1, 2 and 3 hold and consider the parameter space $\mathcal{P}(R)$ given by (5). If setting $\lambda \ge \max\left\{64\kappa^2 L, (2\kappa\vee 5)\,\mu\frac{2L^2R^2+\rho^2/n}{L^2R^2+\rho^2/N}\right\} - \mu$ when $R \le \frac{1}{\sqrt{n}}$, and $\lambda \le \frac{\rho^2}{n\mu R^2}$ when $R > \frac{1}{\sqrt{n}}$, then for all $i \in [m]$, the local model $\widetilde{\boldsymbol{w}}^{(i)}$ obtained by solving Problem (2) satisfy*

$$\mathbb{E}\left\|\widetilde{\boldsymbol{w}}^{(i)} - \boldsymbol{w}_\star^{(i)}\right\|^2 \le C_3\frac{1}{N} + C_4\left(R^2 \wedge \frac{1}{n}\right), \qquad (9)$$

*where constants $C_3$ and $C_4$ are defined in (91).*

See Appendix A.3 for the proof. In addition, Theorem 8 in Chen et al. (2023c) states for all $i \in [m]$ and any estimator $\hat{\boldsymbol{w}}^{(i)}$ that is a measurable function of data $\{S_i\}_{i=1}^m$, we have

$$\inf_{\hat{\boldsymbol{w}}^{(i)}}\sup_{\{\boldsymbol{w}_\star^{(i)}\}_{i=1}^m\in\mathcal{P}(R)}\mathbb{E}\left\|\hat{\boldsymbol{w}}^{(i)} - \boldsymbol{w}_\star^{(i)}\right\|^2 \gtrsim \frac{1}{N} + R^2 \wedge \frac{1}{n}.$$

Therefore, Corollary 1 shows that with an appropriate choice of $\lambda$, the established rate achieves the minimax lower bound. A detailed comparison with prior results can be found in Appendix A.5, along with a discussion of several novel techniques developed to derive our result. While this result focuses on statistical accuracy and is not directly tied to the accuracy-communication trade-off, it highlights the tightness of our analysis. Furthermore, the techniques introduced to obtain the results are of independent interest. To the best of our knowledge, this is the first work to establish minimax-optimality for the solutions of Problem (2).

**Remark 2** (Technical Novelty of The Analysis). *Unlike previous results (Chen et al., 2023c) which are algorithm-dependent, our analysis does not rely on any algorithm but directly tackles the objective function, establishing rates using purely the properties of the loss. Specifically, using the*

*strong convexity of the loss function, we first proved the estimation error is bounded by the gradient norm and an extra statistical heterogeneity term controlled by $\lambda$, as detailed in Equation (33). This allows us to show for large $R$, a small $\lambda$ can be chosen to yield rate $O(1/n)$ and match one of the worst cases in the lower bound. For the complementary case $R \leq 1/\sqrt{n}$, we leveraged the GlobalTrain solution $\widetilde{w}_{GT}$ as a bridge and proved the solutions of Problem (2) to $\widetilde{w}_{GT}$ are bounded by a term inversely proportional to $\lambda$ (cf. Lemma 4), implying that they can be made arbitrarily close to $\widetilde{w}_{GT}$ by setting $\lambda$ small. This, together with the rate of $\widetilde{w}_{GT}$, yields a rate of $O(1/N + R^2)$. Combining the two cases, we proved minimax optimality. Not only each piece of the above results are new, but more importantly, identifying they are the key ingredients to show the solution of (2) is minimax optimal, are the technical novelties of this proof.*

Notice that Theorem 1 and Corollary 1 focus on establishing the statistical rate for the solution in Problem (2), $\widetilde{w}^{(i)}$. To fully quantify the error of the output of PFL algorithms, we need to establish the optimization error from the algorithm as well, as detailed in the next section.

### 4.2. Effect of Personalization on Communication Efficiency

In this section, we provide an FL algorithm for solving Problem (2) along with its convergence analysis, showing the impact of the personalization degree on communication and computation complexity. Notice that if we define the local objective of each client $i$ as

$$h_i(\boldsymbol{w}^{(i)}, \boldsymbol{w}^{(g)}) := L_i(\boldsymbol{w}^{(i)}, S_i) + \frac{\lambda}{2}\|\boldsymbol{w}^{(g)} - \boldsymbol{w}^{(i)}\|^2,$$

then Problem (2) can be rewritten in the following bilevel (iterated minimization) form:

$$\min_{\boldsymbol{w}^{(g)}} F(\boldsymbol{w}^{(g)}) := \frac{1}{m}\sum_{i=1}^{m} F_i(\boldsymbol{w}^{(g)}),$$

$$\text{where} \quad F_i(\boldsymbol{w}^{(g)}) := \min_{\boldsymbol{w}^{(i)}} h_i(\boldsymbol{w}^{(i)}, \boldsymbol{w}^{(g)}). \tag{10}$$

The reformulation (10) has a finite-sum minimization structure, with each component $F_i$ being the Moreau envelope (Moreau, 1965; Yosida, 1964) of $L_i$. Let $\boldsymbol{w}_\star^{(i)}(\boldsymbol{w}^{(g)})$ be the minimizer of $h_i(\,\cdot\,, \boldsymbol{w}^{(g)})$, i.e.,

$$\boldsymbol{w}_\star^{(i)}(\boldsymbol{w}^{(g)}) = \text{prox}_{L_i/\lambda}(\boldsymbol{w}^{(g)})$$
$$:= \arg\min_{\boldsymbol{w}^{(i)}} h_i(\boldsymbol{w}^{(i)}, \boldsymbol{w}^{(g)}). \tag{11}$$

The following lemmas provide properties of $F_i$, $h_i$ and $\boldsymbol{w}_\star^{(i)}$, instrumental to the algorithm design and rate analysis. The proof can be found in Appendix B.2.

**Lemma 1.** *Under Assumption 1 and 2, $F_i$ is $\mu_g$-strongly convex and $L_g$-smooth, with $\mu_g = \frac{\lambda\mu}{\lambda+\mu}$ and $L_g = \frac{\lambda L}{\lambda+L}$, each $h_i$ is $\mu_\ell$-strongly convex and $L_\ell$-smooth, with $\mu_\ell = \mu + \lambda$ and $L_\ell = L + \lambda$, and the mapping $\boldsymbol{w}_\star^{(i)} : \mathbb{R}^d \to \mathbb{R}^d$ is $L_w$-Lipschitz with $L_w = \frac{\lambda}{\lambda+\mu}$.*

**Lemma 2** (Lemaréchal & Sagastizábal (1997))**.** *Under Assumption 2, each $F_i : \mathbb{R}^d \to \mathbb{R}$ is continuously differentiable, and the gradient is given by $\nabla F_i(\boldsymbol{w}^{(g)}) = \lambda(\boldsymbol{w}^{(g)} - \boldsymbol{w}_\star^{(i)}(\boldsymbol{w}^{(g)}))$.*

Lemma 1 and Lemma 2 suggest applying a simple gradient algorithm to optimize $\boldsymbol{w}^{(g)}$:

$$\begin{aligned}
\boldsymbol{w}_{t+1}^{(g)} &= \boldsymbol{w}_t^{(g)} - \gamma \cdot \nabla F(\boldsymbol{w}_t^{(g)}) \\
&= \boldsymbol{w}_t^{(g)} - \gamma\lambda \cdot \frac{1}{m}\sum_{i=1}^{m}\left(\boldsymbol{w}_t^{(g)} - \boldsymbol{w}_\star^{(i)}(\boldsymbol{w}_t^{(g)})\right),
\end{aligned} \tag{12}$$

where $\boldsymbol{w}_t^{(g)}$ is the update at the communication round $t$, $\gamma > 0$ is a step size to be properly set (cf. Theorem 2). To implement (12), in each communication round $t$ the server broadcasts $\boldsymbol{w}_t^{(g)}$ to all clients, then each client $i$ updates its local model $\boldsymbol{w}_\star^{(i)}(\boldsymbol{w}_t^{(g)})$ and uploads to the server.

Note that executing the update (12) requires each client $i$ computing the minimizer $\boldsymbol{w}_\star^{(i)}(\boldsymbol{w}_t^{(g)})$. In general, the subproblem (11) does not have a closed-form solution. Therefore, computing the exact gradient $\nabla F$ will incur a high computation cost as well as high latency for the learning process. Leveraging recent advancements in bilevel optimization algorithm design (Ji et al., 2022), we address this issue by approximating $\boldsymbol{w}_\star^{(i)}(\boldsymbol{w}_t^{(g)})$ with a finite number of $K$ gradient steps. Specifically, we let each client $i$ maintain a local model $\boldsymbol{w}_{t,k}^{(i)}$. Per communication round $t$, $\boldsymbol{w}_{t,k}^{(i)}$ is initialized to be $\boldsymbol{w}_{t,0}^{(i)} = \boldsymbol{w}_{t-1,K}^{(i)}$ as a warm start, and updated according to

$$\boldsymbol{w}_{t,k+1}^{(i)} = \boldsymbol{w}_{t,k}^{(i)} - \eta\nabla h_i(\boldsymbol{w}_{t,k}^{(i)}, \boldsymbol{w}_t^{(g)}), \tag{13}$$

for $k = 0, \ldots, K-1$, where $\nabla h_i(\boldsymbol{w}_{t,k}^{(i)}, \boldsymbol{w}_t^{(g)})$, for notation simplicity, denotes the partial gradient of $h_i$ with respect to $\boldsymbol{w}^{(i)}$. The overall procedure is summarized in Alg. 1, see Appendix B.1.

**Theorem 2.** *Suppose Assumptions 1 and 2 hold. Let $\{\boldsymbol{w}_t^{(g)}\}_{t\geq 0}$ and $\{\boldsymbol{w}_{t,K}^{(i)}\}_{t\geq 0}$ be the sequence generated by Algorithm 1 with $\gamma < 1/L_g$, $\eta \leq 1/L_\ell$, and the inner loop iteration number satisfying*

$$\left(2 + 64L_w^2(1/\mu_g)^2\lambda^2\right)(1 - \eta\mu_\ell)^K \leq (1 - \gamma L_g)^4, \tag{14}$$

*then $\boldsymbol{w}_t^{(g)}$ converges to $\widetilde{w}^{(g)}$ linearly at rate $1 - (\gamma\mu_g)/2 - (\gamma\mu_g)^2/2$, and $\boldsymbol{w}_{t,0}^{(i)}$ converges to $\widetilde{w}^{(i)}$ linearly at the same rate for any $i \in [m]$.*

The proof of Theorem 2 can be found in Appendix B.3 and B.3.2. Theorem 2 shows that with sufficiently small step sizes, both the local and global models converge linearly. The personalization degree, controlled by $\lambda$, plays a critical role in influencing the algorithm's communication efficiency. A small $\lambda$ (indicating higher personalization) yields a small $L_g$, and thus permits a larger choice of the step size $\gamma$. By the expression of the convergence rate $1 - (\gamma\mu_g)/2 - (\gamma\mu_g)^2/2$, we can see less communication round would be required to reach an $\varepsilon$-optimal solution. However, increasing $\gamma$ will also decrease the right hand side of (14), implying that a larger number of local gradient steps $K$ should be taken to fulfill the condition. To provide a more concrete characterization of these dynamics, we present Corollary 2, which quantifies the communication and computation complexities under specific choices of tuning parameters. The proof of Corollary 2 can be found in Appendix B.4.

**Corollary 2.** *In the setting of Theorem 2, if we further choose the step size $\eta = (\lambda + L)^{-1}$, $\gamma = (\lambda + L)/(2\lambda L)$, and the inner loop iteration number $K = \tilde{\mathcal{O}}((\lambda + L)/(\lambda + \mu))$, then $\boldsymbol{w}_t^{(g)}$ converges to $\widetilde{\boldsymbol{w}}^{(g)}$ linearly at rate $1 - (\lambda + L)/(4\kappa(\lambda + \mu))$, and $\boldsymbol{w}_{t,0}^{(i)}$ converges to $\widetilde{\boldsymbol{w}}^{(i)}$ linearly at the same rate for any $i \in [m]$. Thus the communication and computation complexity for Algorithm 1 to find an $\varepsilon$-solution (i.e., $\|\boldsymbol{w}_T^{(g)} - \tilde{\boldsymbol{w}}^{(g)}\|^2 \leq \varepsilon$ and $\|\boldsymbol{w}_{T,0}^{(i)} - \tilde{\boldsymbol{w}}^{(i)}\|^2 \leq \varepsilon$) are as follows ($\kappa := L/\mu$):*

*(i) # communication rounds $= \mathcal{O}\left(\kappa \cdot \dfrac{\lambda + \mu}{\lambda + L} \cdot \log \dfrac{1}{\varepsilon}\right)$,*

*(ii) # gradient evaluations $= \tilde{\mathcal{O}}\left(\kappa \cdot \log \dfrac{1}{\varepsilon}\right)$.*

Corollary 2 clearly shows the influence of the personalization degree $\lambda$ on the communication cost. Specifically, as $\lambda$ increases from 0 to $\infty$, the communication complexity increases from $\mathcal{O}(\log(1/\epsilon))$ to $\mathcal{O}(\kappa \log(1/\epsilon))$. This fact corroborates our intuition that a higher degree of collaboration requires more communication resources. Moreover, the result given by Corollary 2 also indicates that the total computation cost is independent of $\lambda$. As such, we can see that model personalization can provably reduce the communication cost without any extra computation overhead. Notice that although personalization indeed enhances communication efficiency, doing so may deteriorate the statistical accuracy of the local models. In the next section, we will discuss in detail the trade-off between communication and statistical accuracy.

We also provide an extension of Alg. 1 to the stochastic setting using mini-batch stochastic gradients for the local updates. The proposed algorithm, termed `FedCLUP`, is given in Appendix B.5. Following similar argument to Theorem 2, Theorem 3 in Appendix B.5 establishes the convergence

rate of `FedCLUP`, showing that it takes $\mathcal{O}\left(\kappa\frac{\lambda+\mu}{\lambda+L} \cdot \log \frac{1}{\varepsilon}\right)$ rounds of communication to obtain an output $\boldsymbol{w}_t^{(g)}$ within an error ball of size $\left(\mathcal{O}(\varepsilon) + \frac{\sigma^2}{\mu^2 Bm}\right)$ around $\widetilde{\boldsymbol{w}}^{(g)}$. Here $\sigma^2$ is the variance of the stochastic gradient defined in Assumption 4 in Appendix B.5 and $B$ is the mini-batch size. Therefore, the influence of $\lambda$ on communication complexity is consistent with the noiseless setting presented in Corollary 2. Additionally, one may note that in the stochastic setting, the error due to the stochastic noise scales inversely with the number of participating clients, showing the advantage of client collaboration in reducing the variance of $\boldsymbol{w}_t^{(g)}$.

## 5. Communication-Accuracy Trade-off

Building on the results established in Section 4.1 and 4.2, we cast insights on the trade-off between communication efficiency and statistical accuracy, and discuss its implications on real-world practice.

**Corollary 3.** *Under the conditions of Theorem 1 and Corollary 2, $\boldsymbol{w}_{T,K}^{(i)}$ generated by Algorithm 1 satisfies*

$$
\mathbb{E}\left\|\boldsymbol{w}_{T,K}^{(i)} - \boldsymbol{w}_\star^{(i)}\right\|^2
$$
$$
\leq 2\left(1 - \frac{1}{4\kappa} - \frac{L - \mu}{4\kappa(\lambda + \mu)}\right)^T \mathbb{E}\left\|\boldsymbol{w}_{0,0}^{(i)} - \widetilde{\boldsymbol{w}}^{(i)}\right\|^2 \quad (15)
$$
$$
+ \mathcal{O}\left[\left(\frac{1}{N} + R^2 + \frac{1}{q_1(\lambda)}\right) \wedge \left(\frac{1}{n} + q_2(\lambda)\right)\right]
$$

*for any $i \in [m]$, where $q_1(\lambda)$ and $q_2(\lambda)$, defined in (70) and (67), respectively, are monotonically increasing functions of $\lambda$ with $\lim_{\lambda \to \infty} q_1(\lambda) = \infty$ and $\lim_{\lambda \to 0} q_2(\lambda) = 0$.*

See Appendix C for the proof. When collaborative learning is beneficial, i.e., when $R^2 \lesssim 1/n$, in this case, Corollary 1 establishes that the minimax-optimal statistical rate is $\mathcal{O}(1/N + R^2)$. As $\lambda$ increases, $[q_1(\lambda)]^{-1}$ monotonically decreases to zero, leading to improved statistical accuracy, approaching the optimal accuracy. However, since the optimization error in (15) is monotonically increasing with $\lambda$, a higher degree of collaboration will also results in slower convergence over communication rounds. Consequently, Corollary 3 implies that when the data heterogeneity is relatively low, increasing personalization will improve communication efficiency but at the expense of lower statistical accuracy, and vice versa. Such an opposite effect of the personalization degree on the statistical accuracy and communication efficiency leads to an accuracy-communication trade-off.

As the optimization error vanishes when $T \to \infty$, the statistical error dominates the total error in Corollary 3. However, the associated communication cost becomes prohibitively large to achieve such an error. Therefore, balancing the

*Table 1.* Comparison of the existing works analyzing Problem (2) (cvx denotes convexity and s-cvx denotes strong convexity).

| Methods | Convexity | Bounded Domain | Computation Cost per Round | Communication Cost | Statistical Error |
|---|---|---|---|---|---|
| FedProx (Li et al., 2020) | cvx | ✗ | —[1] | $\mathcal{O}(\Delta/(\rho\varepsilon))$ | ✗ |
| pFedMe (T Dinh et al., 2020) | s-cvx | ✗ | —[2] | $\mathcal{O}(1/(mR\varepsilon))$ | ✗ |
| L2GD (Hanzely & Richtárik, 2020) | s-cvx | ✗ | $\mathcal{O}(1+\frac{L}{\lambda})$ | $\mathcal{O}(\frac{2\lambda\kappa}{\lambda+L}\log(\frac{1}{\varepsilon}))$ | ✗ |
| AL2SGD (Hanzely et al., 2020) | s-cvx | ✗ | $\mathcal{O}\left(\frac{(m+\sqrt{m(L+\lambda)/\mu})}{\sqrt{\min\{L,\lambda\}/\mu}}\right)$ | $\mathcal{O}(\sqrt{\min\{L,\lambda\}/\mu}\log\frac{1}{\varepsilon})$ | ✗ |
| Algorithm 3 (Chen et al., 2023c) | s-cvx | ✔ | $\mathcal{O}(\lambda/\varepsilon)$ | $\mathcal{O}((\lambda\vee 1)/\varepsilon)$ | ✔ |
| **FedCLUP** | s-cvx | ✗ | $\mathcal{O}(\frac{\lambda+L}{\lambda+\mu})$ | $\mathcal{O}(\kappa\frac{\lambda+\mu}{\lambda+L}\log\frac{1}{\varepsilon})$ | ✔ |

[1] Controlled by the precision of inexact solution. $\rho>0$ measures the subproblem solution accuracy.
[2] Related to the subiteration number $R$ and the precision $v$.

optimization and statistical errors is crucial for achieving a target total error with efficient resource usage. Under the influence of the personalization degree, a practical guideline emerges: when collaborative learning is beneficial (i.e., $R^2 \le 1/\sqrt{n}$), to achieve a target magnitude of total error, we control the optimization errors at the same magnitude as the statistical error. Specifically, we increase $\lambda$ to reduce the statistical error to the same magnitude of the total error, and then gradually increase the communication rounds $T$ until the optimization error reaches the same magnitude as well. Since the communication efficiency will decrease will an increasing $\lambda$, the above strategy can achieve a given target total error at the highest possible communication efficiency. Later, we will also show that, with the optimal choice of personalization degree, FedCLUP improves communication efficiency significantly compared with GlobalTrain and achieve a total error smaller than LocalTrain in Section 6.

**Comparison with Existing Literature.** As reported in Table 1, our work is the first to quantitatively characterize how changing the personalization degree leads to the trade-off between communication cost and statistical accuracy. In contrast, most prior studies either do not explicitly analyze statistical convergence or fail to provide a tight convergence guarantee for optimization and statistical error. For example, Chen et al. (2023c) attempted to establish both optimization and statistical rates simultaneously; however, when $\lambda \to \infty$, Problem (2) reduces to GlobalTrain, yet their results incorrectly imply that the communication and computation costs diverge to infinity.

# 6. Empirical Study

In this section, we provide empirical validation for our theoretical results, evaluating Problem (2) on both synthetic and real datasets with convex and non-convex loss functions. We first present the experimental setup, and then analyze the impact of personalization on statistical accuracy and communication efficiency, and conclude by analyzing the trade-off between the two. Details about the experimental setup are available in Section 6.1 and Appendix D, and the complete anonymized codebase is accessible at https://github.com/ZLHe0/fedclup.

## 6.1. Experimental Details

**Synthetic Dataset.** We generate two cases of synthetic dataset: (1)As our theoretical analysis is established under strong convexity, first, we consider an overdetermined linear regression task, where the choice of hyparamter is strictly followed Corollary 3. (2) We follow a similar procedure to prior works (Li et al., 2020) but with some modifications to align with the setup in this paper. Specifically, for each client we generate samples $(\boldsymbol{X}_k, \boldsymbol{y}_k)$, where the labels $\boldsymbol{y}_k$ are produced by a logistic regression model $\boldsymbol{y}_k = \mathrm{argmax}(\sigma(\boldsymbol{w}_k^\top \boldsymbol{X}_k))$, with $\sigma$ being the sigmoid function. The feature vectors $\boldsymbol{X}_k$ are drawn from a multivariate normal distribution $\mathcal{N}(\boldsymbol{v}_k, \boldsymbol{\Sigma})$, where $v_k \sim \mathcal{N}(0,1)$ and the covariance matrix $\boldsymbol{\Sigma}$ follows a diagonal structure $\boldsymbol{\Sigma}_{j,j} = j^{-1.2}$. The heterogeneity is introduced by sampling the model weights $\boldsymbol{w}_k$ for each client from a normal distribution $\mathcal{N}(0, R)$, where $R \in [0,3]$ controls the statistical heterogeneity across clients' data.

**Real Dataset.** We use the MNIST, EMNIST, CIFAR10, Sent140 an CelebA datasets for real data analysis. For MNIST, EMNIST and CIFAR10, they aren't naturally partitioned datasets. Therefore, following Li et al. (2020), to impose statistical heterogeneity, we distribute the data across clients in a way that each client only has access to a fixed number of classes. The fewer classes each client has access to, the higher the statistical heterogeneity. Details on data preprocessing and heterogeneity settings for each dataset are provided in Appendix D. For Sent140 and CelebA datasets, as they are naturally partitioned, we don't need to create the data heterogeneity and directly apply our algorithm to these datasets with advanced models, like CNN and LSTM model (Sak et al., 2014). We strictly follow the previous work (Li et al., 2021) and (Duan et al., 2021) to pre-process the Sent140 and CelebA dataset, respectively.

**Implementation and Evaluation.** For each setup, we evaluate `FedCLUP` with three different degree of personalization: low, medium, and high, with specific personalization degree varying based on the dataset (details provided in Appendix D). For the synthetic dataset, in alignment with theorem 2, we implement `FedCLUP` using a global step size $\gamma = (\lambda + L)/(\lambda L)$ and a local step size $\eta = (L + \lambda)^{-1}$.

| Dataset | LocalTrain | FedCLUP | | | pFedMe | | | GlobalTrain |
|---|---|---|---|---|---|---|---|---|
| | | High | Medium | Low | High | Medium | Low | |
| MNIST Logit | 0.7828 | 0.8123 | 0.8213 | 0.8312 | 0.8001 | 0.8239 | 0.8291 | 0.8391 |
| MNIST CNN2 | 0.8194 | 0.8753 | 0.8893 | 0.9032 | 0.8756 | 0.8771 | 0.8749 | 0.9098 |
| MNIST CNN5 | 0.7633 | 0.9091 | 0.9102 | 0.9421 | 0.9340 | 0.9385 | 0.9440 | 0.9433 |
| EMNIST CNN5 | 0.6111 | 0.6278 | 0.6415 | 0.6672 | 0.6345 | 0.6532 | 0.6717 | 0.6611 |
| CIFAR10 CNN3 | 0.6238 | 0.6102 | 0.6712 | 0.7302 | 0.6744 | 0.7443 | 0.7732 | 0.8041 |

*Table 2.* Test accuracy of algorithms across datasets under varying personalization degrees. "Logit" refers to logistic regression, while "CNN-k" denotes a convolutional neural network with k convolutional layers. "Low," "Medium," and "High" indicate low, moderate, and high personalization levels, respectively. As the personalization degree decreases, the statistical accuracy of `FedCLUP` and `pFedMe` transitions from resembling LocalTrain to GlobalTrain.

For the real dataset, since $L$ is unknown, we implement the same algorithm with a global step size $\gamma$ set to $1/\lambda$, while the local step size is determined via grid search. In terms of evaluation, for the synthetic dataset, we evaluate by tracking the ground truth models and measuring error as the distance from these ground true models. For the real datasets, we report training loss and testing accuracy. Additional details, including hyperparameter settings and evaluation metric definition, can be found in Appendix D.

### 6.2. Results

**Effect of Personalization on Statistical Accuracy.** We compare FedCLUP, under different personalization degrees, with GlobalTrain, LocalTrain and `pFedMe` (T Dinh et al., 2020), an alternative algorithm for solving Problem (2). Implementation details are provided in Appendix D. All methods are run until stable convergence, and their test accuracy is reported in Table 2 in the low heterogeneity setting. Table 2 shows that as the personalization degree decreases and collaborative learning increases, the solution of Problem (2) becomes closer to GlobalTrain, leading to improved statistical accuracy due to increased information sharing across clients. A similar trend is observed for both `FedCLUP` and `pFedMe` across different datasets and models, showing that under low heterogeneity, *increased collaboration improves statistical accuracy*.

**Effect of Personalization on Communication Efficiency.** Figure 2 investigates how personalization impacts communication efficiency in `FedCLUP` by analyzing the benefit of increasing local updates under different personalization levels. In the low-personalization setting (left column), more local updates significantly accelerate convergence, reducing the reliance on frequent communication. However, in the high-personalization setting (right column), increasing local updates has a limited effect on convergence, indicating that frequent communication is essential for effective learning. This demonstrates that higher personalization requires more communication rounds to achieve comparable performance, aligning with our theoretical findings, show-

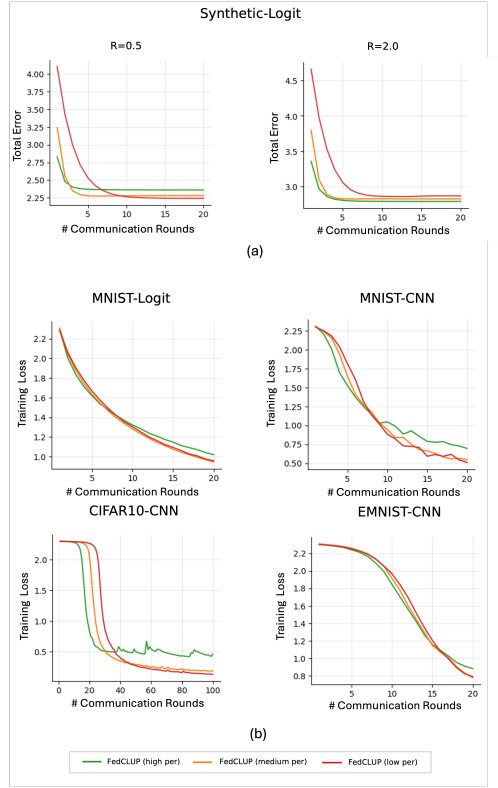

*Figure 1.* Total error and training loss of `FedCLUP` with varying personalization degrees. Total error (y-axis in (a)) quantifies the distance between the model at each communication round and the ground truth model, i.e. $\|\boldsymbol{w}_{T,K}^{(i)} - \boldsymbol{w}_{\star}^{(i)}\|^2$. (a) compares total error under low ($R = 0.5$) and high ($R = 2.0$) heterogeneity. (b) presents training loss across different datasets and models under low heterogeneity. Synthetic-Logit and MNIST-Logit represent logistic regression on synthetic and MNIST data, respectively, while MNIST-CNN, CIFAR10-CNN, and EMNIST-CNN represent CNN models trained on MNIST, CIFAR-10, and EMNIST datasets. In a low-heterogeneity setting, higher personalization (high per) accelerates convergence, while lower personalization (low per) improves final error. In a high-heterogeneity setting, higher personalization achieves both smaller error and faster convergence.

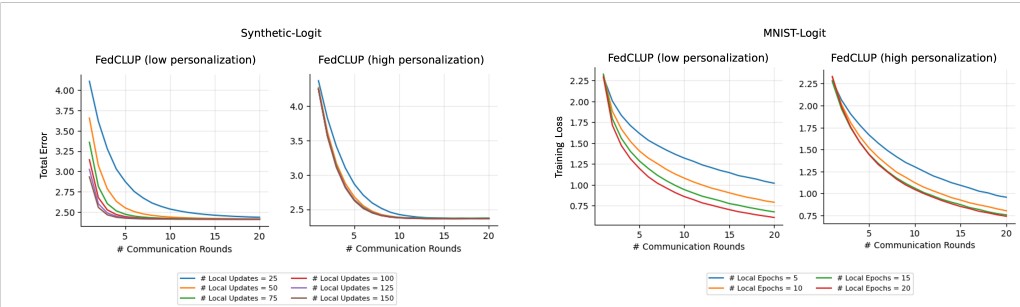

*Figure 2.* Effect of personalization on communication efficiency in `FedCLUP` for Synthetic-Logit (left) and MNIST-Logit (right). Each subfigure compares the impact of varying the number of local updates per communication round under different personalization levels.

ing that *increased personalization improves communication efficiency*.

**The Trade-off under Personalization.** The above two observations imply that there shall be a accuracy-communication trade-off in choosing the personalization degree to achieve the smallest total error under given communication budget (See Figure 1). To further demonstrate this point, in the left part of Figure 3, we compare the total error over iterations among LocalTrain, GlobalTrain, and FedCLUP under varying levels of personalization. Once the algorithms converge, for any given total error, there exists a specific degree of personalization in FedCLUP that incurs the least communication cost among all considered methods. This observation supports the conclusion in Corollary 3 that a unique personalization degree exists in FedCLUP to achieve communication efficiency for a fixed total error. Furthermore, we observe that no fixed personalization degree consistently outperforms others across all error levels. This implies that adaptively adjusting the personalization degree is essential for FedCLUP to maintain communication efficiency across different total error regimes. To further validate this finding, we conduct the same experiment on real-world datasets. Results on the CelebA and Sent140 datasets are presented in Figures 9 and 10, respectively. Additionally, we observe a similar phenomenon in pFedMe, another method solving Problem 2, when we vary its personalization degree. Witnessing the insightful results, it motivates us to adaptively change the personalization degree in FedCLUP for achieving communication efficiency over different total errors.

**Potential Solution in Practice.** After understanding the trade-off under personalization, we propose a dynamic tuning strategy for the optimal $\lambda$ with communication efficiency(See Figure 3 Right ). Beginning with a small $\lambda$ for efficiency, as soon as the validation performance plateaus or the statistical error stops improving significantly, we gradually increase $\lambda$. This allows the model to benefit from enhanced generalization through increased collaboration at the expense of slightly higher communication costs. By progressively adjusting in this way, one can finally identify the optimal personalization degree that balances the trade-off and minimizes the total communication cost required to meet a desired performance threshold.

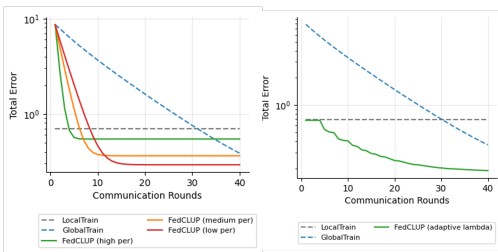

*Figure 3.* Total error versus communication rounds. **Left:** Comparison among LocalTrain, GlobalTrain, and FedCLUP with varying levels of personalization. For a given target total error, an optimal $\lambda$ exists that has the minimum communication rounds. **Right:** A dynamic $\lambda$ strategy: we begin with local training (highest personalization degree), and once the decrease of the validation error slows down, we gradually decrease the personalization degree. Comparing the right figure with the left one shows that the dynamic strategy approximates the optimal personalization level needed to meet a desired error threshold with minimal communication rounds, demonstrating the practical effectiveness of tuning personalization dynamically according to the accuracy-communication trade-off.

## 7. Conclusion

In this paper, we provide a precise theoretical characterization of the statistical and optimization convergence of a widely used personalized federated learning problem. Our analysis reveals that when collaborative learning is beneficial, increasing personalization reduces communication complexity but comes at the cost of statistical accuracy due to limited information sharing across clients. We then validate our theoretical findings across convex and non-convex settings, multiple datasets, and different model architectures.

## Acknowledgment

This work was supported in part by the U.S. National Science Foundation (NSF) grant CCF-2007823 and National Institutes of Health (NIH) grants R01AI170249 and R01GM152812.

## Impact Statement

This paper aims to contribute to the advancement of the field of Machine Learning. While our work has the potential for various societal implications, we believe none require specific emphasis in this context.

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

# A. Proof of statistical converngence

To facilitate our analysis, we denote

$$\tilde{\Delta}_{\text{stat}}^{(g)} = \boldsymbol{w}_\star^{(g)} - \widetilde{\boldsymbol{w}}^{(g)}, \tag{16}$$

$$\tilde{\Delta}_{\text{stat}}^{(i)} = \boldsymbol{w}_\star^{(i)} - \widetilde{\boldsymbol{w}}^{(i)} \quad \forall i \in [m], \tag{17}$$

where $\boldsymbol{w}_\star^{(g)} := \sum_{i \in [m]} p_i \boldsymbol{w}_\star^{(i)}$, $\tilde{\Delta}_{\text{stat}}^{(g)}$ is the difference between the true global model and the optimal solution of the global model in Problem (2) and $\tilde{\Delta}_{\text{stat}}^{(i)}$ is the difference between the true local model and the optimal solution of the local model in (2) for any $i \in [m]$. The statistical error bound is established as $\mathbb{E}\|\tilde{\Delta}_{\text{stat}}^{(g)}\|^2$ and $\mathbb{E}\|\tilde{\Delta}_{\text{stat}}^{(i)}\|^2$, where the expectation is taken over all the data. Notably, we should stress that the definition of $\boldsymbol{w}_\star^{(g)}$ is just to serve as a bridge to establish the convergence rate of local models, and it is not our focus in the theoretical analysis.

We also denote

$$\boldsymbol{\delta}_\star^{(i)} := \boldsymbol{w}_\star^{(g)} - \boldsymbol{w}_\star^{(i)}, \quad \tilde{\boldsymbol{\delta}}^{(i)} := \tilde{\boldsymbol{w}}^{(g)} - \tilde{\boldsymbol{w}}^{(i)}, \tag{18}$$

where $\boldsymbol{\delta}_\star^{(i)}$ measures the difference between the true global model $\boldsymbol{w}_\star^{(g)}$ and the true local model $\boldsymbol{w}_\star^{(i)}$, and $\tilde{\boldsymbol{\delta}}^{(i)}$ is the estimator of such a difference.

## A.1. Discussion on the Parameter Space

When assuming the statistical heterogeneity of each client is different ($R$ is related to the client index $i$), it would provide a more delicate description of statistical heterogeneity, and it also requires different personalization degree per client. Therefore, let's consider the following problem with different heterogeneity in personalized federated learning

$$\min_{\boldsymbol{w}^{(g)}, \{\boldsymbol{w}^{(i)}\}_{i \in [m]}} \sum_{i=1}^m p_i \left( L_i(\boldsymbol{w}^{(i)}, S_i) + \frac{\lambda_i}{2} \|\boldsymbol{w}^{(g)} - \boldsymbol{w}^{(i)}\|^2 \right), \tag{19}$$

where the $i$-th client will be shrunk to the global with strength $\lambda_i$. Therefore, solving (19) will consider different personalization degree for different clients. Establish the minimax statistical error bound for Problem (19) would be open for future work.

Under Assumptions 1 and 2 as we assume, statistical heterogeneity in parameter space (5) would be equivalent to many other similar assumptions for statistical heterogeneity, like $B$-dissimilarity (Li et al., 2020), parameter difference (Chen et al., 2023c) and gradient diversity (T Dinh et al., 2020; Deng et al., 2020).

## A.2. Useful Lemmas

To facilitate our analysis, we first present two important lemmas and provide their proof.

**Lemma 3.** *Under Assumption 2, for the optimal solution $\tilde{\boldsymbol{w}}^{(g)}$ and $\tilde{\boldsymbol{w}}^{(i)}$'s in Problem (2), we have*

$$\|\tilde{\boldsymbol{w}}^{(g)} - \tilde{\boldsymbol{w}}^{(i)}\|_2 \leq \frac{2\|\nabla L_i(\tilde{\boldsymbol{w}}^{(g)}, S_i)\|_2}{\mu + \lambda}.$$

*Proof.* By the $\mu$-strongly convexity of $\nabla L_i$, we have that for any $\boldsymbol{w} \in \mathbb{R}^d$,

$$L_i(\boldsymbol{w}, S_i) + \frac{\lambda}{2}\|\boldsymbol{w} - \tilde{\boldsymbol{w}}^{(g)}\|^2 \geq L_i(\tilde{\boldsymbol{w}}^{(g)}, S_i) + \left\langle \nabla L_i(\tilde{\boldsymbol{w}}^{(g)}, S_i), \boldsymbol{w} - \tilde{\boldsymbol{w}}^{(g)} \right\rangle + \frac{\mu + \lambda}{2}\|\boldsymbol{w} - \tilde{\boldsymbol{w}}^{(g)}\|^2$$

$$\geq L_i(\tilde{\boldsymbol{w}}^{(g)}, S_i) + \|\boldsymbol{w} - \tilde{\boldsymbol{w}}^{(g)}\| \left( \frac{\mu + \lambda}{2}\|\boldsymbol{w} - \tilde{\boldsymbol{w}}^{(g)}\| - \|\nabla L_i(\tilde{\boldsymbol{w}}^{(g)}, S_i)\| \right), \tag{20}$$

where in the last step, we used the Cauchy inequality.

In addition, notice that $\tilde{\boldsymbol{w}}^{(i)}$ is the optimal solution for the local model $\boldsymbol{w}^{(i)}$ in Problem (2), thus we have

$$\tilde{\boldsymbol{w}}^{(i)} = \underset{\boldsymbol{w}}{\operatorname{argmin}} \left\{ L_i(\boldsymbol{w}, S_i) + \frac{\lambda}{2} \|\boldsymbol{w} - \tilde{\boldsymbol{w}}^{(g)}\|^2 \right\}.$$

This result together with (20) implies that

$$L_i(\tilde{\boldsymbol{w}}^{(g)}, S_i) = L_i(\tilde{\boldsymbol{w}}^{(g)}, S_i) + \frac{\lambda}{2} \|\tilde{\boldsymbol{w}}^{(g)} - \tilde{\boldsymbol{w}}^{(g)}\|^2 \geq L_i(\tilde{\boldsymbol{w}}^{(i)}, S_i) + \frac{\lambda}{2} \|\tilde{\boldsymbol{w}}^{(i)} - \tilde{\boldsymbol{w}}^{(g)}\|^2$$

$$\geq L_i(\tilde{\boldsymbol{w}}^{(g)}, S_i) + \|\tilde{\boldsymbol{w}}^{(i)} - \tilde{\boldsymbol{w}}^{(g)}\| \left( \frac{\mu + \lambda}{2} \|\tilde{\boldsymbol{w}}^{(i)} - \tilde{\boldsymbol{w}}^{(g)}\| - \|\nabla L_i(\tilde{\boldsymbol{w}}^{(g)}, S_i)\| \right).$$

Therefore, since $\mu + \lambda > 0$ and $\|\tilde{\boldsymbol{w}}^{(i)} - \tilde{\boldsymbol{w}}^{(g)}\| \geq 0$, rearranging the terms leads to

$$\|\tilde{\boldsymbol{w}}^{(g)} - \tilde{\boldsymbol{w}}^{(i)}\| \leq \frac{2\|\nabla L_i(\tilde{\boldsymbol{w}}^{(g)}, S_i)\|}{\mu + \lambda}, \tag{21}$$

as claimed. $\qquad\square$

**Remark**: Lemma 3 is crucial to bridge the solution of the global model with the local model in Problem 2. Note that the difference is bounded by the gradient norm of local loss functions evaluated at the optimal solution of the global model, which would be decomposed further and controlled by the global model's statistical accuracy, as we will show later.

**Lemma 4.** *Under Assumption 1, 2, 3, for $\tilde{\boldsymbol{w}}_{GT}$, the solution of the GlobalTrain problem in (4), we have*

$$\mathbb{E}\|\tilde{\boldsymbol{w}}_{GT} - \boldsymbol{w}_\star^{(g)}\| \leq \frac{\rho\sqrt{\sum_{i \in [m]} \frac{p_i^2}{n_i}} + LR}{\mu/2}.$$

*Proof.* By the optimality condition of the GlobalTrain problem in (4) and the strong convexity of $L_i$'s, we have

$$0 \geq \sum_{i \in [m]} p_i \left( L_i(\tilde{\boldsymbol{w}}_{GT}, S_i) - L_i(\boldsymbol{w}_\star^{(g)}, S_i) \right)$$

$$\geq \sum_{i \in [m]} p_i \left\langle \nabla L_i(\boldsymbol{w}_\star^{(g)}, S_i), \tilde{\boldsymbol{w}}_{GT} - \boldsymbol{w}_\star^{(g)} \right\rangle + \frac{\mu}{2} \|\tilde{\boldsymbol{w}}_{GT} - \boldsymbol{w}_\star^{(g)}\|^2$$

$$\geq -\left\| \sum_{i \in [m]} p_i \nabla L_i(\boldsymbol{w}_\star^{(g)}, S_i) \right\| \left\| \tilde{\boldsymbol{w}}_{GT} - \boldsymbol{w}_\star^{(g)} \right\| + \frac{\mu}{2} \|\tilde{\boldsymbol{w}}_{GT} - \boldsymbol{w}_\star^{(g)}\|^2.$$

Therefore, we have

$$\|\tilde{\boldsymbol{w}}_{GT} - \boldsymbol{w}_\star^{(g)}\| \leq \frac{\left\| \sum_{i \in [m]} p_i \nabla L_i(\boldsymbol{w}_\star^{(g)}, S_i) \right\|}{\mu/2}$$

$$\leq \frac{\left\| \sum_{i \in [m]} p_i \nabla L_i(\boldsymbol{w}_\star^{(i)}, S_i) \right\| + \sum_{i \in [m]} p_i \left\| \nabla L_i(\boldsymbol{w}_\star^{(g)}, S_i) - \nabla L_i(\boldsymbol{w}_\star^{(i)}, S_i) \right\|}{\mu/2}. \tag{22}$$

By the $L$-smoothness property of $L_i$'s in Assumption 1 and the bounded gradient property listed in Assumption 3, we have

$$\sum_{i \in [m]} p_i \left\| \nabla L_i(\boldsymbol{w}_\star^{(g)}, S_i) - \nabla L_i(\boldsymbol{w}_\star^{(i)}, S_i) \right\| \leq \sum_{i \in [m]} p_i L \|\boldsymbol{w}_\star^{(g)} - \boldsymbol{w}_\star^{(i)}\| \leq LR,$$

$$\mathbb{E}\left\| \sum_{i \in [m]} p_i \nabla L_i(\boldsymbol{w}_\star^{(i)}, S_i) \right\| = \mathbb{E}\left\| \sum_{i \in [m]} \sum_{j \in [n_i]} \frac{p_i}{n_i} \nabla \ell(\boldsymbol{w}_\star^{(i)}, z_{ij}) \right\| \leq \rho \sqrt{\sum_{i \in [m]} \frac{p_i^2}{n_i}}.$$

These results together with (22) implies

$$\mathbb{E}\|\tilde{\boldsymbol{w}}_{\mathrm{GT}} - \boldsymbol{w}_\star^{(g)}\| \leq \frac{\rho\sqrt{\sum_{i\in[m]}\frac{p_i^2}{n_i}} + LR}{\mu/2}$$

as claimed. □

**Remark**: Lemma 4 implies that if we solve the *GlobalTrain* problem exactly, the solution will be close to the ground truth of the global model with a certain error, which is independent of $\lambda$. After presenting the two useful lemmas in Appendix A.2, we will start to prove the theoretical results in Section 4.1. In Section A.3, we will prove Theorem 1 first, then we will prove the statistical error bound in Problem 2 as demonstrated in Corollary 1.

### A.3. Proof of Statistical Convergence in Theorem 1

In the discuss below, we always assume $\boldsymbol{w}_\star^{(i)}$ discussed below comes from the parameter space (5). As the local model is shrunk towards the global model under the influence of personalization degree, how fast the global model $\tilde{\boldsymbol{w}}^{(g)}$ convergence to $\boldsymbol{w}_\star^{(g)}$ can also influence the convergence rate of the local model. In Appendix A.3.1, we will establish the statistical error bound of the global model first. Then in Appendix A.3.2, we will provide the local model error bound, where we explicitly show that the statistical accuracy of the local model depends on the statistical accuracy of the global model. Finally, in Appendix A.3.3, we combine the results of local models and the global model to establish the one-line rate for the local model explicitly stated in Theorem 1.

#### A.3.1. GLOBAL STATISTICAL ERROR BOUND

To analyze the statistical error bound, we consider two separate cases: $R > \sqrt{\sum_i \frac{p_i}{n_i}}$ and $R \leq \sqrt{\sum_i \frac{p_i}{n_i}}$. The motivation for this distinction lies in whether collaboration among different clients is beneficial.

**Case 1**: We first consider the case when $R > \sqrt{\sum_i \frac{p_i}{n_i}}$.

Recall that $\tilde{\boldsymbol{w}}^{(g)}$ and $\{\tilde{\boldsymbol{w}}^{(i)}\}_{i\in[m]}$ are the minimizers of Problem (2). According to the first-order condition, we have

$$\nabla_{\boldsymbol{w}^{(i)}} L_i\left(\boldsymbol{w}^{(i)}, S_i\right)\big|_{\boldsymbol{w}^{(i)}=\tilde{\boldsymbol{w}}^{(i)}} = \lambda\left(\tilde{\boldsymbol{w}}^{(g)} - \tilde{\boldsymbol{w}}^{(i)}\right), \tag{23}$$

$$\tilde{\boldsymbol{w}}^{(g)} = \sum_i p_i \tilde{\boldsymbol{w}}^{(i)}. \tag{24}$$

We start with the optimality condition of $\tilde{\boldsymbol{w}}^{(i)}$ and $\tilde{\boldsymbol{w}}^{(g)}$, which yields

$$0 \geq \sum_{i\in[m]} p_i\left[L_i(\tilde{\boldsymbol{w}}^{(i)}, S_i) + \frac{\lambda}{2}\|\tilde{\boldsymbol{w}}^{(g)} - \tilde{\boldsymbol{w}}^{(i)}\|^2\right] - \sum_{i\in[m]} p_i\left[L_i(\boldsymbol{w}_\star^{(i)}, S_i) + \frac{\lambda}{2}\|\boldsymbol{w}_\star^{(g)} - \boldsymbol{w}_\star^{(i)}\|^2\right]. \tag{25}$$

Reorganizing the terms, we obtain

$$0 \geq \sum_{i\in[m]}\sum_{j\in[n_i]} \frac{p_i}{n_i}\left(\ell(\tilde{\boldsymbol{w}}^{(i)}, z_{ij}) - \ell(\boldsymbol{w}_\star^{(i)}, z_{ij})\right) + \sum_{i\in[m]} p_i\frac{\lambda}{2}\|\tilde{\boldsymbol{w}}^{(g)} - \tilde{\boldsymbol{w}}^{(i)}\|^2 - \sum_{i\in[m]} p_i\frac{\lambda}{2}\|\boldsymbol{w}_\star^{(g)} - \boldsymbol{w}_\star^{(i)}\|^2. \tag{26}$$

For the first term on the R.H.S. in (26), apply the $\mu$-strongly convexity of the loss function $\ell$ (cf. Assumption 2), we then have

$$\sum_{i\in[m]}\sum_{j\in[n_i]} \frac{p_i}{n_i}\left(\ell(\tilde{\boldsymbol{w}}^{(i)}, z_{ij}) - \ell(\boldsymbol{w}_\star^{(i)}, z_{ij})\right) \geq -\sum_{i\in[m]}\sum_{j\in[n_i]} \frac{p_i}{n_i}\left\langle\nabla\ell(\boldsymbol{w}_\star^{(i)}, z_{ij}), \tilde{\Delta}_{\mathrm{stat}}^{(i)}\right\rangle$$
$$+ \frac{\mu}{2}\sum_{i\in[m]}\sum_{j\in[n_i]} \frac{p_i}{n_i}\left\|\tilde{\Delta}_{\mathrm{stat}}^{(i)}\right\|^2, \tag{27}$$

where we use the definition of $\tilde{\Delta}_{\text{stat}}^{(i)}$ in (17).

Plugging (27) back into (26), it yields

$$
\begin{aligned}
0 \geq &- \sum_i \sum_j \frac{p_i}{n_i} \left\langle \nabla \ell(\boldsymbol{w}_\star^{(i)}, z_{ij}), \tilde{\Delta}_{\text{stat}}^{(i)} \right\rangle + \frac{\mu}{2} \sum_i \sum_j \frac{p_i}{n_i} \left\| \tilde{\Delta}_{\text{stat}}^{(i)} \right\|^2 \\
&+ \sum_{i \in [m]} p_i \frac{\lambda}{2} \| \tilde{\boldsymbol{w}}^{(g)} - \tilde{\boldsymbol{w}}^{(i)} \|^2 - \sum_{i \in [m]} p_i \frac{\lambda}{2} \| \boldsymbol{w}_\star^{(g)} - \boldsymbol{w}_\star^{(i)} \|^2.
\end{aligned}
\tag{28}
$$

Recall the definition in (18) and the assumption about statistical heterogeneity in (5)

$$
0 \geq - \sum_i \sum_j \frac{p_i}{n_i} \left\langle \nabla \ell(\boldsymbol{w}_\star^{(i)}, z_{ij}), \tilde{\Delta}_{\text{stat}}^{(i)} \right\rangle + \frac{\mu}{2} \sum_i \sum_j \frac{p_i}{n_i} \left\| \tilde{\Delta}_{\text{stat}}^{(i)} \right\|^2 + \frac{\lambda}{2} \sum_i p_i \left\| \tilde{\boldsymbol{\delta}}^{(i)} \right\|^2 - \frac{\lambda}{2} R^2.
\tag{29}
$$

Applying Cauchy inequality on the first term of (29), it yields

$$
\begin{aligned}
\sum_i \sum_j \frac{p_i}{n_i} \left\langle \nabla \ell(\boldsymbol{w}_\star^{(i)}, z_{ij}), \tilde{\Delta}_{\text{stat}}^{(i)} \right\rangle &= \sum_i p_i \left\langle \sum_j \frac{1}{n_i} \nabla \ell(\boldsymbol{w}_\star^{(i)}, z_{ij}), \tilde{\Delta}_{\text{stat}}^{(i)} \right\rangle \\
&\leq \sum_i p_i \left\| \sum_j \frac{1}{n_i} \nabla \ell(\boldsymbol{w}_\star^{(i)}, z_{ij}) \right\|_2 \left\| \tilde{\Delta}_{\text{stat}}^{(i)} \right\|_2.
\end{aligned}
\tag{30}
$$

Consider the Assumption 3 and $z_{ij}$ are i.i.d data

$$
\left( \mathbb{E} \left\| \sum_j \frac{1}{n_i} \nabla \ell(\boldsymbol{w}_\star^{(i)}, z_{ij}) \right\|_2 \right)^2 \leq \mathbb{E} \left\| \sum_j \frac{1}{n_i} \nabla \ell(\boldsymbol{w}_\star^{(i)}, z_{ij}) \right\|^2 \leq \frac{\rho^2}{n_i}.
\tag{31}
$$

Combining (30) and (31), it yields

$$
\sum_i \sum_j \frac{p_i}{n_i} \mathbb{E} \left\langle \nabla \ell(\boldsymbol{w}_\star^{(i)}, z_{ij}), \tilde{\Delta}_{\text{stat}}^{(i)} \right\rangle \leq \sum_i \rho \frac{p_i}{\sqrt{n_i}} \mathbb{E} \| \tilde{\Delta}_{\text{stat}}^{(i)} \|_2.
\tag{32}
$$

Plugging (32) back to (29) yields

$$
0 \geq -\rho \sum_i \frac{p_i}{\sqrt{n_i}} \mathbb{E} \left\| \tilde{\Delta}_{\text{stat}}^{(i)} \right\|_2 + \frac{\mu}{2} \sum_i p_i \mathbb{E} \left\| \tilde{\Delta}_{\text{stat}}^{(i)} \right\|^2 + \frac{\lambda}{2} \sum_i p_i \mathbb{E} \left\| \tilde{\boldsymbol{\delta}}^{(i)} \right\|^2 - \frac{\lambda}{2} R^2.
\tag{33}
$$

After dropping the third term and applying Cauchy inequality, we obtain

$$
0 \geq -\rho \left( \left( \sum_{i \in [m]} \frac{p_i}{n_i} \right) \left( \sum_i p_i \mathbb{E} \left\| \tilde{\Delta}_{\text{stat}}^{(i)} \right\|^2 \right) \right)^{\frac{1}{2}} + \frac{\mu}{2} \sum_i p_i \mathbb{E} \left\| \tilde{\Delta}_{\text{stat}}^{(i)} \right\|^2 - \frac{\lambda}{2} R^2.
\tag{34}
$$

If we assume $p_i = \frac{1}{m}$ and $n_i = n_j \forall i \neq j$ (assumptions used in Theorem 1), we have

$$
\mathbb{E} \left\| \tilde{\Delta}_{\text{stat}}^{(g)} \right\|^2 \leq \frac{1}{u^2} \left( \frac{4\rho^2}{n} + 2\mu\lambda R^2 \right).
\tag{35}
$$

**Case 2:** Then we consider $R \leq \sqrt{\sum_i \frac{p_i}{n_i}}$. Recall the definition of the global minimizer of GlobalTraining $\tilde{w}_{\mathrm{GT}} = \arg\min_w \sum_i p_i L_i(w; S_i)$, based on the optimality condition of $\tilde{w}_i, \tilde{w}_g, i \in [m]$, we obtain

$$\sum_i p_i \left( L_i(\tilde{\boldsymbol{w}}^{(i)}, S_i) + \frac{\lambda}{2}\|\tilde{\boldsymbol{w}}^{(i)} - \tilde{\boldsymbol{w}}^{(g)}\|^2 \right) \leq \sum_i p_i \left( L_i(\tilde{\boldsymbol{w}}_{\mathrm{GT}}, S_i) + \frac{\lambda}{2}\| \tilde{\boldsymbol{w}}_{\mathrm{GT}} - \tilde{\boldsymbol{w}}_{\mathrm{GT}}\|^2 \right)$$
$$= \sum_i p_i L_i(\tilde{\boldsymbol{w}}_{\mathrm{GT}}, S_i). \tag{36}$$

Therefore, we obtain

$$\sum_i p_i L_i(\tilde{\boldsymbol{w}}_{\mathrm{GT}}, S_i) \geq \sum_i p_i L_i(\tilde{\boldsymbol{w}}^{(i)}, S_i). \tag{37}$$

In addition, applying the smoothness assumption in Assumption 1 yields

$$\left| L_i(\tilde{\boldsymbol{w}}^{(i)}, S_i) - L_i(\tilde{\boldsymbol{w}}^{(g)}, S_i) - \nabla L_i(\tilde{\boldsymbol{w}}^{(g)}, S_i)^T(\tilde{\boldsymbol{w}}^{(i)} - \tilde{\boldsymbol{w}}^{(g)}) \right| \leq \frac{L}{2}\|\tilde{\boldsymbol{w}}^{(i)} - \tilde{\boldsymbol{w}}^{(g)}\|^2. \tag{38}$$

Combining (37) and (38), we have

$$\sum_i p_i L_i(\tilde{\boldsymbol{w}}_{\mathrm{GT}}, S_i) \geq \sum_i p_i L_i(\tilde{\boldsymbol{w}}^{(i)}, S_i) \tag{39}$$

$$\geq \sum_i p_i L_i(\tilde{\boldsymbol{w}}^{(g)}, S_i) + \sum_i p_i \nabla L_i(\tilde{\boldsymbol{w}}^{(g)}, S_i)^\top(\tilde{\boldsymbol{w}}^{(i)} - \tilde{\boldsymbol{w}}^{(g)}) - \sum_i p_i \frac{L}{2}\|\tilde{\boldsymbol{w}}^{(i)} - \tilde{\boldsymbol{w}}^{(g)}|^2. \tag{40}$$

Using Cauchy inequality on the second term of the R.H.S.,

$$\sum_i p_i L_i(\tilde{\boldsymbol{w}}_{\mathrm{GT}}, S_i) \geq \sum_i p_i L_i(\tilde{\boldsymbol{w}}^{(g)}, S_i) - \sum_i p_i \|\nabla L_i(\tilde{\boldsymbol{w}}^{(g)}, S_i)\|_2 \|\tilde{\boldsymbol{w}}^{(i)} - \tilde{\boldsymbol{w}}^{(g)}\|_2$$
$$- \sum_i p_i \frac{L}{2}\|\tilde{\boldsymbol{w}}^{(i)} - \tilde{\boldsymbol{w}}^{(g)}\|^2. \tag{41}$$

On the other hand, note that the optimality condition of $\tilde{\boldsymbol{w}}_{\mathrm{GT}}$ is

$$\sum_i p_i \nabla L_i(\tilde{\boldsymbol{w}}_{\mathrm{GT}}, S_i) = 0. \tag{42}$$

Using the strong-convexity assumption in Assumption 2, we can show that

$$\sum_i p_i \left\{ L_i(\tilde{\boldsymbol{w}}^{(g)}, S_i) - L_i(\tilde{\boldsymbol{w}}_{\mathrm{GT}}, S_i) - \nabla L_i(\tilde{\boldsymbol{w}}_{\mathrm{GT}}, S_i)^T(\tilde{\boldsymbol{w}}^{(g)} - \tilde{\boldsymbol{w}}_{\mathrm{GT}}) \right\}$$
$$= \sum_i p_i \left\{ L_i(\tilde{\boldsymbol{w}}^{(g)}, S_i) - L_i(\tilde{\boldsymbol{w}}_{\mathrm{GT}}, S_i) \right\}$$
$$\geq \sum_i p_i \frac{\mu}{2}\|\tilde{\boldsymbol{w}}^{(g)} - \tilde{\boldsymbol{w}}_{\mathrm{GT}}\|^2$$
$$= \frac{\mu}{2}\|\tilde{\boldsymbol{w}}^{(g)} - \tilde{\boldsymbol{w}}_{\mathrm{GT}}\|^2. \tag{43}$$

Combining the results (41) and (43), it yields

$$
\sum_i p_i L_i(\tilde{\boldsymbol{w}}_{\mathrm{GT}}, S_i) \geq \sum_i p_i L_i(\tilde{\boldsymbol{w}}_{\mathrm{GT}}, S_i) + \frac{\mu}{2}\|\tilde{\boldsymbol{w}}^{(g)} - \tilde{\boldsymbol{w}}_{\mathrm{GT}}\|^2
$$
$$
- \sum_i p_i \|\nabla L_i(\tilde{\boldsymbol{w}}^{(g)}, S_i)\|_2 \|\tilde{\boldsymbol{w}}^{(i)} - \tilde{\boldsymbol{w}}^{(g)}\|_2 - \sum_i p_i \frac{L}{2}\|\tilde{\boldsymbol{w}}^{(i)} - \tilde{\boldsymbol{w}}^{(g)}\|^2. \tag{44}
$$

Reorganizing these terms yields

$$
\frac{\mu}{2}\|\tilde{\boldsymbol{w}}^{(g)} - \tilde{\boldsymbol{w}}_{\mathrm{GT}}\|^2 \leq \sum_i p_i \left\{ \|\nabla L_i(\tilde{\boldsymbol{w}}^{(g)}, S_i)\|_2 \|\tilde{\boldsymbol{w}}^{(i)} - \tilde{\boldsymbol{w}}^{(g)}\|_2 + \frac{L}{2}\|\tilde{\boldsymbol{w}}^{(i)} - \tilde{\boldsymbol{w}}^{(g)}\|^2 \right\}. \tag{45}
$$

Using Lemma 3 to bound the term $\|\tilde{\boldsymbol{w}}^{(i)} - \tilde{\boldsymbol{w}}^{(g)}\|^2$, we have

$$
\|\tilde{\boldsymbol{w}}^{(g)} - \tilde{\boldsymbol{w}}_{\mathrm{GT}}\|^2 \leq \frac{2}{\mu}\sum_i p_i \left[ \frac{2}{\mu+\lambda} + \frac{2L}{(\mu+\lambda)^2} \right] \|\nabla L_i(\tilde{\boldsymbol{w}}^{(g)}, S_i)\|^2
$$
$$
= \frac{4}{\mu}\left( \frac{1}{\mu+\lambda} + \frac{L}{(\mu+\lambda)^2} \right) \sum_i p_i \|\nabla L_i(\tilde{\boldsymbol{w}}^{(g)}, S_i)\|^2. \tag{46}
$$

Note that by the smoothness Assumption and triangle inequality,

$$
\|\nabla L_i(\tilde{\boldsymbol{w}}^{(g)}, S_i)\|_2^2 \leq 2\|\nabla L_i(\tilde{\boldsymbol{w}}^{(g)}, S_i) - \nabla L_i(\boldsymbol{w}_\star^{(i)}, S_i)\|^2 + 2\|\nabla L_i(\boldsymbol{w}_\star^{(i)}, S_i)\|^2
$$
$$
\leq 2L^2\|\tilde{\boldsymbol{w}}^{(g)} - \boldsymbol{w}_\star^{(i)}\|^2 + 2\|\nabla L_i(\boldsymbol{w}_\star^{(i)}, S_i)\|^2. \tag{47}
$$

Take expectation w.r.t all data and use Assumption 3 in the parameter space (5), then we have

$$
\mathbb{E}[\|\nabla L_i(\tilde{\boldsymbol{w}}^{(g)}, S_i)\|^2] \leq 2L^2\mathbb{E}[\|\tilde{\boldsymbol{w}}^{(g)} - \boldsymbol{w}_\star^{(i)}\|^2] + 2\mathbb{E}\left[ \left\| \frac{1}{n_i}\sum_{j=1}^{n_i} \nabla L(\boldsymbol{w}_\star^{(i)}, z_{ij}) \right\|^2 \right]
$$
$$
\leq 4L^2\mathbb{E}[\|\tilde{\boldsymbol{w}}^{(g)} - \boldsymbol{w}_\star^{(g)}\|^2] + 4L^2 R^2 + 2\frac{\rho^2}{n}, \tag{48}
$$

where the last term comes from (31).

Plugging (48) back into (46), it yields

$$
\mathbb{E}[\|\tilde{\boldsymbol{w}}^{(g)} - \tilde{\boldsymbol{w}}_{\mathrm{GT}}\|^2] \leq \frac{4}{\mu}\left( \frac{1}{\mu+\lambda} + \frac{L}{(\mu+\lambda)^2} \right) \left\{ 4L^2\mathbb{E}[\|\tilde{\boldsymbol{w}}^{(g)} - \boldsymbol{w}_\star^{(g)}\|^2] + 4L^2 R^2 + 2\frac{\rho^2}{n} \right\}. \tag{49}
$$

Next, we are ready to establish the global error bound. Using the result (49) and Lemma 4, we obtain

$$
\mathbb{E}[\|\tilde{\boldsymbol{w}}^{(g)} - \boldsymbol{w}_\star^{(g)}\|^2] \leq \frac{4}{\mu}\left( \frac{1}{\mu+\lambda} + \frac{L}{(\mu+\lambda)^2} \right) \left\{ 4L^2\mathbb{E}[\|\tilde{\boldsymbol{w}}^{(g)} - \boldsymbol{w}_\star^{(g)}\|^2] + 4L^2 R^2 + 2\frac{\rho^2}{n} \right\}
$$
$$
+ \frac{8\rho^2}{\mu^2}\frac{1}{N} + \frac{8L^2}{\mu^2}R^2. \tag{50}
$$

Let's define $g(\lambda) = \frac{4}{\mu} \left( \frac{1}{\mu+\lambda} + \frac{L}{(\mu+\lambda)^2} \right)$ and reorganize (50), which yields

$$\mathbb{E}[\|\tilde{\boldsymbol{w}}^{(g)} - \boldsymbol{w}_\star^{(g)}\|^2] \leq \frac{g(\lambda)\left(4L^2R^2 + 2\frac{\rho^2}{n}\right) + \frac{8\rho^2}{\mu^2}\frac{1}{N} + \frac{8L^2}{\mu^2}R^2}{1 - 4L^2g(\lambda)}. \tag{51}$$

If we assume $p_i = \frac{1}{m}$, $n_i = n_j \ \forall i \neq j$ (assumptions used in Theorem 1) and $g(\lambda) \leq 1/(8L)^2$, we have

$$\mathbb{E}[\|\tilde{\Delta}_{\text{stat}}^{(g)}\|^2] \leq \frac{g(\lambda)}{1 - 4L^2g(\lambda)} \left(4L^2R^2 + 2\frac{\rho^2}{n}\right) + \frac{16\rho^2}{\mu^2}\frac{1}{N} + \frac{16L^2}{\mu^2}R^2. \tag{52}$$

### A.3.2. LOCAL STATISTICAL ERROR BOUND

Analogous to the analysis of the global model, we examine the statistical accuracy of the local model in a similar manner. For simplicity, in the following arguments, we denote the upper bound of $\mathbb{E}\|\tilde{\Delta}_{\text{stat}}^{(g)}\|_2$ as $U_0$. The expliciate expression would be found in results (35) for $R > \sqrt{\sum_i \frac{p_i}{n_i}}$ and (52) for $R \leq \sqrt{\sum_i \frac{p_i}{n_i}}$.

**Case 1**: first we consider the case when $R > \sqrt{\sum_i \frac{p_i}{n_i}}$.

To prove the local statistical error bound, we start with the optimality condition of $\tilde{\boldsymbol{w}}^{(i)}$ and $\tilde{\boldsymbol{w}}^{(g)}$ for a single client, which yields that for $i \in [m]$,

$$0 \geq L_i\left(\tilde{\boldsymbol{w}}^{(i)}, S_i\right) + \frac{\lambda}{2}\left\|\tilde{\boldsymbol{w}}^{(g)} - \tilde{\boldsymbol{w}}^{(i)}\right\|^2 - L_i\left(\boldsymbol{w}_\star^{(i)}, S_i\right) - \frac{\lambda}{2}\left\|\tilde{\boldsymbol{w}}^{(g)} - \boldsymbol{w}_\star^{(i)}\right\|^2. \tag{53}$$

Reorganized the terms, we obtain

$$0 \geq \sum_{j \in [n_i]} \frac{1}{n_i} \left( \ell\left(\tilde{\boldsymbol{w}}^{(i)}, z_{ij}\right) - \ell\left(\boldsymbol{w}_\star^{(i)}, z_{ij}\right) \right) + \frac{\lambda}{2}\left\|\tilde{\boldsymbol{w}}^{(g)} - \tilde{\boldsymbol{w}}^{(i)}\right\|^2 - \frac{\lambda}{2}\left\|\tilde{\boldsymbol{w}}^{(g)} - \boldsymbol{w}_\star^{(i)}\right\|^2. \tag{54}$$

For the first term on the R.H.S., applying the $\mu$-strongly convex of the loss function $\ell$ (cf. Assumption 2) yields

$$\sum_{j \in [n_i]} \frac{1}{n_i} \left( \ell(\tilde{\boldsymbol{w}}^{(i)}, z_{ij}) - \ell(\boldsymbol{w}_\star^{(i)}, z_{ij}) \right) \geq -\sum_{j \in [n_i]} \frac{1}{n_i}\left\langle \nabla\ell(\boldsymbol{w}_\star^{(i)}, z_{ij}), \tilde{\Delta}_{\text{stat}}^{(i)} \right\rangle + \frac{\mu}{2}\sum_{j \in [n_i]} \frac{1}{n_i}\left\|\tilde{\Delta}_{\text{stat}}^{(i)}\right\|^2, \tag{55}$$

where we use the definition of $\tilde{\Delta}_{\text{stat}}^{(i)}$ in (17).

Plugging (55) back into (54), it yields

$$0 \geq -\sum_{j} \frac{1}{n_i}\left\langle \nabla\ell(\boldsymbol{w}_\star^{(i)}, z_{ij}), \tilde{\Delta}_{\text{stat}}^{(i)} \right\rangle + \frac{\mu}{2}\sum_{j} \frac{1}{n_i}\left\|\tilde{\Delta}_{\text{stat}}^{(i)}\right\|^2 + \frac{\lambda}{2}\|\tilde{\boldsymbol{w}}^{(g)} - \tilde{\boldsymbol{w}}^{(i)}\|^2 - \frac{\lambda}{2}\|\tilde{\boldsymbol{w}}^{(g)} - \boldsymbol{w}_\star^{(i)}\|^2. \tag{56}$$

Apply Cauchy inequality on the first term of (56)

$$\sum_{j} \frac{1}{n_i}\left\langle \nabla\ell(\boldsymbol{w}_\star^{(i)}, z_{ij}), \tilde{\Delta}_{\text{stat}}^{(i)} \right\rangle = \left\langle \sum_{j} \frac{1}{n_i}\nabla\ell(\boldsymbol{w}_\star^{(i)}, z_{ij}), \tilde{\Delta}_{\text{stat}}^{(i)} \right\rangle$$
$$\leq \left\| \sum_{j} \frac{1}{n_i}\nabla\ell(\boldsymbol{w}_\star^{(i)}, z_{ij}) \right\|_2 \left\| \tilde{\Delta}_{\text{stat}}^{(i)} \right\|_2. \tag{57}$$

Consider the Assumption 3 and $z_{ij}$ are i.i.d data

$$\left( \mathbb{E} \left\| \sum_j \frac{1}{n_i} \nabla \ell(\boldsymbol{w}_\star^{(i)}, z_{ij}) \right\|_2 \right)^2 \leq \mathbb{E} \left\| \sum_j \frac{1}{n_i} \nabla \ell(\boldsymbol{w}_\star^{(i)}, z_{ij}) \right\|^2 \leq \frac{\rho^2}{n_i}. \tag{58}$$

Combining (57) and (58), it yields

$$\sum_j \frac{1}{n_i} \mathbb{E} \left\langle \nabla \ell(\boldsymbol{w}_\star^{(i)}, z_{ij}), \tilde{\Delta}_{\text{stat}}^{(i)} \right\rangle \leq \rho \frac{1}{\sqrt{n_i}} \mathbb{E} \| \tilde{\Delta}_{\text{stat}}^{(i)} \|_2. \tag{59}$$

Plugging (59) back to (56) yields

$$0 \geq -\rho \frac{1}{\sqrt{n_i}} \mathbb{E} \left\| \tilde{\Delta}_{\text{stat}}^{(i)} \right\|_2 + \frac{\mu}{2} \mathbb{E} \left\| \tilde{\Delta}_{\text{stat}}^{(i)} \right\|^2 + \frac{\lambda}{2} \mathbb{E} \| \tilde{\boldsymbol{w}}^{(g)} - \tilde{\boldsymbol{w}}^{(i)} \|^2 - \frac{\lambda}{2} \mathbb{E} \| \tilde{\boldsymbol{w}}^{(g)} - \boldsymbol{w}_\star^{(i)} \|^2. \tag{60}$$

Note that

$$-\| \tilde{\boldsymbol{w}}^{(g)} - \boldsymbol{w}_\star^{(i)} \|^2 = - \left\| \tilde{\boldsymbol{w}}^{(g)} - \boldsymbol{w}_\star^{(g)} + \boldsymbol{w}_\star^{(g)} - \boldsymbol{w}_\star^{(i)} \right\|^2$$
$$\geq -\| \tilde{\Delta}_{\text{stat}}^{(g)} \|^2 - \| \boldsymbol{w}_\star^{(g)} - \boldsymbol{w}_\star^{(i)} \|^2 - 2 \| \boldsymbol{w}_\star^{(g)} - \boldsymbol{w}_\star^{(i)} \|_2 \left\| \tilde{\Delta}_{\text{stat}}^{(g)} \right\|_2 \tag{61}$$
$$\geq -\| \tilde{\Delta}_{\text{stat}}^{(g)} \|^2 - R^2 - 2R \left\| \tilde{\Delta}_{\text{stat}}^{(g)} \right\|_2.$$

Plug (61) into (60)

$$0 \geq - \frac{\rho}{\sqrt{n_i}} \mathbb{E} \left\| \tilde{\Delta}_{\text{stat}}^{(i)} \right\|_2 + \frac{\mu}{2} \mathbb{E} \left\| \tilde{\Delta}_{\text{stat}}^{(i)} \right\|^2 + \frac{\lambda}{2} \mathbb{E} \left\| \tilde{\boldsymbol{\delta}}^{(i)} \right\|^2 - \frac{\lambda}{2} \mathbb{E} \left\| \tilde{\boldsymbol{w}}^{(g)} - \boldsymbol{w}_\star^{(g)} + \boldsymbol{w}_\star^{(g)} - \boldsymbol{w}_\star^{(i)} \right\|^2$$
$$\geq - \frac{\rho}{\sqrt{n_i}} \mathbb{E} \left\| \tilde{\Delta}_{\text{stat}}^{(i)} \right\|_2 + \frac{\mu}{2} \mathbb{E} \left\| \tilde{\Delta}_{\text{stat}}^{(i)} \right\|^2 + \frac{\lambda}{2} \mathbb{E} \left\| \tilde{\boldsymbol{\delta}}^{(i)} \right\|^2 - \frac{\lambda}{2} \mathbb{E} \left\| \tilde{\Delta}_{\text{stat}}^{(g)} \right\|^2 - \frac{\lambda}{2} R^2 - \lambda R \mathbb{E} \left\| \tilde{\Delta}_{\text{stat}}^{(g)} \right\|_2. \tag{62}$$

Once we denote the upper bound of $\mathbb{E} \left\| \tilde{\Delta}_{\text{stat}}^{(g)} \right\|^2$ as $U_0$, it yields

$$0 \geq - \frac{\rho}{\sqrt{n_i}} \mathbb{E} \left\| \tilde{\Delta}_{\text{stat}}^{(i)} \right\|_2 + \frac{\mu}{2} \mathbb{E} \left\| \tilde{\Delta}_{\text{stat}}^{(i)} \right\|^2 + \frac{\lambda}{2} \mathbb{E} \left\| \tilde{\boldsymbol{\delta}}^{(i)} \right\|^2 - \frac{\lambda}{2} U_0^2 - \frac{\lambda}{2} R^2 - \lambda R \, U_0$$
$$\geq \frac{\mu}{2} \mathbb{E} \left\| \tilde{\Delta}_{\text{stat}}^{(i)} \right\|^2 - \frac{\rho}{\sqrt{n_i}} \mathbb{E} \left\| \tilde{\Delta}_{\text{stat}}^{(i)} \right\|_2 - \frac{\lambda}{2} (U_0 + R)^2. \tag{63}$$

If we assume $p_i = \frac{1}{m}$, $n_i = n_j \, \forall i \neq j$ (assumptions used in Theorem 1), we can obtain

$$\mathbb{E} \left\| \tilde{\Delta}_{\text{stat}}^{(i)} \right\|^2 \leq \frac{1}{u^2} \left( \frac{4\rho^2}{n} + 4u\lambda \left( U_0^2 + R^2 \right) \right). \tag{64}$$

**Case 2:** we then consider the case when $R \leq \sqrt{\sum_i \frac{p_i}{n_i}}$.

If we assume $p_i = \frac{1}{m}$, $n_i = n_j \, \forall i \neq j$ (assumptions used in Theorem 1), we can obtain

$$
\begin{aligned}
&\mathbb{E}[\|\tilde{\boldsymbol{w}}^{(i)} - \boldsymbol{w}_\star^{(i)}\|^2] \\
&\leq 3\mathbb{E}[\|\tilde{\boldsymbol{w}}^{(i)} - \tilde{\boldsymbol{w}}^{(g)}\|^2] + 3\mathbb{E}[\|\tilde{\boldsymbol{w}}^{(g)} - \boldsymbol{w}_\star^{(g)}\|^2] + 3\mathbb{E}[\|\boldsymbol{w}_\star^{(g)} - \boldsymbol{w}_\star^{(i)}\|^2] \\
&\overset{(a)}{\leq} \frac{12}{(\mu+\lambda)^2}\mathbb{E}[\|\nabla L_i(\tilde{\boldsymbol{w}}^{(g)})\|^2] + 3\mathbb{E}[\|\tilde{\boldsymbol{w}}^{(g)} - \boldsymbol{w}_\star^{(g)}\|^2] + 3R^2 \\
&\overset{(b)}{\leq} \frac{12}{(\mu+\lambda)^2}\left[4L^2\mathbb{E}[\|\tilde{\boldsymbol{w}}^{(g)} - \boldsymbol{w}_\star^{(g)}\|^2] + 4L^2R^2 + 2\frac{\rho^2}{n}\right] + 3\mathbb{E}[\|\tilde{\boldsymbol{w}}^{(g)} - \boldsymbol{w}_\star^{(g)}\|^2] + 3R^2,
\end{aligned}
\tag{65}
$$

where step (a) comes from the assumption of parameter space in (5) and Lemma 3 and step (b) comes from result (48).

### A.3.3. PROOF THE STATISTICAL ERROR BOUND IN THEOREM 1

For $R > \sqrt{\sum_i \frac{p_i}{n_i}}$, putting the statistical error of global model (35) into the statistical error of local models (64), we have

$$
\mathbb{E}\left\|\tilde{\Delta}_{\text{stat}}^{(i)}\right\|^2 \leq \left(\frac{4\rho^2}{u^2} + \frac{4\rho^2}{u^3}\lambda\right)\frac{1}{n} + \frac{8}{u^2}\lambda^2 R^2 + \frac{4}{u}\lambda R^2,
\tag{66}
$$

therefore the second part of (8) in Theorem 1 is established. For clarity, we can define

$$
q_2(\lambda) = \frac{8}{u^2}\lambda^2 R^2 + \frac{4}{u}\lambda R^2.
\tag{67}
$$

For $R > \sqrt{\sum_i \frac{p_i}{n_i}}$, plugging the result (52) of the global model into the local model's error bound (65) and reorganizing the terms yields

$$
\mathbb{E}\left[\|\tilde{\Delta}_{\text{stat}}^{(i)}\|^2\right] \leq \frac{12}{(\mu+\lambda)^2}\left[4L^2R^2 + 2\frac{\rho^2}{n}\right] + \left[\frac{48L^2}{(\mu+\lambda)^2} + 3\right]U_0^2 + 3R^2
\tag{68}
$$

$$
\leq \frac{48\rho^2}{\mu^2}\frac{1}{N} + \left[\frac{48L^2}{\mu^2} + 3\right]R^2 + \frac{1}{q_1(\lambda)},
\tag{69}
$$

where

$$
\begin{aligned}
[q_1(\lambda)]^{-1} =& \frac{12}{(\mu+\lambda)^2}\left[4L^2R^2 + 2\frac{\rho^2}{n}\right] + \frac{48L^2}{(\mu+\lambda)^2}\left[\frac{g(\lambda)}{1 - 4L^2 g(\lambda)}\left(4L^2R^2 + 2\frac{\rho^2}{n^2}\right)\right. \\
&\left. + 2\left(\frac{8\rho^2}{\mu^2}\frac{1}{N} + \frac{8L^2}{\mu^2}R^2\right)\right] + \frac{3g(\lambda)}{1 - 4L^2 g(\lambda)}\left(4L^2R^2 + 2\frac{\rho^2}{n^2}\right).
\end{aligned}
\tag{70}
$$

This combining with the definition of $g(\lambda)$ implies that $q_1(\lambda)$ is a monotonically increasing function of $\lambda$ and $\lim_{\lambda \to \infty} q_1(\lambda) = \infty$, as claimed.

## A.4. The Proof of Corollary 1

Analogous to the proof of Theorem 1, we will discuss the two cases of $R^2$ separately as well. For clarity of our analysis, we will assume $p_i = \frac{1}{m}$ and $n_i = n_j \, \forall i \neq j$.

### A.4.1. THE STATISTICAL CONVERGENCE OF THE GLOBAL MODEL WITH A CHOICE OF $\lambda$

**Case 1:** For $R > \frac{1}{\sqrt{n}}$, we have derived the global model error bound in (34) as

$$
0 \geq -\rho\left(\left(\sum_{i \in [m]}\frac{p_i}{n_i}\right)\left(\sum_i p_i \mathbb{E}\left\|\tilde{\Delta}_{\text{stat}}^{(i)}\right\|^2\right)\right)^{\frac{1}{2}} + \frac{\mu}{2}\sum_i p_i \mathbb{E}\left\|\tilde{\Delta}_{\text{stat}}^{(i)}\right\|^2 - \frac{\lambda}{2}R^2.
\tag{71}
$$

If we set the $\lambda$ as

$$\lambda \leq \frac{\rho^2}{\mu n R^2}, \tag{72}$$

then solving the inequality in (35) w.r.t $\sum_i p_i \mathbb{E} \left\| \tilde{\Delta}_{\text{stat}}^{(i)} \right\|^2$ cound yield

$$\mathbb{E} \left\| \tilde{\Delta}_{\text{stat}}^{(g)} \right\| \leq \left( \sum_i p_i \mathbb{E} \left\| \tilde{\Delta}_{\text{stat}}^{(i)} \right\|^2 \right)^{\frac{1}{2}} \leq \frac{(1+\sqrt{3})\rho}{\mu\sqrt{n}}. \tag{73}$$

**Case 2:** For $R \leq \frac{1}{\sqrt{n}}$, we have obtain the results in (52) as

$$\mathbb{E}[\|\tilde{\Delta}_{\text{stat}}^{(g)}\|^2] \leq \frac{g(\lambda)}{1 - 4L^2 g(\lambda)} \left( 4L^2 R^2 + 2\frac{\rho^2}{n} \right) + \frac{16\rho^2}{\mu^2} \frac{1}{N} + \frac{16L^2}{\mu^2} R^2. \tag{74}$$

If we assume

$$g(\lambda) \leq \frac{1}{8L^2} \wedge \left( \left( \frac{4\rho^2}{\mu^2 N} + \frac{4L^2 R^2}{\mu^2} \right) \Big/ \left( 2L^2 R^2 + \frac{\rho^2}{n} \right) \right), \tag{75}$$

we have

$$\mathbb{E}\|\tilde{w}^{(g)} - w_\star^{(g)}\|^2 \leq 32 \left( \frac{\rho^2}{\mu^2} \frac{1}{N} + \frac{L^2}{\mu^2} R^2 \right). \tag{76}$$

Furthermore, we can get a simplified condition for $\lambda$ as

$$\lambda \geq \frac{8\kappa}{\mu} \left\{ 8L^2 \vee \left( \left( 2L^2 R^2 + \frac{\rho^2}{n} \right) \Big/ \left( \frac{4\rho^2}{\mu^2 N} + \frac{4L^2 R^2}{\mu^2} \right) \right) \right\} - \mu, \tag{77}$$

where $\kappa = \frac{L}{\mu} \geq 1$ and the condition that $\lambda$ is non-negative holds trivially as $\frac{\kappa L^2}{\mu} = \kappa^2 L \geq \mu$.

Combine two-case arguments, we have

$$\mathbb{E}\|\tilde{w}^{(g)} - w_\star^{(g)}\|^2 \leq \begin{cases} \left( \frac{(1+\sqrt{3})\rho}{\mu\sqrt{n}} \right)^2 & R \leq \frac{1}{\sqrt{n}} \\ 32 \left( \frac{\rho^2}{\mu^2} \frac{1}{N} + \frac{L^2}{\mu^2} R^2 \right) & R \leq \frac{1}{\sqrt{n}}. \end{cases} \tag{78}$$

Thus it yields a one-line rate

$$\mathbb{E}\|\tilde{w}^{(g)} - w_\star^{(g)}\|^2 \leq C_1 \frac{1}{N} + C_2 \left( \frac{1}{n} \wedge R^2 \right), \tag{79}$$

where

$$\begin{aligned} C_1 &= 32\frac{\rho^2}{\mu^2}, \\ C_2 &= \left( \frac{(1+\sqrt{3})\rho}{\mu} \right)^2 \vee \frac{32L^2}{\mu^2}. \end{aligned} \tag{80}$$

A.4.2. THE STATISTICAL CONVERGENCE OF THE LOCAL MODEL WITH A CHOICE OF $\lambda$

**Case 1:** For $R > \frac{1}{\sqrt{n}}$, we have derived the result in (64) as

$$\mathbb{E}\left\|\tilde{\Delta}_{\text{stat}}^{(i)}\right\|^2 \leq \frac{1}{u^2}\left(\frac{4\rho^2}{n} + 4u\lambda\left(U_0^2 + R^2\right)\right). \tag{81}$$

In addition, recall that, when $R > \frac{1}{\sqrt{n}}$, we set $\lambda \leq \frac{\rho^2}{\mu n R^2}$ in the statistical convergence analysis of the global model. Therefore, putting the global model's error bound (See result in (78)) into (64), which yields

$$\begin{aligned}
\mathbb{E}\|\tilde{\Delta}_{\text{stat}}^{(i)}\|_2 &\leq \frac{1}{\mu}\left[\frac{\rho}{\sqrt{n}} + \left(\frac{\rho^2}{n} + \frac{2\rho^2}{nR^2}\left(U_0 + R\right)^2\right)^{\frac{1}{2}}\right]. \\
&= \frac{1}{\sqrt{n}}\frac{1}{\mu}\left(\rho + \left(\rho^2 + \frac{2\rho^2}{R^2}(U_0 + R)^2\right)^{\frac{1}{2}}\right) \\
&\leq \frac{1}{\sqrt{n}}\frac{\rho + (\rho^2 + 4\rho^2(C_2 + 1))^{\frac{1}{2}}}{\mu} \\
&= \frac{1}{\sqrt{n}}\frac{\rho(1 + (1 + 4(C_2 + 1))^{\frac{1}{2}})}{\mu},
\end{aligned} \tag{82}$$

where $C_2$ has been defined in (80).

**Case 2:** For $R \leq \frac{1}{\sqrt{n}}$, from the result (65), we know

$$\mathbb{E}[\|\tilde{\boldsymbol{w}}^{(i)} - \boldsymbol{w}_\star^{(i)}\|^2] \leq \frac{12}{(\mu + \lambda)^2}\left[4L^2U_0^2 + 4L^2R^2 + 2\frac{\rho^2}{n}\right] + 3U_0^2 + 3R^2. \tag{83}$$

Then putting the global model's results (78) into (65), it yields

$$\mathbb{E}[\|\tilde{\boldsymbol{w}}^{(i)} - \boldsymbol{w}_\star^{(i)}\|^2] \tag{84}$$

$$\leq \frac{12}{(\mu + \lambda)^2}\left[128L^2\left(\frac{\rho^2}{\mu^2}\frac{1}{N} + \frac{L^2}{\mu^2}R^2\right) + 4L^2R^2 + 2\frac{\rho^2}{n}\right] + 96\left(\frac{\rho^2}{\mu^2}\frac{1}{N} + \frac{L^2}{\mu^2}R^2\right) + 3R^2$$

$$= \frac{12}{(\mu + \lambda)^2}\left[\frac{128L^2\rho^2}{\mu^2}\frac{1}{N} + (128L^4/\mu^2 + 4L^2)R^2 + 2\frac{\rho^2}{n}\right] + \frac{96\rho^2}{\mu^2}\frac{1}{N} + (96\frac{L^2}{\mu^2} + 3)R^2. \tag{85}$$

If we assume $\lambda$ satisfies the following condition

$$\frac{12}{(\mu + \lambda)^2} \leq \frac{96\rho^2/(\mu^2 N) + 99\kappa^2 R^2}{128\rho^2/N + 132L^2R^2 + 2\rho^2/n}, \tag{86}$$

then we will have

$$\mathbb{E}[\|\tilde{\boldsymbol{w}}^{(i)} - \boldsymbol{w}_\star^{(i)}\|^2] \leq \frac{192\rho^2}{\mu^2}\frac{1}{N} + 198\kappa^2 R^2. \tag{87}$$

Furthermore, we can get a simplified condition for $\lambda$ as

$$\lambda \geq \sqrt{\frac{8(64\rho^2/N + 66L^2R^2 + \rho^2/n)}{32\rho^2/(\mu^2 N) + 33\kappa^2 R^2}} - \mu. \tag{88}$$

Combining two-case arguments, we have

$$\Rightarrow \mathbb{E}\left\|\tilde{\Delta}_{\text{stat}}^{(i)}\right\|^2 \le \begin{cases} \frac{1}{n}\left(\frac{\rho(1+(1+4(C_2+1))^{\frac{1}{2}})}{\mu}\right)^2 & R > \sqrt{\frac{m}{N}} \\ \frac{192\rho^2}{\mu^2}\frac{1}{N} + 198\kappa^2 R^2 & R \le \sqrt{\frac{m}{N}} \end{cases}. \tag{89}$$

And it yields a one-line rate

$$\mathbb{E}\left\|\tilde{\Delta}_{\text{stat}}^{(i)}\right\|^2 \le C_3\frac{1}{N} + C_4(R^2 \wedge \frac{1}{n}), \tag{90}$$

where

$$\begin{aligned} C_3 &= \frac{192\rho^2}{\mu^2}, \\ C_4 &= \left(\frac{\rho(1+(4C_2+5)^{\frac{1}{2}})}{\mu}\right)^2 \vee 198\kappa^2, \end{aligned} \tag{91}$$

and $C_2$ are specified in (80).

Therefore, combining results from Section A.3.1 and A.3.2 we can obtain the upper bound in (9).

To make the global statistical error bound and local statistical error bound hold simultaneously, we should be aware about the condition of the adaptive strategy of $\lambda$ specified in our proof. It could be summarized as

$$\begin{cases} \lambda \ge max(a_1, a_2) & R < \sqrt{\frac{m}{N}} \\ \lambda \le \frac{\rho^2}{n\mu R^2} & R \ge \sqrt{\frac{m}{N}} \end{cases}, \tag{92}$$

where

$$a_1 = \frac{8\kappa}{\mu}\left\{8L^2 \vee \left(\frac{2L^2R^2 + \rho^2/n}{4\rho^2/(\mu^2 N) + 4\kappa^2 R^2}\right)\right\} - \mu,$$

and

$$a_2 = \sqrt{\frac{8(64\rho^2/N + 66L^2R^2 + \rho^2/n)}{32\rho^2/(\mu^2 N) + 33\kappa^2 R^2}} - \mu.$$

We can induce a sufficient condition of $\lambda$ for $R < \frac{1}{\sqrt{n}}$ as

$$\lambda \ge \max\left\{64\kappa^2 L, (2\kappa \vee 5)\mu\frac{2L^2R^2 + \rho^2/n}{L^2R^2 + \rho^2/N}\right\} - \mu. \tag{93}$$

To sum up, we give the one-line rate in (90) for Corollary 1 and provide the potential solution of $\lambda$ with a closed-form solution in (92).

## A.5. Discussion on the Statistical Convergence Rate and Comparison with Existing Results

Combining the results in A.3.1 and A.3.2, we leave a remark below to further interpret the effect of personalization in Problem (2) and discuss how to incorporate the conditions of $\lambda$ when establishing the global and local statistical accuracy.

To better understand how such an interpolation is achieved through an adaptive personalization degree, we examine the role of $\lambda$ in navigating between the two extreme cases: GlobalTrain and LocalTrain. As the statistical heterogeneity $R \to \infty$, we have $\lambda \to 0$, leading to a high degree of personalization. The statistical error bound becomes $\mathcal{O}(n^{-1})$, matching the rate of LocalTrain; as the statistical heterogeneity $R \to 0$, $\lambda$ increases, leading to a low degree of personalization. The statistical error bound eventually converges to $\mathcal{O}(N^{-1})$, matching the rate of GlobalTrain under a homogeneous setting.

As $R \in (0, \infty)$, $\lambda$ transits between the two extremes, making PFL Problem (2) perform no worse than LocalTrain while benefiting from global training when clients share similarities.

Our contribution can be summarized into three aspects :

• Our results are derived under more realistic assumptions. Chen et al. (2023c) imposes additional stringent conditions including a bounded parameter space $D$ and a uniform bound on the loss function $\ell(\cdot, z)$. Such conditions are difficult to satisfy in practice. For example, even a simple linear regression model with sub-Gaussian covariates would violate these assumptions.

• Our result is established for both global and local models with the same choice of $\lambda$. In contrast, Chen et al. (2023c) proposes a two-stage process where the global model is first estimated, followed by an additional local training phase with a different choice of $\lambda$. This two-step approach is likely the artifact of their analysis, as existing work on Problem (2) also shows that a single-stage optimization with a chosen $\lambda$ suffices to achieve the desired performance (T Dinh et al., 2020).

• We establish a state-of-the-art minimax statistical rate. As shown in Table 3, the established upper bound matches the lower bound established in (Chen et al., 2023c). In a heterogeneous case when $R \geq n^{-1/2}$, Chen et al. (2023c) establishes a rate that is at least of the order $\mathcal{O}\left(n^{-1}R^2\right)$ as $D \geq R$. This bound becomes increasingly loose with larger $R$. In contrast, our bound is $\mathcal{O}\left(n^{-1}\right)$, which shows that properly-tuned Problem 2 is always no worse than LocalTrain, independent of $R$. In a homogeneous case when $R \leq m^{-1}n^{-1/2}$, they can only establish a rate slower than $\mathcal{O}\left(1/(\sqrt{mn})\right)$, while our result achieves a rate of $\mathcal{O}\left(1/(mn)\right)$, leveraging all $mn$ samples and matching the rate of GlobalTrain on the IID data. As the client number could be extremely large ($10^5$ devices in (Chen et al., 2023a)), order of $m$ is non-trivial. During the transition period when $m^{-1}n^{-1/2} < R < n^{-1/2}$, we achieve a rate of $\mathcal{O}(R^2)$, again matching the lower bound, and is strictly faster than the rate $\mathcal{O}(n^{-1/2}R)$ established in Chen et al. (2023c).

*Table 3.* Results Comparison. $D := \sup_{\boldsymbol{w}, \boldsymbol{w}' \in \mathcal{D}} \|\boldsymbol{w} - \boldsymbol{w}'\|$ is the diameter of the parameter space ($D \geq R$) and $\|\ell\|_\infty = \inf\{M \in \mathbb{R} : \ell(\cdot, z) \leq M, \text{for any } z\}$ is a uniform upper bound on the loss function. Statistical rate refers to $\mathbb{E}\|\boldsymbol{w}_\star^{(i)} - \widetilde{\boldsymbol{w}}^{(i)}\|^2$, with constants neglected.

| Source | Assumption | Paradigm | Statistical Rate | | |
|---|---|---|---|---|---|
| | | | $R > \frac{1}{\sqrt{n}}$ | $\frac{1}{m\sqrt{n}} < R \leq \frac{1}{\sqrt{n}}$ | $R \leq \frac{1}{m\sqrt{n}}$ |
| (Chen et al., 2023c) | A1,2,3, $D \vee \|\ell\|_\infty \leq C$ | Two-stage | $\frac{D^2 + \|\ell\|_\infty}{n}$ | $\frac{\|\ell\|_\infty}{\sqrt{n}} R$ | $\frac{\|\ell\|_\infty}{\sqrt{mn}}$ |
| Ours | A1,2,3 | One-stage | $\frac{1}{n}$ | $R^2 \wedge \frac{1}{mn}$ | $\frac{1}{mn}$ |
| Lower Bound | - | - | $\frac{1}{n}$ | $R^2 \wedge \frac{1}{mn}$ | $\frac{1}{mn}$ |

Next, we discuss our theoretical contributions. Prior work (Chen et al., 2023c) established the rate through algorithmic stability, and to obtain bounded stability, the boundedness of the loss function is needed. This assumption (or bounded gradients) is, in fact, commonly imposed for analysis based on the tool of algorithmic stability. To relax this assumption and obtain a bound independent of the diameter of the domain/gradient norm/loss function norm, we need to jump out of the stability analysis framework and develop new techniques.

Our analysis does not rely on any algorithm but directly tackles the objective function, establishing rates using purely the properties of the loss. Specifically, using the strong convexity of the loss function, we first proved the estimation error is bounded by the gradient variance and an extra statistical heterogeneity term controlled by $\lambda$, as detailed in Lemma 1. This allows us to show for large $R$, a small $\lambda$ can be chosen to yield rate $O(1/n)$ and match one of the worst cases in the lower bound. For the complementary case $R \leq 1/\sqrt{n}$, we leveraged the GlobalTrain solution $\widetilde{\boldsymbol{w}}_{GT}$ as a bridge and proved the solutions of Problem (2) to $\widetilde{\boldsymbol{w}}_{GT}$ are bounded by a term inversely proportional to $\lambda$ (cf. Lemma 4), implying that they can be made arbitrarily close to $\widetilde{\boldsymbol{w}}_{GT}$ by setting $\lambda$ small. This, together with the rate of $\widetilde{\boldsymbol{w}}_{GT}$ as proved in Lemma 5, yields a rate of $O(1/N + R^2)$. Combining the two cases, we proved minimax optimality.

# B. Algorithmic Convergence Rate

In this section, we will start to prove the algorithmic convergence rate rigorously. First, we present our Algorithm 1 in Appendix 1. In Appendix B.2, we provide some useful lemmas to facilitate our analysis, including the proof of Lemma 1 presented in the main paper and the inner loop and outer loop error contractions in Algorithm 1. Next, in Appendix B.3, we are ready to prove Theorem 2 with the local and global model convergence. In Appendix B.4, to show the communication cost and computation cost of our method, the proof of Corollary 2 is established. Finally, in Appendix B.5, as stochastic gradient descent would reduce the computation cost over the full sample, we present Algorithm 2 for the noise setting and provably show the benefit of collaboration to reduce the noise.

## B.1. Algorithm

---

**Algorithm 1** Federated Gradient Descent with K-Step Local Optimization

---

**Input:** Initial global model $\boldsymbol{w}_1^{(g)}$, initial local models $\{\boldsymbol{w}_{0,K}^{(i)}\}_{i\in[m]}$, global rounds $T$, global step sizes $\gamma$, local rounds $K$, local step sizes $\eta$

**Output:** Local models $\{\boldsymbol{w}_{T,K}^{(i)}\}_{i\in[m]}$ and global model $\boldsymbol{w}_T^{(g)}$

**for** $t = 1, \ldots, T$ **do**

    The server sends $\boldsymbol{w}_t^{(g)}$ to client $i$, $\forall i \in [m]$

    Set $\boldsymbol{w}_{t,0}^{(i)} = \boldsymbol{w}_{t-1,K}^{(i)}$ **for** $k = 0, \ldots, K-1$ **do**

        $\boldsymbol{w}_{t,k+1}^{(i)} = \boldsymbol{w}_{t,k}^{(i)} - \frac{\eta}{n_i} \sum_{j\in[n_i]} \left\{ \nabla \ell(\boldsymbol{w}_{t,k}^{(i)}, z_{i,j}) + \lambda \left( \boldsymbol{w}_{t,k}^{(i)} - \boldsymbol{w}_t^{(g)} \right) \right\}$

    Push $\widehat{\nabla} F_i(\boldsymbol{w}_t^{(g)}) = \lambda(\boldsymbol{w}_t^{(g)} - \boldsymbol{w}_{t,K}^{(i)})$ to the server

    $\boldsymbol{w}_{t+1}^{(g)} \leftarrow \boldsymbol{w}_t^{(g)} - \frac{\gamma}{m} \sum_{i\in[m]} \widehat{\nabla} F_i(\boldsymbol{w}_t^{(g)})$

---

## B.2. Proof of lemmas

### B.2.1. PROOF OF LEMMA 1

To facilitate our analysis, we state a claim first.

**Claim**: If $f$ is $\mu$-strongly-convex, then the proximal operator

$$\text{prox}_{f/\lambda}(x) = \arg\min_v \left( f(v) + \frac{\lambda}{2}\|x - v\|^2 \right), \tag{94}$$

is $L_w$- Lipschitz continuous with $L_w = \frac{\lambda}{\lambda+\mu}$.

*Proof.* Based on the definition of $\text{prox}_{f/\lambda}(x)$, it will satisfy the first order condition

$$\nabla f(\text{prox}_{\frac{1}{\lambda}f}(x)) - \lambda(x - \text{prox}_{\frac{1}{\lambda}f}(x)) = 0. \tag{95}$$

Then we have

$$\nabla f(x_1) - \lambda(v_1 - x_1) = 0 \quad \text{where } x_1 = \text{prox}_{\frac{1}{\lambda}f}(v), \tag{96}$$

$$\nabla f(x_2) - \lambda(v_2 - x_2) = 0 \quad \text{where } x_2 = \text{prox}_{\frac{1}{\lambda}f}(v_2). \tag{97}$$

Based on the $\mu$-strongly convexity of $f$, we have

$$\langle \nabla f(x_1) - \nabla f(x_2), x_1 - x_2 \rangle \geq \mu\|x_1 - x_2\|^2 \tag{98}$$

$$\Rightarrow \|\text{prox}_{\frac{1}{\lambda}f}(x_1) - \text{prox}_{\frac{1}{\lambda}f}(x_2)\| \leq \frac{1}{1 + \frac{\mu}{\lambda}}\|x_1 - x_2\|. \tag{99}$$

$\square$

Now, we are ready to prove Lemma 1.

First, from (Mishchenko et al., 2023), we can directly obtain the analytic properties of $F_i$. Next, the analytic properties of $h_i$ are direct results from Assumption 2 and 1. Finally, to analyze the property of the mapping $w_\star^{(i)}$, we just need to use the result from the above claim as we know the definition of $\boldsymbol{w}_\star^{(i)}(\cdot)$ in (11).

### B.2.2. INNER LOOP CONVERGENCE

**Lemma 5** ((Ji et al., 2022), Inner Loop Error Contraction). *Suppose Assumption 2 and 1 hold, then if $\eta \leq 1/L_\ell$, for all $t \geq 0$ and $\epsilon > 0$, the inner loop error bound of all clients can be formulated as:*

$$e_{t+1}^{(i)} := \|\boldsymbol{w}_{t+1,K}^{(i)} - \boldsymbol{w}_{t+1,\star}^{(i)}\|^2 \leq (1+\epsilon)(1-\eta\mu_\ell)^K e_t^{(i)} + \left(1 + \frac{1}{\epsilon}\right)(1-\eta\mu_\ell)^K \|\boldsymbol{w}_{t,\star}^{(i)} - \boldsymbol{w}_{t-1,\star}^{(i)}\|^2. \tag{100}$$

*Proof.* The proof can be found in Lemma 2 (Ji et al., 2022). $\qquad\square$

### B.2.3. OUTER LOOP CONVERGENCE

**Lemma 6** (Outer Loop Error Contraction). *Suppose Assumption 2 and 1 hold, then if $\gamma \leq L_g$, we have for all $t > 0$ and $\epsilon_g > 0$:*

$$\|\boldsymbol{w}_{t+1}^{(g)} - \widetilde{\boldsymbol{w}}^{(g)}\|^2 \leq (1 - \gamma\mu_g + \gamma\epsilon_g)\|\boldsymbol{w}_t^{(g)} - \widetilde{\boldsymbol{w}}^{(g)}\|^2 - \gamma^2\|\nabla F(\boldsymbol{w}_t^{(g)})\|^2$$
$$+ (\gamma^3 L_g)\|\widehat{\nabla} F(\boldsymbol{w}_t^{(g)})\|^2 + \left(\frac{\gamma}{\epsilon_g} + \gamma^2\right)\|\nabla F(\boldsymbol{w}_t^{(g)}) - \widehat{\nabla} F(\boldsymbol{w}_t^{(g)})\|^2. \tag{101}$$

*Proof.* Note

$$\widehat{\nabla} F_i(\boldsymbol{w}_t^{(g)}) = \lambda(\boldsymbol{w}_t^{(g)} - \boldsymbol{w}_{t,K}^{(i)}). \tag{102}$$

We define

$$\widehat{\nabla} F(\boldsymbol{w}_t^{(g)}) := \frac{1}{m}\sum_{i\in[m]} \widehat{\nabla} F_i(\boldsymbol{w}_t^{(g)}) = \frac{1}{m}\sum_{i\in[m]} \lambda(\boldsymbol{w}_t^{(g)} - \boldsymbol{w}_{t,K}^{(i)}). \tag{103}$$

Thus the global update rule can be written as $\boldsymbol{w}_{t+1}^{(g)} = \boldsymbol{w}_t^{(g)} - \gamma\widehat{\nabla} F(\boldsymbol{w}_t^{(g)})$. For the outer loop,

$$\begin{aligned}
&\|\boldsymbol{w}_{t+1}^{(g)} - \widetilde{\boldsymbol{w}}^{(g)}\|^2 \\
=&\|\boldsymbol{w}_t^{(g)} - \widetilde{\boldsymbol{w}}^{(g)} - \gamma\widehat{\nabla} F(\boldsymbol{w}_t^{(g)})\|^2 \\
=&\|\boldsymbol{w}_t^{(g)} - \widetilde{\boldsymbol{w}}^{(g)}\|^2 - 2\gamma\langle \boldsymbol{w}_t^{(g)} - \widetilde{\boldsymbol{w}}^{(g)}, \widehat{\nabla} F(\boldsymbol{w}_t^{(g)})\rangle + \gamma^2\|\widehat{\nabla} F(\boldsymbol{w}_t^{(g)})\|^2 \\
\leq&(1 + \epsilon_g\gamma)\|\boldsymbol{w}_t^{(g)} - \widetilde{\boldsymbol{w}}^{(g)}\|^2 - 2\gamma\langle \boldsymbol{w}_t^{(g)} - \widetilde{\boldsymbol{w}}^{(g)}, \nabla F(\boldsymbol{w}_t^{(g)})\rangle + \gamma^2\|\widehat{\nabla} F(\boldsymbol{w}_t^{(g)})\|^2 \\
&+ \left(\frac{\gamma}{\epsilon_g}\right)\|\widehat{\nabla} F(\boldsymbol{w}_t^{(g)}) - \nabla F(\boldsymbol{w}_t^{(g)})\|^2,
\end{aligned} \tag{104}$$

where we apply the Young's inequality with $\epsilon_g > 0$ in the last line.

Next, we bound the inner product term in the last line using the $\mu_g$-strong convexity of $F$:

$$\begin{aligned}
-\gamma\langle \boldsymbol{w}_t^{(g)} - \widetilde{\boldsymbol{w}}^{(g)}, \nabla F(\boldsymbol{w}_t^{(g)})\rangle &\leq -\gamma\left(F(\boldsymbol{w}_t^{(g)}) - \widehat{F} + \frac{\mu_g}{2}\|\boldsymbol{w}_t^{(g)} - \widetilde{\boldsymbol{w}}^{(g)}\|^2\right) \\
&= -\gamma(F(\boldsymbol{w}_{t+1}^{(g)}) - \widehat{F}) - \frac{\gamma\mu_g}{2}\|\boldsymbol{w}_t^{(g)} - \widetilde{\boldsymbol{w}}^{(g)}\|^2 + \gamma\left(F(\boldsymbol{w}_{t+1}^{(g)}) - F(\boldsymbol{w}_t^{(g)})\right).
\end{aligned} \tag{105}$$

where $\hat{F} := F(\widetilde{\boldsymbol{w}}^{(g)})$ and the last term $F(\boldsymbol{w}_{t+1}^{(g)}) - F(\boldsymbol{w}_t^{(g)})$ can be upper bounded using the $L_g$-smoothness of $F$:

$$
\begin{aligned}
F(\boldsymbol{w}_{t+1}^{(g)}) \leq &F(\boldsymbol{w}_t^{(g)}) - \gamma\langle\nabla F(\boldsymbol{w}_t^{(g)}), \widehat{\nabla}F(\boldsymbol{w}_t^{(g)})\rangle + \frac{L_g\gamma^2}{2}\|\widehat{\nabla}F(\boldsymbol{w}_t^{(g)})\|^2 \\
&\overset{(a)}{=} F(\boldsymbol{w}_t^{(g)}) + \frac{\gamma}{2}\left(\|\nabla F(\boldsymbol{w}_t^{(g)}) - \widehat{\nabla}F(\boldsymbol{w}_t^{(g)})\|^2 - \|\nabla F(\boldsymbol{w}_t^{(g)})\|^2 - \|\widehat{\nabla}F(\boldsymbol{w}_t^{(g)})\|^2\right) \\
&\quad + \frac{L_g\gamma^2}{2}\|\widehat{\nabla}F(\boldsymbol{w}_t^{(g)})\|^2 \\
= &F(\boldsymbol{w}_t^{(g)}) - \frac{\gamma}{2}\|\nabla F(\boldsymbol{w}_t^{(g)})\|^2 - \left(\frac{\gamma}{2} - \frac{L_g\gamma^2}{2}\right)\|\widehat{\nabla}F(\boldsymbol{w}_t^{(g)})\|^2 + \frac{\gamma}{2}\|\nabla F(\boldsymbol{w}_t^{(g)}) - \widehat{\nabla}F(\boldsymbol{w}_t^{(g)})\|^2,
\end{aligned}
\tag{106}
$$

where the step (a) uses the fact $-\langle a, b\rangle = \frac{1}{2}\|a - b\|^2 - \frac{1}{2}\|a\|^2 - \frac{1}{2}\|b\|^2$.

Substituting (105) and (106) into (104) leads to

$$
\begin{aligned}
&\|\boldsymbol{w}_{t+1}^{(g)} - \widetilde{\boldsymbol{w}}^{(g)}\|^2 \\
\leq &(1 + \epsilon_g\gamma)\|\boldsymbol{w}_t^{(g)} - \widetilde{\boldsymbol{w}}^{(g)}\|^2 + \gamma^2\|\widehat{\nabla}F(\boldsymbol{w}_t^{(g)})\|^2 + \left(\frac{\gamma}{\epsilon_g}\right)\|\widehat{\nabla}F(\boldsymbol{w}_t^{(g)}) - \nabla F(\boldsymbol{w}_t^{(g)})\|^2 \\
&- 2\gamma(F(\boldsymbol{w}_{t+1}^{(g)}) - \hat{F}) - \gamma\mu_g\|\boldsymbol{w}_t^{(g)} - \widetilde{\boldsymbol{w}}^{(g)}\|^2 \\
&- \gamma^2\|\nabla F(\boldsymbol{w}_t^{(g)})\|^2 - \gamma\left(\gamma - L_g\gamma^2\right)\|\widehat{\nabla}F(\boldsymbol{w}_t^{(g)})\|^2 + \gamma^2\|\nabla F(\boldsymbol{w}_t^{(g)}) - \widehat{\nabla}F(\boldsymbol{w}_t^{(g)})\|^2 \\
\leq &(1 - \gamma\mu_g + \gamma\epsilon_g)\|\boldsymbol{w}_t^{(g)} - \widetilde{\boldsymbol{w}}^{(g)}\|^2 - \gamma^2\|\nabla F(\boldsymbol{w}_t^{(g)})\|^2 + (\gamma^3 L_g)\|\widehat{\nabla}F(\boldsymbol{w}_t^{(g)})\|^2 \\
&+ \left(\frac{\gamma}{\epsilon_g} + \gamma^2\right)\|\nabla F(\boldsymbol{w}_t^{(g)}) - \widehat{\nabla}F(\boldsymbol{w}_t^{(g)})\|^2.
\end{aligned}
\tag{107}
$$

$\square$

## B.3. Proof of Theorem 2

### B.3.1. PROOF OF THE GLOBAL MODEL CONVERGENCE RATE IN THEOREM 2

Recall the definition of $e_t^{(i)}$ in Lemma 5, we define

$$
e_t = \frac{1}{m}\sum_{i\in[m]} e_t^{(i)}.
\tag{108}
$$

Recall the definition $\nabla F(\boldsymbol{w}_t^{(g)}), \widehat{\nabla}F(\boldsymbol{w}_t^{(g)})$ in Lemma 2 and eq (103), the gradient approximation error is

$$
\|\nabla F(\boldsymbol{w}_t^{(g)}) - \widehat{\nabla}F(\boldsymbol{w}_t^{(g)})\|^2 = \lambda^2\left\|\frac{1}{m}\sum_{i=1}^m\left(\boldsymbol{w}_{t,K}^{(i)} - \boldsymbol{w}_{t,\star}^{(i)}\right)\right\|^2 \leq \lambda^2 e_t.
\tag{109}
$$

Therefore, the recursion of optimization error given by Lemma 6 can be further bounded as:

$$
\begin{aligned}
&\|\boldsymbol{w}_{t+1}^{(g)} - \widetilde{\boldsymbol{w}}^{(g)}\|^2 \\
\overset{(a)}{\leq} &(1 - \gamma\mu_g + \gamma\epsilon_g)\|\boldsymbol{w}_t^{(g)} - \widetilde{\boldsymbol{w}}^{(g)}\|^2 - \left(\gamma^2 - \gamma^3 L_g(1+\zeta)\right)\|\nabla F(\boldsymbol{w}_t^{(g)})\|^2 \\
&+ \left(\frac{\gamma}{\epsilon_g} + \gamma^2 + L_g\gamma^3(1+\zeta^{-1})\right)\|\nabla F(\boldsymbol{w}_t^{(g)}) - \widehat{\nabla}F(\boldsymbol{w}_t^{(g)})\|^2 \\
\overset{(b)}{\leq} &(1 - \gamma\mu_g + \gamma\epsilon_g)\|\boldsymbol{w}_t^{(g)} - \widetilde{\boldsymbol{w}}^{(g)}\|^2 - \gamma^2\left(1 - \gamma L_g(1+\zeta)\right)\|\nabla F(\boldsymbol{w}_t^{(g)})\|^2 \\
&+ \left(\frac{\gamma}{\epsilon_g} + \gamma^2(2 + \zeta^{-1})\right)\lambda^2 e_t,
\end{aligned}
\tag{110}
$$

where the step (a) comes from the Young's inequality with $\zeta > 0$ to be chosen and the step (b) follows from the step size condition $\gamma \leq 1/L_g$.

Invoking Lemma 1 and Lemma 5, the inner loop optimization error is updated as

$$
\begin{aligned}
e_{t+1} &\leq (1+\epsilon)(1-\eta\mu_\ell)^K e_t + \left(1+\frac{1}{\epsilon}\right)(1-\eta\mu_\ell)^K \frac{1}{m}\sum_{i=1}^m \|\boldsymbol{w}_{t+1,\star}^{(i)} - \boldsymbol{w}_{t,\star}^{(i)}\|^2 \\
&\overset{(a)}{\leq} (1+\epsilon)(1-\eta\mu_\ell)^K e_t + \left(1+\frac{1}{\epsilon}\right)(1-\eta\mu_\ell)^K L_w^2 \|\boldsymbol{w}_t^{(g)} - \boldsymbol{w}_{t+1}^g\|^2 \\
&\overset{(b)}{\leq} (1+\epsilon)(1-\eta\mu_\ell)^K e_t + \left(1+\frac{1}{\epsilon}\right)(1-\eta\mu_\ell)^K L_w^2 \gamma^2 \|\widehat{\nabla}F(\boldsymbol{w}_t^{(g)})\|^2 \\
&\overset{(c)}{\leq} (1+\epsilon)(1-\eta\mu_\ell)^K e_t + \left(1+\frac{1}{\epsilon}\right)(1-\eta\mu_\ell)^K L_w^2 \gamma^2 \left(2\|\nabla F(\boldsymbol{w}_t^{(g)})\|^2 + 2\lambda^2 e_t\right) \\
&= \underbrace{\left(1+\epsilon+2(1+\epsilon^{-1})L_w^2\gamma^2\lambda^2\right)(1-\eta\mu_\ell)^K}_{:=q} e_t + 2\left(1+\frac{1}{\epsilon}\right)(1-\eta\mu_\ell)^K L_w^2\gamma^2\|\nabla F(\boldsymbol{w}_t^{(g)})\|^2,
\end{aligned}
$$

(111)

where step (a) is due to Lemma 1, step (b) uses the global update: $\boldsymbol{w}_{t+1}^{(g)} - \boldsymbol{w}_t^{(g)} = -\gamma\widehat{\nabla}F(\boldsymbol{w}_t^{(g)})$ and in step (c), we just apply Cauchy inequality instead of Young's inequality.

Under $\eta \leq 1/L_\ell$, we have $1 - \eta\mu_\ell < 1$. Therefore, by choosing a large $K$ we can always drive $q$ arbitrarily small. Let $K$ be such that

$$
q \leq 1 - \gamma\mu_g \leq 1 - \gamma\mu_g + \gamma\epsilon_g. \tag{112}
$$

Then, we can rewrite (111) as:

$$
\begin{aligned}
e_{t+1} \leq &\frac{1-\gamma\mu_g+\gamma\epsilon_g+q}{2}e_t - \frac{1-\gamma\mu_g+\gamma\epsilon_g-q}{2}e_t \\
&+ 2\left(1+\frac{1}{\epsilon}\right)(1-\eta\mu_\ell)^K L_w^2\gamma^2\|\nabla F(\boldsymbol{w}_t^{(g)})\|^2.
\end{aligned}
$$

(113)

Letting $c_1 = \frac{2}{1-\gamma\mu_g+\gamma\epsilon_g-q} \cdot \left(\frac{\gamma}{\epsilon_g} + \gamma^2(2+\zeta^{-1})\right)\lambda^2$, we can combine (113) with (110) and obtain

$$
\begin{aligned}
&\|\boldsymbol{w}_{t+1}^{(g)} - \widetilde{\boldsymbol{w}}^{(g)}\|^2 + c_1 \cdot e_{t+1} \\
\leq &(1-\gamma\mu_g+\gamma\epsilon_g)\left(\|\boldsymbol{w}_t^{(g)} - \widetilde{\boldsymbol{w}}^{(g)}\|^2 + c_1 \cdot e_t\right) - \gamma^2\left(1-\gamma L_g(1+\zeta)\right)\|\nabla F(\boldsymbol{w}_t^{(g)})\|^2 \\
&+ \left(\frac{2}{1-\gamma\mu_g+\gamma\epsilon_g-q} \cdot \left(\frac{\gamma}{\epsilon_g} + \gamma^2(2+\zeta^{-1})\right)\lambda^2\right) \cdot 2\left(1+\frac{1}{\epsilon}\right)(1-\eta\mu_\ell)^K L_w^2\gamma^2\|\nabla F(\boldsymbol{w}_t^{(g)})\|^2.
\end{aligned}
$$

(114)

Therefore, if the condition (112) and

$$
\frac{4\lambda^2\gamma}{1-\gamma\mu_g+\gamma\epsilon_g-q}\left(\frac{1}{\epsilon_g}+\gamma(2+\zeta^{-1})\right)\left(1+\frac{1}{\epsilon}\right)(1-\eta\mu_\ell)^K L_w^2 \leq 1 - \gamma L_g(1+\zeta) \tag{115}
$$

hold then (114) implies both $\|\boldsymbol{w}_t^{(g)} - \widetilde{\boldsymbol{w}}^{(g)}\|^2$ and $e_t$ converge to zero at rate $1 - \gamma\mu_g + \gamma\epsilon_g$.

It remains to specify the free parameters $(\epsilon_g, \epsilon, \zeta)$ in (112), (115) and others listed in Lemma 6 and Lemma 5.

• *Convergence Rate.* For the connection term in (114), if we set

$$
\epsilon_g = \frac{\mu_g}{2}\cdot(1-\gamma\mu_g) > 0, \tag{116}
$$

which, substituting into $1 - \gamma\mu_g + \gamma\epsilon_g$, gives the convergence rate

$$r := 1 - \frac{\gamma\mu_g}{2} - \frac{(\gamma\mu_g)^2}{2}. \tag{117}$$

Under condition $\gamma \leq 1/L_g$, one can verify that $r \in (0,1)$.

- *Step size conditions.* Under the requirement (112), we have

$$1 - \gamma\mu_g + \gamma\epsilon_g - q \geq \gamma\epsilon_g, \tag{118}$$

which gives the following sufficient condition for (115):

$$\frac{2}{\epsilon_g} \cdot \left(\frac{1}{\epsilon_g} + \gamma(2 + \zeta^{-1})\right)\lambda^2 \cdot 2\left(1 + \frac{1}{\epsilon}\right)(1 - \eta\mu_\ell)^K L_w^2 \leq 1 - \gamma L_g(1 + \zeta). \tag{119}$$

Further restricting $\gamma$ such that

$$\gamma(2 + \zeta^{-1})\frac{\mu_g}{2} \leq 1 \quad \Longrightarrow \quad \gamma(2 + \zeta^{-1}) \leq \frac{1}{\epsilon_g}, \tag{120}$$

it suffices to require the following for (119) to hold:

$$\left(\frac{2}{\epsilon_g}\right)^2 \lambda^2 \cdot 2\left(1 + \frac{1}{\epsilon}\right)(1 - \eta\mu_\ell)^K L_w^2 \leq 1 - \gamma L_g(1 + \zeta). \tag{121}$$

Letting $\epsilon = 1$ and $\zeta = 1 - \gamma L_g > 0$, we collect all the conditions (116), (121) and (112) on $\gamma$ respectively as follows:

$$1 - \gamma L_g > 0, \quad 1 - \gamma\mu_g > 0, \tag{122}$$

$$(1 - \gamma\mu_g)^2(1 - \gamma L_g)^2 \geq 4\lambda^2 \cdot 4(1 - \eta\mu_\ell)^K L_w^2 \cdot \left(\frac{2}{\mu_g}\right)^2, \tag{123}$$

$$1 - \gamma\mu_g \geq \left(2 + 4L_w^2\gamma^2\lambda^2\right)(1 - \eta\mu_\ell)^K. \tag{124}$$

Using the fact that $\mu_g \leq L_g$, the above conditions simply to

$$\gamma < 1/L_g, \quad \left(2 + 64L_w^2(1/\mu_g)^2\lambda^2\right)(1 - \eta\mu_\ell)^K \leq (1 - \gamma L_g)^4. \tag{125}$$

### B.3.2. PROOF OF THE LOCAL MODEL CONVERGENCE RATE IN THEOREM 2

*Proof.* Note that

$$\|\boldsymbol{w}_{t,K}^{(i)} - \widetilde{\boldsymbol{w}}^{(i)}\|^2 \leq 2\|\boldsymbol{w}_{t,K}^{(i)} - \boldsymbol{w}_{t,\star}^{(i)}\|^2 + 2\|\boldsymbol{w}_{t,\star}^{(i)} - \widetilde{\boldsymbol{w}}^{(i)}\|^2. \tag{126}$$

For the first part on the right-hand side, recalling the definition of $e_t^{(i)}$ in Lemma 5

$$2\|\boldsymbol{w}_{t,K}^{(i)} - \boldsymbol{w}_{t,\star}^{(i)}\|^2 = 2\|e_t^{(i)}\|^2. \tag{127}$$

For the second term on the right-hand side, first, note the property in (11)

$$\boldsymbol{w}_{t,\star}^{(i)} = \text{prox}_{L_i/\lambda}(\boldsymbol{w}_t^{(g)}). \tag{128}$$

Based on the first order condition of (2)

$$\widetilde{\boldsymbol{w}}^{(i)} = \text{prox}_{L_i/\lambda}(\widetilde{\boldsymbol{w}}^{(g)}). \tag{129}$$

Next, plugging (128) and (129) into (126), it yields

$$
\begin{aligned}
\|\boldsymbol{w}_{t,K}^{(i)} - \widetilde{\boldsymbol{w}}^{(i)}\|^2 &\leq 2\|e_t^{(i)}\|^2 + 2\|\operatorname{prox}_{L_i/\lambda}(\boldsymbol{w}_t^{(g)}) - \operatorname{prox}_{L_i/\lambda}(\widetilde{\boldsymbol{w}}^{(g)})\|^2 \\
&\leq 2\|e_t^{(i)}\|^2 + 2L_w^2\|\boldsymbol{w}_{t,\star}^{(i)} - \widetilde{\boldsymbol{w}}^{(i)}\|^2,
\end{aligned}
\tag{130}
$$

where the last we use Lemma 1.

In the proof of Theorem 2, we have established that both $\|\boldsymbol{w}_t^{(g)} - \widetilde{\boldsymbol{w}}^{(g)}\|^2$ and $e_t$ converges to zero linearly at rate $1 - \gamma\mu_g/2 - (\gamma\mu_g)^2/2$, combining with (130), the proof is finished. $\qquad\square$

### B.4. Proof of Corollary 2

With $\eta = 1/L_\ell = (\lambda + L)^{-1}$, $\gamma = 1/(2L_g) = \frac{\lambda+L}{2\lambda L}$, and using the fact that $L_w = \frac{\lambda}{\lambda+\mu}$ and $\mu_g = \frac{\lambda\mu}{\lambda+\mu}$ (cf. Lemma 1), condition (125) for $K$ becomes

$$
\begin{aligned}
&\left(2 + 64L_w^2(1/\mu_g)^2\lambda^2\right)(1 - \eta\mu_\ell)^K \\
&= \left(2 + 64\frac{\lambda^2}{(\lambda+\mu)^2}\frac{(\lambda+\mu)^2}{(\lambda\mu)^2}\lambda^2\right)\left(1 - \frac{\lambda+\mu}{\lambda+L}\right)^K \\
&= \left(2 + 64\frac{\lambda^2}{\mu^2}\right)\left(\frac{L-\mu}{\lambda+L}\right)^K \leq \frac{1}{16}.
\end{aligned}
\tag{131}
$$

If $L = \mu$, then the above condition trivially holds. Otherwise, a sufficient condition for it is

$$
66\kappa^2\left(\frac{L-\mu}{\lambda+L}\right)^{K-2} \leq \frac{1}{16},
\tag{132}
$$

where we have used the fact that

$$
\frac{\lambda^2}{\mu^2}\left(\frac{L-\mu}{\lambda+L}\right)^K = \left(\frac{\lambda}{\lambda+L}\right)^2 \cdot \left(\frac{L-\mu}{\mu}\right)^2\left(\frac{L-\mu}{\lambda+L}\right)^{K-2} \leq \kappa^2\left(\frac{L-\mu}{\lambda+L}\right)^{K-2}
\tag{133}
$$

and $\kappa := L/\mu \geq 1$. Finally, using the inequality $\log(1/x) \geq 1 - x$ for $0 < x \leq 1$ we obtain

$$
K \geq 2 + \frac{\lambda+L}{\lambda+\mu}\cdot\log(1056\kappa^2).
\tag{134}
$$

### B.5. Global Convergence Rate with Stochastic Noise

Similar to the noiseless case, we propose FedCLUP below which uses stochastic gradient descent to solve Problem (2).

---

**Algorithm 2** FedCLUP: Federated Learning with Constant Local Update Personalization

---

**Input:** Initial global model $\boldsymbol{w}_1^{(g)}$, initial local models $\{\boldsymbol{w}_{0,K}^{(i)}\}_{i\in[m]}$, global rounds $T$, global step sizes $\gamma$, local rounds $K$, local step sizes $\eta$, stochastic batch size $B$

**Output:** Local models $\{\boldsymbol{w}_{T,K}^{(i)}\}_{i\in[m]}$ and global model $\boldsymbol{w}_T^{(g)}$

**for** $t = 1, \ldots, T$ **do**
 The server sends $\boldsymbol{w}_t^{(g)}$ to client $i$, $\forall i \in [m]$;
 Each client randomly draw $B$ data without replacement;
 Set $\boldsymbol{w}_{t,0}^{(i)} = \boldsymbol{w}_{t-1,K}^{(i)}$ **for** $k = 0, \ldots, K-1$ **do**
  $\boldsymbol{w}_{t,k+1}^{(i)} = \boldsymbol{w}_{t,k}^{(i)} - \frac{\eta}{B}\sum_{j\in[B]}\left\{\nabla\ell(\boldsymbol{w}_{t,k}^{(i)}, z_{i,j}) + \lambda\left(\boldsymbol{w}_{t,k}^{(i)} - \boldsymbol{w}_t^{(g)}\right)\right\}$
 Push $\widehat{\nabla}F_i(\boldsymbol{w}_t^{(g)}) = \lambda(\boldsymbol{w}_t^{(g)} - \boldsymbol{w}_{t,K}^{(i)})$ to the server
 $\boldsymbol{w}_{t+1}^{(g)} \leftarrow \boldsymbol{w}_t^{(g)} - \frac{\gamma}{m}\sum_{i\in[m]}\widehat{\nabla}F_i(\boldsymbol{w}_t^{(g)})$

---

We use the following assumption to characterize the stochastic noise when sampling new data in each iteration over all clients without replacement.

**Assumption 4** (Stochastic noise). *For any $i \in [m], j \in [n_i], k \in [K], t > 0$, we assume*

$$\mathbb{E}\left[\left\|\nabla_{\boldsymbol{w}_i}\hat{h}_i(\boldsymbol{w}_{t,k}^{(i)}, \boldsymbol{w}_t^{(i)}, s_{ij}) - \nabla_{\boldsymbol{w}_i}h_i(\boldsymbol{w}_{t,\star}^{(i)}, \boldsymbol{w}_t^{(i)})\right\|^2\right] \leq \sigma^2 \tag{135}$$

Let we define $\mathcal{F}_{t,k}$ to be the sigma algebra generated by the randomness by Algorithm 2 up to $\boldsymbol{w}_{t,k}^{(i)}$.

**Lemma 7** (Inner Loop Error Contraction with stochastic). *Suppose Assumption 1 and 2 hold, then if $\eta \leq 1/L_\ell$, for all $t \geq 0$ and $\epsilon > 0$, the inner loop error bound of all clients can be formulated as:*

$$
\begin{aligned}
g_{t+1} = \mathbb{E}&\left(\left\|\frac{1}{m}\sum_{i\in[m]}\boldsymbol{w}_{t+1,k}^{(i)} - \boldsymbol{w}_{t+1,*}^{(i)}\right\|^2 \bigg| \mathcal{F}_{t+1,k}\right) \\
\leq &\left[1 + \epsilon + 2\left(1 + \frac{1}{\epsilon}\right)L_w^2\gamma^2\lambda^2\right](1 - \eta_t\mu_\ell)^k g_t \\
&+ 2\left(1 + \frac{1}{\epsilon}\right)(1 - \mu_\ell\eta_t)^k L_w^2\gamma^2\|\nabla F(\boldsymbol{w}_t^{(g)})\|^2 \\
&+ \frac{2\eta_t}{\mu_\ell}\frac{\sigma^2}{Bm}.
\end{aligned}
\tag{136}
$$

*Proof.*

$$
\begin{aligned}
g_{t+1} = \mathbb{E}&\left(\left\|\frac{1}{m}\sum_{i\in[m]}\boldsymbol{w}_{t+1,k}^{(i)} - \boldsymbol{w}_{t+1,*}^{(i)}\right\|^2 \bigg| \mathcal{F}_{t+1,k}\right) \\
= \mathbb{E}&\left(\left\|\frac{1}{m}\sum_{i\in[m]}\left[\boldsymbol{w}_{t+1,k-1}^{(i)} - \frac{\eta_t}{B}\sum_j\nabla_{w_i}\hat{h}_i(\boldsymbol{w}_{t+1,k-1}^{(i)}, \boldsymbol{w}_{t+1}^{(i)}, s_{ij}) - \boldsymbol{w}_{t+1,*}^{(i)}\right]\right\|^2 \bigg| \mathcal{F}_{t+1,k}\right) \\
= \mathbb{E}&\left(\left\|\frac{1}{m}\sum_i\boldsymbol{w}_{t+1,k-1}^{(i)} - \boldsymbol{w}_{t+1,*}^{(i)}\right\|^2 \bigg| \mathcal{F}_{t+1,k}\right) \\
&+ \eta_t^2\mathbb{E}\left(\left\|\frac{1}{mB}\sum_i\sum_j\nabla_{w_i}\hat{h}_i(\boldsymbol{w}_{t+1,k-1}^{(i)}, \boldsymbol{w}_{t+1}^{(i)}, s_{ij})\right\|^2 \bigg| \mathcal{F}_{t+1,k}\right) \\
&- 2\eta_t\mathbb{E}\left[\left\langle\frac{1}{m}\sum_i\boldsymbol{w}_{t+1,k-1}^{(i)} - \boldsymbol{w}_{t+1,*}^{(i)}, \frac{1}{Bm}\sum_j\nabla_{w_i}\hat{h}_i(\boldsymbol{w}_{t+1,k-1}^{(i)}, \boldsymbol{w}_{t+1}^{(i)}, s_{ij})\right\rangle \bigg| \mathcal{F}_{t+1,k}\right].
\end{aligned}
\tag{137}
$$

To simplify the result, we shall note

$$
\mathbb{E}\left(\left\|\frac{1}{mB}\sum_i\sum_j\nabla_{\boldsymbol{w}_i}\hat{h}_i(\boldsymbol{w}_{t+1,k-1}^{(i)},\boldsymbol{w}_{t+1}^{(i)},s_{ij})\right\|^2\bigg|\mathcal{F}_{t+1,k}\right)
$$

$$
=\mathbb{E}\left(\left\|\frac{1}{mB}\sum_i\sum_j\nabla_{\boldsymbol{w}_i}\hat{h}_i(\boldsymbol{w}_{t+1,k-1}^{(i)},\boldsymbol{w}_{t+1}^{(i)},s_{ij})-\nabla_{\boldsymbol{w}_i}h_i(\boldsymbol{w}_{t+1,k-1}^{(i)},\boldsymbol{w}_{t+1}^{(i)})\right\|^2\bigg|\mathcal{F}_{t+1,k}\right)
$$

$$
+\left\|\frac{1}{m}\sum_i\nabla_{\boldsymbol{w}_i}h_i(\boldsymbol{w}_{t+1,k-1}^{(i)},\boldsymbol{w}_{t+1}^{(i)})\right\|^2
$$

$$
\overset{(a)}{\leq}\frac{\sigma^2}{mB}+\left\|\frac{1}{m}\sum_i\left(\nabla_{\boldsymbol{w}_i}h_i(\boldsymbol{w}_{t+1,k-1}^{(i)},\boldsymbol{w}_{t+1}^{(i)})-\nabla_{\boldsymbol{w}_i}h_i(\boldsymbol{w}_{t+1,\star}^{(i)},\boldsymbol{w}_{t+1}^{(i)})\right)\right\|^2
$$

$$
\overset{(b)}{\leq}\frac{\sigma^2}{mB}+2L_\ell\frac{1}{m}\sum_i\left(h_i(\boldsymbol{w}_{t+1,k-1}^{(i)},\boldsymbol{w}_{t+1}^{(i)})-h_i(\boldsymbol{w}_{t+1,*}^{(i)},\boldsymbol{w}_{t+1}^{(i)})\right). \tag{138}
$$

where step (a) stands due to Assumption 4 and the optimality of $\boldsymbol{w}_{t+1,\star}^{(i)}$, step (b) uses the Lemma 2.29 in (Garrigos & Gower, 2023). Moreover, for the third term in (137), it yields

$$
\mathbb{E}\left[\left\langle\frac{1}{m}\sum_i\boldsymbol{w}_{t+1,k-1}^{(i)}-\boldsymbol{w}_{t+1,*}^{(i)},\frac{1}{Bm}\sum_{i,j}\nabla_{\boldsymbol{w}_i}\hat{h}_i(\boldsymbol{w}_{t+1,k-1}^{(i)},\boldsymbol{w}_{t+1}^{(i)},s_j)\right\rangle\bigg|\mathcal{F}_{t+1,k}\right]
$$

$$
=\left\langle\frac{1}{m}\sum_i\boldsymbol{w}_{t+1,k-1}^{(i)}-\boldsymbol{w}_{t+1,*}^{(i)},\frac{1}{m}\sum_i\nabla_{\boldsymbol{w}_i}h_i(\boldsymbol{w}_{t+1,k-1}^{(i)},\boldsymbol{w}_{t+1}^{(i)})\right\rangle \tag{139}
$$

$$
\geq\frac{1}{m}\sum_i\left(h_i(\boldsymbol{w}_{t+1,k-1}^{(i)},\boldsymbol{w}_{t+1}^{(i)})-h_i(\boldsymbol{w}_{t+1,*}^{(i)},\boldsymbol{w}_{t+1}^{(i)})\right)+\frac{\mu_\ell}{2}\left\|\frac{1}{m}\sum_i\boldsymbol{w}_{t+1,k-1}^{(i)}-\boldsymbol{w}_{t+1,*}^{(i)}\right\|^2,
$$

where the last line comes from the strong convexity of $h_i$ in Assumption 2.

Plugging the results (138) and (139) into (137), we obtain

$$
\mathbb{E}\left[\left\|\frac{1}{m}\sum_i\boldsymbol{w}_{t+1,k}^{(i)}-\boldsymbol{w}_{t+1,*}^{(i)}\right\|^2\bigg|\mathcal{F}_{t+1,k}\right]
$$

$$
\leq(1-\eta_t\mu_\ell)\left\|\frac{1}{m}\sum_i\boldsymbol{w}_{t+1,k-1}^{(i)}-\boldsymbol{w}_{t+1,*}^{(i)}\right\|^2
$$

$$
+2\eta_t(L_\ell\eta_t-1)\left(\frac{1}{m}\sum_i\left(h_i(\boldsymbol{w}_{t+1,k-1}^{(i)},\boldsymbol{w}_{t+1}^{(i)})-h_i(\boldsymbol{w}_{t+1,*}^{(i)},\boldsymbol{w}_{t+1}^{(i)})\right)\right) \tag{140}
$$

$$
+\eta_t^2\frac{\sigma^2}{Bm}.
$$

If we assume $\eta_t\leq\frac{1}{L_\ell}$ we can simplify (140) as

$$\mathbb{E}\left[\left\|\frac{1}{m}\sum_i \boldsymbol{w}_{t+1,k}^{(i)} - \boldsymbol{w}_{t+1,*}^{(i)}\right\|^2 \Big| \mathcal{F}_{t+1,k}\right]$$

$$\leq (1-\mu_\ell\eta_t)\left\|\frac{1}{m}\sum_i \boldsymbol{w}_{t+1,k-1}^{(i)} - \boldsymbol{w}_{t+1,*}^{(i)}\right\|^2 + \eta_t^2\frac{\sigma^2}{Bm}$$

$$\leq (1-\mu_\ell\eta_t)^k\left\|\frac{1}{m}\sum_i \boldsymbol{w}_{t,k}^{(i)} - \boldsymbol{w}_{t,*}^{(i)} + \boldsymbol{w}_{t,*}^{(i)} - \boldsymbol{w}_{t+1,*}^{(i)}\right\|^2 + \sum_{l=0}^{k-1}(1-\mu_\ell\eta_t)^{k-l}\eta_t^2\frac{\sigma^2}{Bm}$$

$$\leq (1+\epsilon)(1-\mu_\ell\eta_t)^k g_t + \sum_{l=0}^{k-1}(1-\mu_\ell\eta_t)^{k-l}\eta_t^2\frac{\sigma^2}{Bm} + \left(1+\frac{1}{\epsilon}\right)(1-\mu_\ell\eta_t)^k L_w^2\gamma^2\left(2\|\nabla F(\boldsymbol{w}_t^{(g)})\|^2 + 2\lambda^2 g_t\right).$$

$$(141)$$

Reorganizing all the terms in (141), we get

$$\mathbb{E}\left[\left\|\frac{1}{m}\sum_i \boldsymbol{w}_{t+1,k}^{(i)} - \boldsymbol{w}_{t+1,*}^{(i)}\right\|^2 \Big| \mathcal{F}_{t+1,k}\right]$$

$$\leq \left[1 + \epsilon + 2\left(1+\frac{1}{\epsilon}\right)L_w^2\gamma^2\lambda^2\right](1-\eta_t\mu_\ell)^k g_t + 2\left(1+\frac{1}{\epsilon}\right)(1-\mu_\ell\eta_t)^k L_w^2\gamma^2\|\nabla F(\boldsymbol{w}_t^{(g)})\|^2 + \frac{2\eta_t}{\mu_\ell}\frac{\sigma^2}{Bm}. \quad (142)$$

$$\square$$

**Theorem 3.** *Suppose Assumptions 1, 2 and 4 hold. Let $\{\boldsymbol{w}_t^{(g)}\}_{t\geq 0}$ and $\{\boldsymbol{w}_{t,K}^{(i)}\}_{t\geq 0}$ be the sequence generated by Algorithm 2 with $\gamma < 1/L_g$, $\eta \leq 1/L_\ell$, and the inner loop iteration number satisfying*

$$\left(2 + 64L_w^2(1/\mu_g)^2\lambda^2\right)(1-\eta\mu_\ell)^K \leq (1-\gamma L_g)^4, \quad (143)$$

*then $\boldsymbol{w}_t^{(g)}$ converges to $\widetilde{\boldsymbol{w}}^{(g)}$ linearly at rate $1 - (\gamma\mu_g)/2 - (\gamma\mu_g)^2/2$ within a neighbor of*

$$\frac{8}{min\{\frac{1}{2}\gamma\mu_g, \mu_\ell\eta\}}\frac{1}{\mu_g^2}\lambda^2\eta^2\frac{\sigma^2}{Bm}.$$

*Proof.* Analogy to the proof of Theorem 2, we define

$$q := \left(1 + \epsilon + 2\left(1+\frac{1}{\epsilon}\right)L_w^2\gamma^2\lambda^2\right)(1-\eta_t\mu_\ell)^k. \quad (144)$$

If we assume

$$q \leq 1 - \gamma\mu_g \leq 1 - \gamma\mu_g + \gamma\epsilon_g, \quad (145)$$

and define

$$c_1 = \frac{1}{1 - \gamma\mu_g + \gamma\epsilon_g - q}\left(\frac{\gamma}{\epsilon_g} + \gamma^2(2 + \zeta^{-1})\right)\lambda^2, \quad (146)$$

combining Lemma 7 and Lemma 6, it yields

$$\mathbb{E}\left[\left\|\boldsymbol{w}_{t+1}^{(g)} - \tilde{\boldsymbol{w}}^{(g)}\right\|^2 \Big| \mathcal{F}_{t+1,k}\right] + c_1 g_{t+1}$$

$$\leq (1 - \gamma\mu_g + \gamma\epsilon_g)\left(\left\|\boldsymbol{w}_t^{(g)} - \tilde{\boldsymbol{w}}^{(g)}\right\|^2 + c_1 g_t\right) - \gamma^2(1 - \gamma L_g(1+\zeta))\left\|\nabla F(\boldsymbol{w}_t^{(g)})\right\|^2$$

$$+ \left(\frac{2}{1 - \gamma\mu_g + \gamma\epsilon_g - q}\left(\frac{\gamma}{\epsilon_g} + \gamma^2(2 + \zeta^{-1})\right)\lambda^2\right) 2\left(1 + \frac{1}{\epsilon}\right)(1 - \mu_\ell\eta_t)^k L_w^2\gamma^2\left\|\nabla F(\boldsymbol{w}_t^{(g)})\right\|^2$$

$$+ c_1\frac{2\eta_t}{\mu_\ell}\frac{\sigma^2}{Bm}.$$

For $\left\|\nabla F(\boldsymbol{w}_t^{(g)})\right\|^2$, it suffices to show

$$\frac{4\gamma}{1 - \gamma\mu_g + \gamma\epsilon_g - q}\left(\frac{1}{\epsilon_g} + \gamma(2 + \zeta^{-1})\right)\lambda^2(1 + \frac{1}{\epsilon})(1 - \mu_\ell\eta_t)^k L_w^2 \leq 1 - \gamma L_g(1+\zeta). \tag{147}$$

If we set $\epsilon_g = \frac{\mu_g}{2}$, it yields the convergence rate

$$1 - \frac{\gamma\mu_g}{2}. \tag{148}$$

And the condition (147) can be reformulated as

$$\frac{4}{\epsilon_g}\left(\frac{1}{\epsilon_g} + \gamma(2 + \zeta^{-1})\right)\lambda^2(1 + \epsilon^{-1})(1 - \mu_\ell\eta_t)^k L_w^2 \leq 1 - \gamma L_g(1+\zeta). \tag{149}$$

Restrict $\gamma$

$$\gamma: \quad \gamma(2 + \zeta^{-1}) \leq \frac{1}{\epsilon_g}. \tag{150}$$

Then condition (149) yields

$$\frac{8}{\epsilon_g^2}\lambda^2(1 + \frac{1}{\epsilon})(1 - \mu_\ell\eta_t)^k L_w^2 \leq 1 - \gamma L_g(1+\zeta). \tag{151}$$

Then if we assume $\epsilon = 1$ and $\zeta = 1 - \gamma L_g$, the conditions (145), (150) and (151) can be satisfied with the following sufficient conditions

$$\begin{aligned}
&\gamma < \frac{1}{L_g}, \\
&\eta_t \leq \frac{1}{L_\ell}, \\
&(2 + 4L_w^2\gamma^2\lambda^2)(1 - \eta_t\mu_\ell)^k \leq 1 - \gamma\mu_g, \\
&\frac{64}{\mu_g^2}(1 - \eta_t\mu_\ell)^k L_w^2\lambda^2 \leq (1 - \gamma L_g)^2.
\end{aligned} \tag{152}$$

And we can get a more straightforward sufficient condition

$$\gamma < \frac{1}{L_g}, \tag{153}$$

$$\eta_t \le \frac{1}{L_\ell}, \tag{154}$$

$$\left(2 + 64\frac{L_w^2}{\mu_g^2}\lambda^2\right)(1 - \eta_t\mu_\ell)^k \le (1 - \gamma L_g)^2. \tag{155}$$

Under such a sufficient condition, the contraction of the combination of inner loop and the outer loop would be rewritten as

$$\mathbb{E}\left[\left\|\boldsymbol{w}_{t+1}^{(g)} - \tilde{\boldsymbol{w}}^{(g)}\right\|^2 \Big| \mathcal{F}_{t+1,k}\right] + c_1 g_{t+1}$$

$$\le (1 - \gamma\mu_g + \gamma\epsilon_g)\left(\left\|\boldsymbol{w}_t^{(g)} - \tilde{\boldsymbol{w}}^{(g)}\right\|^2 + c_1 g_t\right) + \sum_{l=0}^{t}\left(1 - \frac{1}{2}\gamma\mu_g\right)^{t-l}\frac{2}{\epsilon_g^2}\lambda^2\sum_{l=0}^{k-1}(1 - \mu_\ell\eta)^{k-l}\eta^2\frac{\sigma^2}{Bm}$$

$$\le (1 - \frac{1}{2}\gamma\mu_g)^{t+1}\left(\left\|\boldsymbol{w}_0^{(g)} - \tilde{\boldsymbol{w}}^{(g)}\right\|^2 + c_1 g_0\right) + \sum_{l=0}^{t}\left(1 - \frac{1}{2}\gamma\mu_g\right)^{t-l}\frac{2}{\epsilon_g^2}\lambda^2\sum_{l=0}^{k-1}(1 - \mu_\ell\eta)^{k-l}\eta^2\frac{\sigma^2}{Bm} \tag{156}$$

$$\le (1 - \frac{1}{2}\gamma\mu_g)^{t+1}\left(\left\|\boldsymbol{w}_0^{(g)} - \tilde{\boldsymbol{w}}^{(g)}\right\|^2 + c_1 g_0\right) + \sum_{l=0}^{tk-1}2\left(\max(1 - \frac{1}{2}\gamma\mu_g, 1 - \mu_\ell\eta)\right)^l\frac{1}{\epsilon_g^2}\lambda^2\eta^2\frac{\sigma^2}{Bm}$$

$$\le (1 - \frac{1}{2}\gamma\mu_g)^{t+1}\left(\left\|\boldsymbol{w}_0^{(g)} - \tilde{\boldsymbol{w}}^{(g)}\right\|^2 + c_1 g_0\right) + \frac{8}{\min\{\frac{1}{2}\gamma\mu_g, \mu_\ell\eta\}}\frac{1}{\mu_g^2}\lambda^2\eta^2\frac{\sigma^2}{Bm}.$$

□

With $\eta = \min\{\frac{\mu(\lambda+L)}{4L(\lambda+\mu)^2}, \frac{1}{\lambda+L}\}$, $\gamma = 1/(2L_g) = \frac{\lambda+L}{2\lambda L}$, we can get the solution within an error ball $\left(\mathcal{O}(\varepsilon) + \min\{\frac{1}{4\mu L}, \frac{\lambda+\mu}{\mu^2(\lambda+L)}\}\frac{\sigma^2}{Bm}\right)$ with the communication cost $\mathcal{O}(\kappa_g \log(1/\varepsilon)) = \mathcal{O}\left(\kappa \cdot \frac{\lambda+\mu}{\lambda+L} \cdot \log\frac{1}{\varepsilon}\right)$ and the computation cost $\tilde{\mathcal{O}}\left(\kappa \cdot \log\frac{1}{\varepsilon}\right)$.

As stated in Theorem 4.2 of (Hanzely & Richtárik, 2020), when $p^\star = \frac{\lambda}{\lambda+L}$, the stochastic noise term is independent of the number of clients. Consequently, it does not theoretically demonstrate the expected benefit of collaboration in reducing noise. In contrast, our analysis reveals that even when adjusting $\lambda$, the noise remains of the same order, specifically $\mathcal{O}\left(\frac{1}{Bm}\right)$.

## C. Proof in Section 5

**Corollary 4.** *Under the assumptions of Theorem 1 and Corollary 2, $\boldsymbol{w}_{T,K}^{(i)}$ generated by Algorithm 1 satisfies*

$$\mathbb{E}\left\|\boldsymbol{w}_{T,K}^{(i)} - \boldsymbol{w}_\star^{(i)}\right\|^2$$

$$\le \left(1 - \frac{1}{\kappa} - \frac{L-\mu}{\kappa(\lambda+\mu)}\right)^T\mathbb{E}\left\|\boldsymbol{w}_{0,0}^{(i)} - \tilde{\boldsymbol{w}}^{(i)}\right\|^2 \tag{157}$$

$$+ \mathcal{O}\left[\left(\frac{1}{N} + R^2 + \frac{1}{q_1(\lambda)}\right) \wedge \left(\frac{1}{n} + q_2(\lambda)\right)\right]$$

*for any $i \in [m]$, where $q_1(\lambda)$ and $q_2(\lambda)$ are monotone increasing functions of $\lambda$ with $\lim_{\lambda\to+\infty} q_1(\lambda) = +\infty$ and $\lim_{\lambda\to0} q_2(\lambda) = 0$.*

*Proof.* The corollary is a direct result of Theorem 1 and Theorem 2, where $q_1(\lambda)$ and $q_2(\lambda)$ are defined in (70) and (67).

□

# D. Additional Experiment Details and Results

## D.1. More Experiment Details

Federated Learning (FL) has emerged as a widely adopted approach with numerous real-world applications, like Large Language Model (LLM) (Bai et al., 2024; Wu et al., 2024; Zhao et al., 2023), Reinforcement Learning (Zheng et al., 2024; 2025; Zhang et al., 2025), Diffusion Model (Li et al., 2024a). In this section, we provide more details on the implementation of the methods used in the empirical analysis.

**Real Dataset.** The MNIST dataset consists of 70,000 grayscale images of handwritten digits, each 28x28 pixels, classified into 10 classes (digits 0-9). The EMNIST dataset extends MNIST, including handwritten letters and digits. The Balanced split contains 131,600 28x28 grayscale images across 47 balanced classes (digits and uppercase/lowercase letters). The CIFAR-10 dataset contains 60,000 color images (32x32 pixels, 3 RGB channels), categorized into 10 classes representing real-world objects (e.g., airplane, car, and dog).

**Algorithms**. To fully investigate the effect of personalization in Problem (2) with the Algorithm B.1, we compare the two extreme algorithms (GlobalTraining and LocalTraining). Furthermore, since there exist other methods solving Problem (2) to do personalized federated learning, we also try to investigate the effect of personalization in other methods, like `pFedMe` (T Dinh et al., 2020). The algorithm `pFedMe` has a similar design with a double loop to solve the subproblem in each communication round.

**Selection of $\lambda$.** For the logistic regression on the synthetic dataset, we selected three values for $\lambda$: small $\lambda = 0.02$, medium $\lambda = 0.1$, and large $\lambda = 0.5$. For logistic regression on the MNIST dataset, we used small $\lambda = 0.5$, medium $\lambda = 1.5$, and large $\lambda = 2.5$. For CNN on the MNIST dataset, we used small $\lambda = 0.2$, medium $\lambda = 0.5$, and large $\lambda = 2.5$. Such a choice is to make sure that each line in the figure is clearly separable. Additionally, we implemented an adaptive choice of $\lambda$ following the formula outlined in Corrollary 1, with $\rho$ set to 2. The value of $\rho$ was determined through grid search over the range $[1, 10]$ to ensure optimal performance. For other real datasets, the regularization term $\lambda$ is set as small $\lambda = 0.1$, medium $\lambda = 0.5$, and large $\lambda = 1$.

**Selection of Step Size.** For the synthetic dataset, aligned with Theorem 2, we set the global step size $\gamma = (\lambda + L)/(\lambda L)$ and the local step size $\eta = 1/(L + \lambda)$. The smoothness constant $L$ is computed as the upper bound of the $L$-smoothness constant for logistic regression, specifically $4^{-1}\Lambda_{\max}$, where $\Lambda_{\max}$ denotes the largest eigenvalue of the Gram matrix $\boldsymbol{X}^\top \boldsymbol{X}$. The local step size for the synthetic dataset was further tuned through grid search in $\{0.001, 0.01, 0.1\}$. For the real dataset, the global learning rate was set to $1/\lambda$, while the local learning rate was set to 0.01, chosen via grid search. For both GlobalTrain and LocalTrain training baselines, the learning rate was also chosen via grid search within the range $\{0.001, 0.01, 0.1\}$. LocalTrain is implemented using a local (S)GD, and GlobalTrain is implemented using FedAvg (McMahan et al., 2017).

**Selection of Hyperparameters and Convolutional Neural Network Model.** For the MNIST dataset, we implemented logistic regression using the SGD solver with 5 epochs, a batch size of 32, and 20 total runs, and the same hyperparameter setup was used, except the batch size was reduced to 16 to ensure stable convergence in CNN with two convolution layers. To make our experiment more comprehensive, we also utilized a five-layer CNN to train the MNIST dataset with 10 epochs, a batch size of 16, and 50 total runs. For the EMNIST dataset, we used the same five-layer CNN with the same settings. And for CIFAR10, we implemented a three-layer CNN with 10 epochs, a batch size of 16, and 50 total runs. To study the data heterogeneity, we set the number of classes for each client in the MNIST, EMNIST, and CIFAR10 datasets as $\{2, 6, 10\}$, $\{30, 40, 47\}$, and $\{2, 6, 10\}$, respectively.

**Optimizer.** For the synthetic dataset, following the analysis in the paper, the loss function is optimized using gradient descent (GD). For the MNIST dataset, all models are implemented in PyTorch, and optimization is performed using the stochastic gradient descent (SGD) solver.

**Evaluation Criterion.** In terms of evaluation, for the synthetic dataset, we track the ground truth local model $\boldsymbol{w}_\star^{(i)}$. We compute the local statistical error as $\|\widetilde{\boldsymbol{w}}^{(i)} - \boldsymbol{w}_\star^{(i)}\|^2$. To capture the total error at the $t$-th iteration, we report the local total error $\|\boldsymbol{w}_{t,0}^{(i)} - \boldsymbol{w}_\star^{(i)}\|^2$, which combine both optimization and statistical errors. For the MNIST dataset, we report test accuracy on the MNIST testing set as a measure of the statistical error, and training loss for each client's training set as total error at the $t$-th iteration.

## D.2. Additional Experiment Results

**The Effect of Personalization on Accuracy over different data heterogeneity** To further validate our conclusions about the effect of personalization on solving Problem 2, we conduct additional experiments on real datasets, including EMNIST and CIFAR-10. For each dataset, we explore three levels of statistical heterogeneity, controlled by varying the number of classes each client has access to, which follows a widely used setup in federated learning literature. The models are trained using different convolutional neural network (CNN) architectures, denoted as CNN2 and CNN5, where the numbers indicate the number of convolutional layers in the network. As the ground truth underlying model is not accessible for these real datasets, we evaluate model performance based on classification accuracy on the test set, as shown in Tables 4. We also conduct experiments another regularization-based personalization method focusing on Problem (2), `pFedMe` (T Dinh et al., 2020), to demonstrate that our results could be generalized to other regularization-based personalized federated learning methods as well. For the columns corresponding to `FedCLUP` and `pFedMe`, we conduct experiments with varying regularization parameter $\lambda$, investigating the role of personalization in the algorithms. Specifically, we experiment with small, medium, and large $\lambda$ values (see Appendix D.1 for detailed parameter settings). In Table 4, we can see that consistently across datasets, as

| Dataset | Client Class | FedCLUP | | | GlobalTrain | LocalTrain | pFedMe | | |
|---|---|---|---|---|---|---|---|---|---|
| | | high | median | low | | | high | median | low |
| MNIST Logit | 2 | 0.5001 | 0.6011 | 0.6109 | 0.7001 | 0.3214 | 0.5534 | 0.6312 | 0.6712 |
| | 6 | 0.6813 | 0.7001 | 0.7241 | 0.7718 | 0.6000 | 0.6312 | 0.7123 | 0.7681 |
| | 10 | 0.8123 | 0.8213 | 0.8312 | 0.8391 | 0.7828 | 0.8001 | 0.8239 | 0.8291 |
| MNIST CNN2 | 2 | 0.3567 | 0.4721 | 0.6019 | 0.6123 | 0.3019 | 0.3973 | 0.5828 | 0.6172 |
| | 6 | 0.7001 | 0.7833 | 0.7843 | 0.8092 | 0.6231 | 0.7918 | 0.8093 | 0.8312 |
| | 10 | 0.8753 | 0.8893 | 0.9032 | 0.9098 | 0.8194 | 0.8756 | 0.8771 | 0.8749 |
| MNIST CNN5 | 2 | 0.8891 | 0.9000 | 0.9333 | 0.9377 | 0.6536 | 0.9000 | 0.9003 | 0.9380 |
| | 6 | 0.8451 | 0.8941 | 0.9392 | 0.9446 | 0.7333 | 0.9288 | 0.9186 | 0.9306 |
| | 10 | 0.9091 | 0.9102 | 0.9421 | 0.9433 | 0.7633 | 0.9340 | 0.9385 | 0.9440 |
| EMNIST CNN5 | 30 | 0.6084 | 0.6089 | 0.6011 | 0.6102 | 0.5583 | 0.6133 | 0.6129 | 0.6122 |
| | 40 | 0.6097 | 0.6123 | 0.6357 | 0.6423 | 0.5644 | 0.6012 | 0.6102 | 0.6134 |
| | 47 | 0.6278 | 0.6415 | 0.6672 | 0.6611 | 0.6111 | 0.6345 | 0.6532 | 0.6717 |
| CIFAR10 CNN3 | 2 | 0.6101 | 0.6292 | 0.6340 | 0.6444 | 0.5801 | 0.6056 | 0.6012 | 0.6033 |
| | 6 | 0.6211 | 0.6311 | 0.6712 | 0.6949 | 0.5623 | 0.6163 | 0.6163 | 0.6439 |
| | 10 | 0.6102 | 0.6712 | 0.7302 | 0.8041 | 0.6238 | 0.6744 | 0.7443 | 0.7732 |

*Table 4.* Evaluation accuracy of algorithms (`FedCLUP`, GlobalTrain, LocalTrain, and `pFedMe`) across datasets and client classes. (CNN$k$ means the CNN has $k$ convolutional layers, low, medium, and high indicate different degree of personalization, and Client Class indicates the number of classes on each client).

personalization degree increase, the accuracy approaches that of LocalTrain, which trains separate models for each client. Conversely, as personalization degree decreases, the accuracy approaches that of GlobalTrain, which represents purely global training. This interpolation behavior between LocalTrain and GlobalTrain, enabled by Problem (2), aligns well with our theoretical results in Theorem 1.

Moreover, as dataset homogeneity increases (i.e., clients share more similar data distributions), we observe a general improvement in accuracy across all methods, regardless of the level of personalization. This result demonstrates that while personalization offers significant benefits in highly heterogeneous settings, its necessity diminishes as heterogeneity decreases. Additionally, experiments with different CNN architectures confirm the robustness of these findings across model designs. Although we did not provide a theoretical analysis for `pFedMe` in this paper, our results suggest that the empirical behavior of `pFedMe` is consistent with the theoretical insights derived for `FedCLUP`. In particular, we observe that `pFedMe` also exhibits interpolation between local and global learning under varying levels of data heterogeneity. We observe in Table 4 that *LocalTrain consistently underperforms across all levels of statistical heterogeneity*. This is because, while statistical heterogeneity is introduced by restricting each client's exposure to only a subset of classes in the training process, evaluation is conducted on the entire dataset, including classes not seen during training. As a result, LocalTrain's performance remains lower than that of other methods.

**The Effect of Personalization on Statistical Error over different data heterogeneity** Figure 4 demonstrates the effect of personalization on statistical error across varying levels of statistical heterogeneity $R$. As demonstrated in the theory, increasing personalization (moving from `FedCLUP` (low per) to `FedCLUP` (high per)) shifts the solution closer to LocalTrain,

while decreasing personalization makes the model behave more like GlobalTrain. For low heterogeneity, methods with lower personalization (i.e., greater collaboration) achieve lower statistical error, as information can be effectively shared across clients to improve learning. GlobalTrain achieves the lowest error in this regime, while `FedCLUP` with less personalization also performs well. However, as statistical heterogeneity increases, collaborative learning becomes less effective. In this setting, LocalTrain outperforms other methods. Highly personalized `FedCLUP` also performs better than less personalized variants, aligning with the theory we established in Theorem 1. In Figure 5, we track testing loss as a

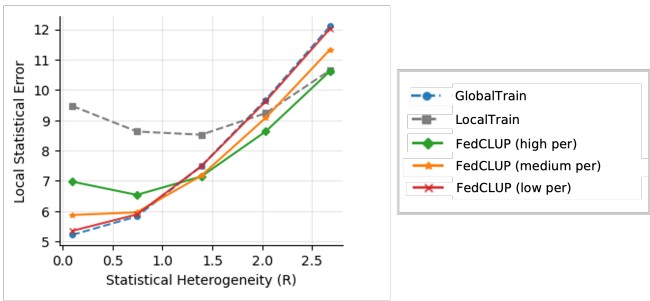

*Figure 4.* Impact of personalization on statistical error of `FedCLUP` solution across different levels of heterogeneity (high per means training `FedCLUP` with high personalization degree).

function of communication rounds. Across all sub-figures, `FedCLUP` with smaller $\lambda$ (higher personalization) demonstrates faster convergence, especially in the early stages of training. Thus, it shows that the trade-off between statistical accuracy and computation cost under the influence of personalization widely exists over statistical heterogeneity. **Different Number**

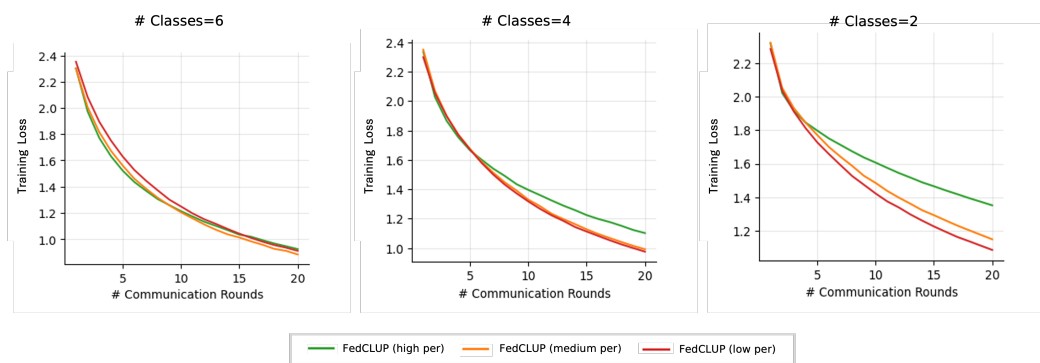

*Figure 5.* Logistic Regression training loss versus communication rounds for the `FedCLUP` methods on MNIST dataset (high per refers to training `FedCLUP` with high personalization degree).

**of Clients** In Figure 6, the top row shows results for 100 clients, and the bottom row shows results for 1,000 clients, and in both results we adopt a client sampling of 10% in each communication round. This tests the scalability and generalizability of the our results under typical federated learning setups. Consistent with the findings from Figure 1, `FedCLUP` with smaller $\lambda$ (higher personalization) consistently achieves faster convergence at the early stages of training, while `FedCLUP` with larger $\lambda$ (lower personalization) takes more communication rounds to converge. As statistical heterogeneity increases (measured by $R$), `FedCLUP` with smaller $\lambda$ gradually outperforms models with larger $\lambda$, showcasing the adaptability of solving Problem 2 to different levels of heterogeneity. These results further validate the theoretical insights from Theorem 1 and Theorem 2, highlighting the influence of personalization on statistical and optimization performance in federated setups with larger numbers of clients.

**The Effect of Personalization on Communication Cost** Figure 2 investigates how personalization impacts communication efficiency in `FedCLUP` by analyzing the benefit of increasing local updates under different personalization levels. In the low-personalization setting (left column), more local updates significantly accelerate convergence, reducing the reliance on frequent communication. However, in the high-personalization setting (right column), increasing local updates has a limited effect on convergence, indicating that frequent communication is essential for effective learning. This demonstrates

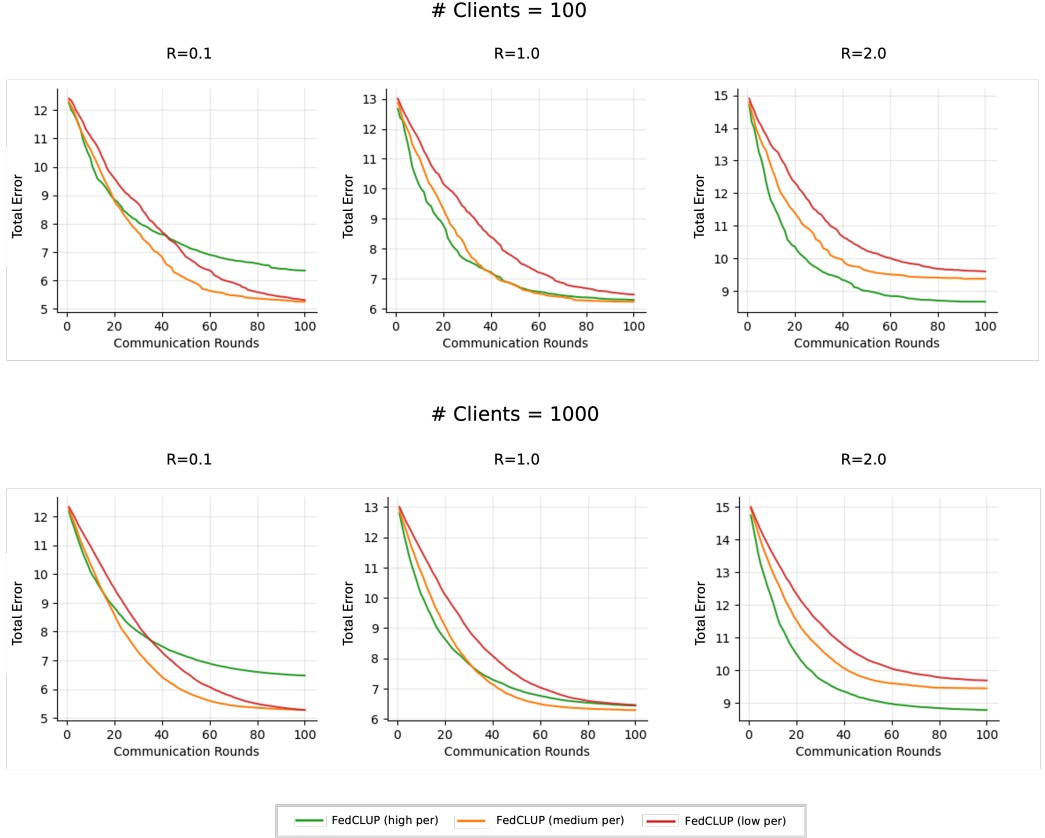

*Figure 6.* Performance of `FedCLUP` on the Synthetic dataset with 100 clients (top row) and 1,000 clients (bottom row) with 10% client sampling under different levels of statistical heterogeneity ($R$).

that higher personalization requires more communication rounds to achieve comparable performance, aligning with our theoretical findings.

### D.3. Additional Experiment for Rebuttal

#### D.3.1. STRONGLY CONVEX SETTING WITH SYNTHETIC DATA

As the theoretical results is established under strongly-convex settings, hence some expiremnts under strongly convex setting is expected with discussion related to the conditional number $\kappa$. We conduct an experiment on strongly convex problem: an overdeterminded linear regression task. We strictly follow the choice of local step size, local computation rounds, and global step size as specified in Corollary 2. As shown in Table 7 and 8, for a fixed personalization parameter $\lambda$, we observe that longer value of $\kappa$ result in slower convergence rates with respect tot he number of communication rounds. This expirical trend is consistent with our theoretical prediction in Corollary 2, where the number of communication rounds required to achieve a given target error $\epsilon$ scales with $\mathcal{O}(\kappa \frac{\lambda+L}{\lambda+\mu} \log(1/\varepsilon))$. Moreover, we observe that the impact of increasing $\kappa$ becomes stronger as $\lambda$ increases. This phenomenon aligns with our theoretical analysis, which shows that the sensitivity of communication complexity to $\kappa$ is amplified for larger values of $\lambda$. Similarly, we observe that increasing the condition number $\kappa$ also leads to a higher total number of gradient evaluations required to reach a target error. This observation is in agreement with our theoretical results where the total number of local gradient evaluations scales as $\mathcal{O}(\kappa \log(1/\varepsilon))$.

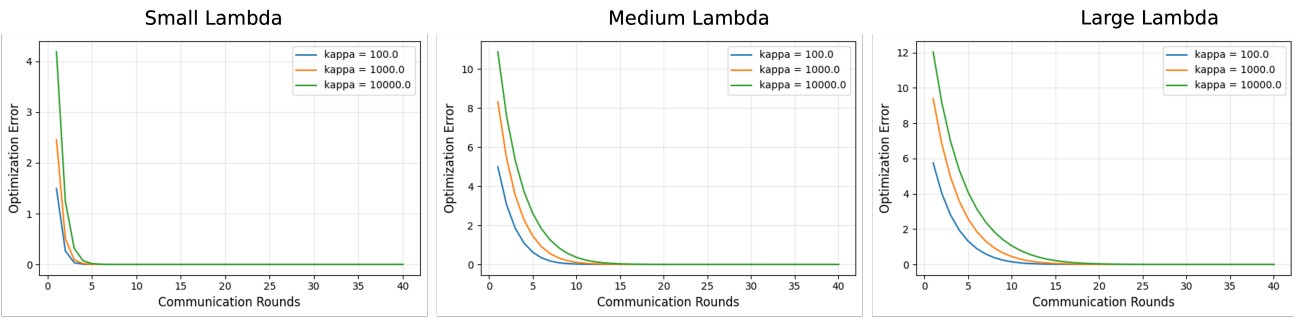

*Figure 7.* Optimization error versus communication rounds under varying condition numbers $\kappa$. The left, middle, and right panels correspond to FedCLUP with small, medium, and large values of $\lambda$, respectively. Across all settings, we observe that larger $\kappa$ leads to slower convergence, with the effect becoming more pronounced as $\lambda$ increases.

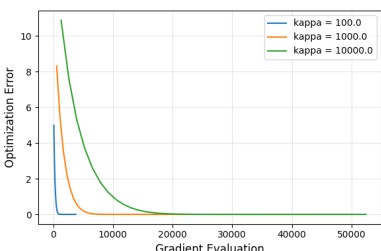

*Figure 8.* Optimization error versus total number of local gradient evaluations per client under varying condition numbers $\kappa$. As $\kappa$ increases, the number of gradient evaluations required to achieve a given optimization error also increases.

### D.3.2. DATASETS WITH NATURAL PARTITIONS

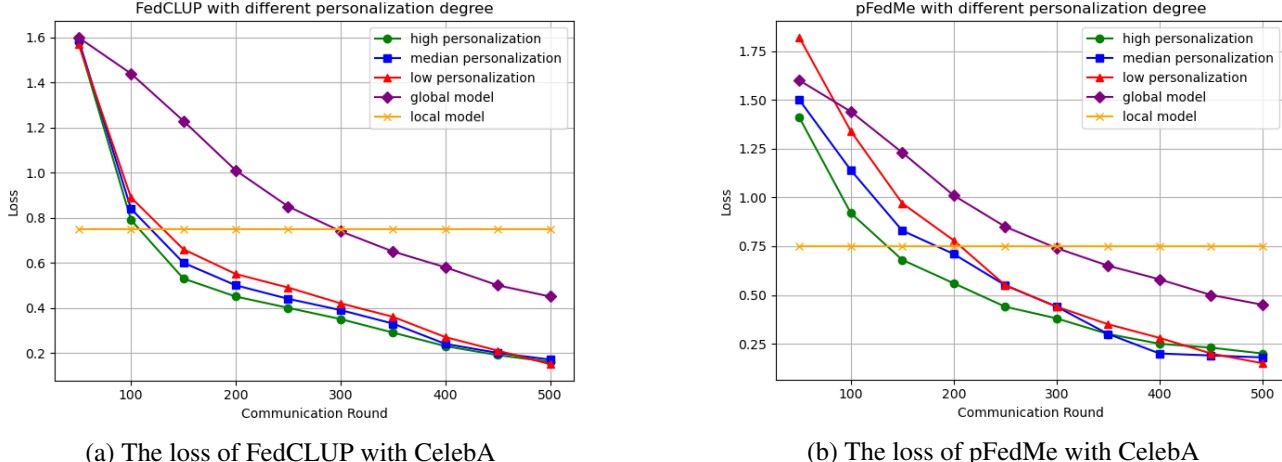

(a) The loss of FedCLUP with CelebA

(b) The loss of pFedMe with CelebA

*Figure 9.* Loss of **FedCLUP** (a) and **pFedMe** (b) with different personalization degrees on the CelebA dataset. Low, median, and high personalization correspond to regularization terms 0.5, 0.1, and 0.001, respectively. For FedCLUP, the test accuracy is highest with low personalization (0.919), followed by median (0.915) and high (0.888). For pFedMe, the test accuracy is highest for low personalization (0.910), followed by median (0.901) and high (0.890). The test accuracy for GlobalTrain and LocalTrain is 0.911 and 0.561, respectively.

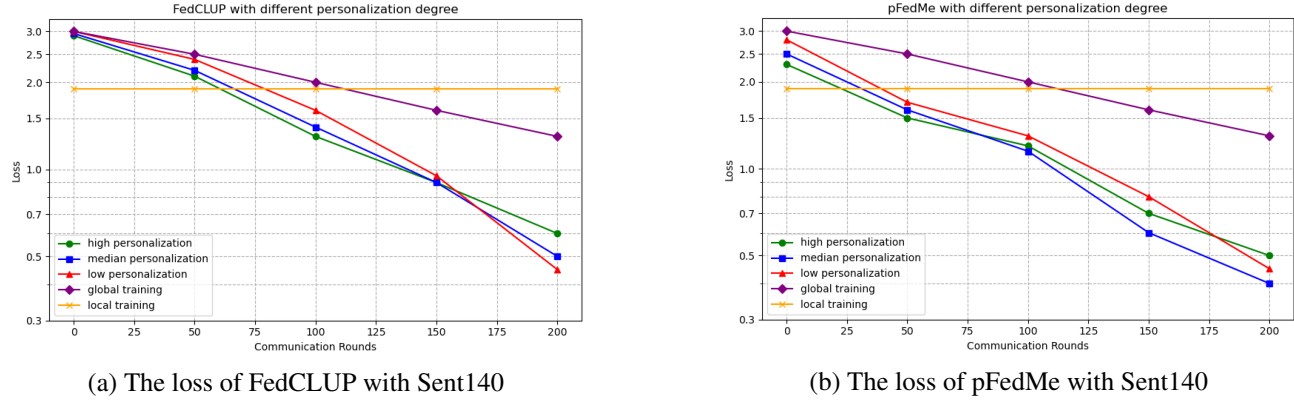

(a) The loss of FedCLUP with Sent140

(b) The loss of pFedMe with Sent140

*Figure 10.* Loss of **FedCLUP** (a) and **pFedMe** (b) with different personalization degrees on the Sent140 dataset. Low, median, and high personalization correspond to regularization terms 0.1, 0.005, and 0.001, respectively. For FedCLUP, the test accuracy is highest for low personalization (0.734), followed by median (0.702) and high (0.696). For pFedMe, the test accuracy is highest for low personalization (0.727), followed by median (0.703) and high (0.701). The test accuracy for GlobalTrain and LocalTrain is 0.730 and 0.353, respectively.

