# OpenReview forum: "Understanding the Statistical Accuracy-Communication Trade-off in Personalized Federated Learning with Minimax Guarantees"
_ICML.cc/2025/Conference — ICML 2025 poster_

### Official Review · Reviewer_g4zn · 2025-03-09

**Overall Recommendation:** 1

**Summary:**

In this paper, the authors study a personalized FL objective and showed its statistical accuracy under strongly convex, smooth model. The authors then propose a new algorithm to solve the problem. Empirical results show that as the personalization level changes, the model is able to interpolate between pure local training and pure global training.

**Claims And Evidence:**

- Problem 2 is different from the objective studied in Hanzley & Richtarik et al. 2020. The latter proposed a mean-regularized objective, where Problem 2 is a global-regularized objective. The proposed objective is more similar to what is proposed in Li et al. 2021 [1].
- In Section 5, it is incorrect to claim "Our work is the first to quantitatively characterize how changing the personalization degree
leads to the trade-off between communication cost and statistical accuracy." See section below for details.


[1] Li, T., Hu, S., Beirami, A., & Smith, V. (2021, July). Ditto: Fair and robust federated learning through personalization. In International conference on machine learning (pp. 6357-6368). PMLR.

**Essential References Not Discussed:**

As mentioned earlier, the key contribution of this work has a non trivial overlap with Li et al 2021, in terms of the method and findings. Please see previous sections for a detailed discussions.

**Experimental Designs Or Analyses:**

- Compared to the results in Li et al. 2021, I'm wondering whether the authors also observe the existence of an optimal lambda where the personalized model outperforms both pure global and pure local training. Currently, it looks like the evaluation does not outperform max{local, global}.
- Limited evaluation, the authors should perform experiments in settings with 1. varying level of heterogeneity, 2. datasets with natural partitions, such as CelebA, Sent140, etc.

**Methods And Evaluation Criteria:**

- The proposed FedCLUP method seems to be exactly the same as the method is Li et al. 2021 [1]. I just read through that paper more carefully. It seems [1] has already performed convergence analysis and showed how personalization degree leads to tradeoff between communication and utility. Hence, the novelty and contribution of the proposed work is over claimed. In that case, I wonder whether the contribution is mainly an extended theoretical analysis of this prior work?

[1] Li, T., Hu, S., Beirami, A., & Smith, V. (2021, July). Ditto: Fair and robust federated learning through personalization. In International conference on machine learning (pp. 6357-6368). PMLR.

**Other Comments Or Suggestions:**

NA

**Other Strengths And Weaknesses:**

NA

**Questions For Authors:**

NA

**Relation To Broader Scientific Literature:**

N/A

**Theoretical Claims:**

- I did not check all the steps of the proof. But the convergence rate seems correct.

---

> ### Author Rebuttal · Authors · 2025-04-01
>
> We thank the reviewer for the insightful and constructive comments. Below, we provide our detailed responses to each point. We hope these clarifications help address your concerns.
>
> **Comparison with Ditto [1]**
>
> Our work is significantly different from Ditto (Li et al. 2021 [1]) in multiple aspects and is far from being a simple extension of prior work.
>
> ***Different objective*** The formulation considered in Ditto [1] is given by
>
> $$\min _ {\\{w^{(i)}\\} _ {i = 1}^m } f _ i(w^{(i)}) + \frac{\lambda}{2} \\|w^{(i)} - w^{(g)}\\|^2, \quad \text{s.t.} \quad w^{(g)} \in \operatorname{argmin} _ {w} G(f_1(w), \ldots, f _ m(w)).$$
>
> Ditto first solves the lower-level problem to obtain $w^{(g)}$, then uses it in the upper-level problem to solve for each $w^{(i)}$. This differs from our formulation, where the local models $w^{(i)}$ and the global model $w^{(g)}$ are optimized jointly. Accordingly, we adopt a completely different set of techniques for analyzing the problem.
>
>
> ***Different Trade-off*** More importantly, although Ditto also investigates a trade-off phenomenon, they study a different trade-off compared with ours. They primarily investigate the effect of personalization to balance the trade-off between robustness to adversarial clients and the generalization benefits gained from collaboration. As such, the trade-off they study focuses on the strategy to minimize the statistical error of the final solution, and does not explicitly incorporate optimization error. In contrast, our work investigates the trade-off between statistical accuracy and communication efficiency, aiming to optimize the total error, which includes both statistical error and optimization error.
>
> While Ditto briefly mentions a communication-utility trade-off in Section 3.2, it is only proposed as a potential direction for future analysis. While they provide some experimental results in their paper that illustrate this phenomenon, the trade-off between communication and statistical accuracy hasn't been *quantitatively analyzed* in the paper. In contrast, our work is specifically centered on the theoretical understanding of this trade-off. We provide a comprehensive theoretical analysis that explicitly characterizes how both statistical accuracy and communication complexity depend on the personalization parameter $\lambda$. As a unique contribution, our theory gives a principled understanding of how to select $\lambda$ to optimize the overall performance in federated settings (see our response to the next question for more details).
>
>
> ***Different Theoretical Results*** Additionally, we note that the theoretical analysis in Ditto is restricted to a simplified linear regression setting, as acknowledged in their paper under a "simplified set of attacks and problem settings". In contrast, we provide a tight theoretical analysis for a general function class $f$. Our statistical error bound is proved to be minimax optimal and our optimization convergence is shown to be linear. Establishing both convergence and generalization bounds in this more general setting is technically non-trivial and constitutes a key contribution of our work.
>
>
>
>
> **Observing the existence scenarios that FedCLUP outperforms GlobalTrain and LocalTrain**
>
>
> Such scenarios do exist. As shown in the left panel of [Figure](https://postimg.cc/Dm6w5vdw), FedCLUP achieves a lower total error than both LocalTrain and GlobalTrain between 5 and 40 communication rounds. Moreover, to reach a given target total error, there exists a unique optimal $\lambda$ that minimizes the required number of communication rounds. Based on this, we propose a dynamic tuning strategy to approximate the optimal $\lambda$, as demonstrated in the right panel of the same figure. Please refer to our response to Reviewer Krd5 for further discussion.
>
> **Perform experiments with varying levels of heterogeneity**
>
> We indeed consider different levels of heterogeneity in Appendix D.2. Table 4 sets the data heterogeneity by varying the number of classes for the clients' data.
>
> **Perform experiment with natural partitions**
>
> Thanks for the reviewer's suggestion. We add more experiments to study the CelebA and Sent140 datasets. The experiment details follow [1] and [2] respectively. As an example, in [Figure](https://postimg.cc/H8r5drTd), we observe similar trends as in Figure 1 and Table 1. We will cover more results, elaborate on these findings, and include them in the final version of the paper.
>
>
>
>
> [1] Li, T., Hu, S., Beirami, A., \& Smith, V. (2021, July). Ditto: Fair and robust federated learning through personalization. In International conference on machine learning (pp. 6357-6368). PMLR.
>
> [2] Duan M, Liu D, Ji X, et al. Fedgroup: Efficient federated learning via decomposed similarity-based clustering. IEEE SustainCom, 2021: 228-237.

---

> > ### Comment · Reviewer_g4zn · 2025-04-05
> >
> > Thanks for your detailed response.
> > - *Different objective*: Thanks for pointing this out. I agree the objective is slightly different. However, looking into Li et al. 2021, I thought their algorithm is essentially also doing joint optimization? I also checked algorithm 1 in Appendix B and found that the solver is indeed different. I think a more proper way to compare the two works is to say that the proposed work is using a different solver than prior work.
> > - *Different objective*: The update in algorithm 1 is a bit hard to understand. What is the intuition of setting the model update for the global model as the gradient of the regularization term? Plugging the second last line into the last line, the update is basically scaling the global model with $(1-\gamma\lambda)$ and then subtract the linear combination local models? What's the motivation that this update finds the best global model for the objective?
> > - *Different Trade-off*: Thanks for the clarification.
> > - *Additional experiments*: I appreciate the authors for performing new experiments. For those plots, I would love to see where the global, local, other personalized FL methods stand.

---

> > > ### Author Response · Authors · 2025-04-07
> > >
> > > We thank the reviewer for acknowledging our previous response and for the follow-up questions. Below we provide clarifications accordingly, and hope they address your concerns.
> > >
> > > **Difference in Algorithm Between Ditto and Our Work**
> > >
> > > We would like to reiterate that the key difference lies in the difference in the trade-off being studied. Next, we discuss the difference in terms of algorithm:
> > >
> > >
> > > $\bullet$ ***Ditto***: Although Ditto updates both local and global models within a single communication round, this is essentially *a merging of two sequential steps*. Conceptually, their objective does not involve an explicit coupling between local and global models, therefore, their objective can be solved *sequentially*: one can first solve the global model (e.g., via FedAvg), and then solve each local model individually. The objective investigated by Ditto itself does not require a joint optimization approach.
> > >
> > > $\bullet$ ***Our Work***: In contrast, our objective, which is fundamentally different from theirs, induces a *natural coupling* between the local and global models, which necessitates a joint optimization approach. This coupled structure is central to both our algorithmic design and theoretical analysis. As discussed in our main text and the previous response, this leads to a different algorithmic structure and analysis techniques.
> > >
> > >
> > > **Understanding of the Model Update**
> > >
> > > We thank the reviewer for raising this point. Due to space constraints, we refer to Section 4.2 of the main paper for full details and summarize the key idea here.
> > >
> > > Problem (2) can be equivalently reformulated into the following bilevel form (Equation 10)
> > > \begin{align*}
> > > & \min \_{w^{(g)}} F(w^{(g)}):=\frac{1}{m} \sum\_{i=1}^m F\_i(w^{(g)}), \\
> > > & \text { where } \quad F\_i(w^{(g)}):=\min \_{w^{(i)}} h\_i(w^{(i)}, w^{(g)}) .
> > > \end{align*}
> > > Thus global and local models can be solved iteratively. For the local model, given the current global model, updating the local model can apply gradient descent directly on $h_i(\cdot)$ (Equation 11). Similarly, for global model, as Lemma 2 states, we can also apply gradient descent on $F_i$, with the gradient given by $\nabla F_i(w^{(g)}) = \lambda(w^{(g)}-w_{\star}^{(i)}(w^{(g)}))$. Therefore, the global gradient indeed involves the regularization term, which naturally comes from the objective.
> > >
> > > Now further notice that $\nabla F_i(w^{(g)})$ involves $w_{\star}^{(i)}(w^{(g)})$, the minimizer of the inner loop problem (defined in (11)), which is not directly available to use. Therefore, a natural approximation for $w_{\star}^{(i)}(w^{(g)})$ is the local model output in FedCLUP, yielding the approximating gradient $\hat{\nabla} F_i(w^{(g)})$. This is precisely the expression used in the second-to-last line of Algorithm 1. This explains why the local models are involved when updating the global model. Since our theoretical analysis considers a convex setting, applying gradient descent ensures convergence to the minimizer of the global model. We hope this clarifies the intuition and addresses the reviewer’s question.
> > >
> > >
> > >
> > > **Additional Experiments**
> > >
> > > Upon the reviewer's request, we further conducted experiments of GlobalTrain, LocalTrain, and an additional PFL method, pFedMe on these two new datasets. As [Figure](https://postimg.cc/23ffpjMh) 6(a) and 7(a) show, the results are consistent with our findings in Figure 1 of the manuscript and the analysis presented in our previous response [Figure](https://postimg.cc/Dm6w5vdw). Similarly, in [Figure](https://postimg.cc/23ffpjMh) 6(b) and 7(b) we observe similar trends for another PFL method, pFedMe, further supporting the observed trade-offs. We will expand on these findings and incorporate the results into the final version of the paper.
> > >
> > > We hope our responses have clarified the reviewer’s questions. We sincerely appreciate the thoughtful feedback and hope it supports a positive re-evaluation of our submission.

---

### Official Review · Reviewer_Krd5 · 2025-03-13

**Overall Recommendation:** 3

**Summary:**

This paper studies the trade-off of accuracy and communication in personalized federated learning and presents the theoretical analysis of the effects of personalization degree. The theoretical findings are validated on synthetic and real-world datasets.

**Claims And Evidence:**

The claims are supported by theoretical analysis and experimental validations.

**Essential References Not Discussed:**

NA

**Experimental Designs Or Analyses:**

The experimental design and analysis are sound in general.

**Methods And Evaluation Criteria:**

The proposed method and evaluation make sense.

**Other Comments Or Suggestions:**

NA

**Other Strengths And Weaknesses:**

**Strength**
- The analysis of the trade-off between personalization and communication is useful for the federated learning community.

- The findings are well supported by experimental results.



**Weakness**
- The practical implication on the choice of \lambda is not clear from the corollary 1. It would be helpful to further discuss the potential solution of \lambda.

- The definitions of high, medium, and low personalization degrees are not clear.

**Questions For Authors:**

Please see the weakness part.

**Relation To Broader Scientific Literature:**

This paper contributes to the general federated learning community.

**Theoretical Claims:**

The theoretical claims and proofs seem to be correct.

---

> ### Author Rebuttal · Authors · 2025-04-01
>
> We thank the reviewer for the insightful and constructive comments. Below, we provide our detailed responses to each point. We hope these clarifications help address your concerns.
>
> **The practical implication on the choice of $\lambda$**
>
> For the practical implication of personalization in our experiments, we include an illustrative example on synthetic data (similar to that in Figure~1a of the manuscript) that demonstrates how the personalization degree, controlled by $\lambda$, influences convergence behavior and total error. Specifically, we compare several settings: (1) local training, corresponding to the limiting case of FedCLUP as $\lambda \to 0$, (2) global training, corresponding to FedCLUP as $\lambda \to \infty$, and (3) FedCLUP with three intermediate values of $\lambda$.
>
> As Figure 1 (See detail in [Figure](https://postimg.cc/Dm6w5vdw)) shows, different values of $\lambda$ yield distinct convergence speeds and final total errors. In particular, a smaller $\lambda$ (i.e., stronger personalization)  results in faster convergence in terms of communication rounds due to the algorithm relying less on communication among clients. However, this comes at the cost of a larger final total error, as reduced collaboration across clients limits the statistical generalization. Conversely, larger $\lambda$ (i.e., weaker personalization or more collaboration) leads to slower convergence but consistently achieves a lower final total error due to the benefit of collective learning.
>
> As a direct implication, shown in the left part of [Figure](https://postimg.cc/Dm6w5vdw) when aiming to reach a target total error, there exists a unique optimal $\lambda$ that minimizes the total number of communication rounds needed to achieve that error compared with FedCLUP with other personalization degree or GlobalTrain. This insight provides an important practical guideline derived from the theory established in Corollary 3.
>
> **The potential solution of $\lambda$ in practice**
>
> For the potential solution in practice, we propose a heuristic dynamic tuning strategy for practical guidance to approximate the optimal $\lambda$. As shown in the right part of [Figure](https://postimg.cc/Dm6w5vdw), the idea is to begin with a small $\lambda$ (i.e., close to LocalTrain) to leverage its communication efficiency. As soon as the validation performance plateaus or the statistical error stops improving significantly, we gradually increase $\lambda$. This allows the model to benefit from enhanced generalization through increased collaboration,  at the expense of slightly higher communication cost. By progressively adjusting $\lambda$ in this way, one can finally identify the optimal personalization degree that balances the trade-off and minimizes the total communication cost required to meet a desired performance threshold.
>
> For the broader impact, the dynamic strategy would possibly be applied to other regularization-based personalization federated learning algorithms as tuning guidance. It would save the communication cost a lot for achieving a target performance.
>
>
>
> **the definition of personalization**
>
>
> In D.1. Additional Experiment Details, for each experiment, the small $\lambda$ value represents a high personalization, the median $\lambda$ represent the median personalization degree, and the large $\lambda$ represents high personalization. The specific choice of $\lambda$ is specified in Appendix D.1.

---

> > ### Comment · Reviewer_Krd5 · 2025-04-07
> >
> > Thank you for your response and clarifications! I don’t have any further questions.

---

> > > ### Author Response · Authors · 2025-04-07
> > >
> > > Dear Reviewer Krd5,
> > >
> > > Thank you for your positive feedback and valuable suggestions. During the period of rebuttal, we have conducted more experiments as shown in [Figure](https://postimg.cc/23ffpjMh), where we used natural partition datasets with more complex models to verify our findings in [Figure](https://postimg.cc/Dm6w5vdw), giving practical guidance in selecting the personalization degree.
> > >
> > > Once again, thank you for your review and positive evaluation, and we hope our response provides additional clarity and encourages a further positive assessment of our work.
> > >
> > > Best regards,
> > >
> > > The Authors

---

### Official Review · Reviewer_5cgC · 2025-03-14

**Overall Recommendation:** 2

**Summary:**

The authors theoretically address the accuracy-communication trade-off in personalized federated learning (FL). In other words, $\lambda$, which controls the regularization between global and local models, represents the accuracy-communication trade-off, and an analysis of this is conducted.

**Claims And Evidence:**

- It is difficult to follow. While the theoretical explanation is well-structured, it would be helpful to summarize the key takeaways more clearly. The main claim is not entirely clear. Please clarify the distinct insights that this theoretical analysis offers beyond the broadly understood existing knowledge.

**Essential References Not Discussed:**

None

**Experimental Designs Or Analyses:**

- The models and datasets used in the experiment are all too outdated.
- The performance is lower compared to other algorithms used for comparison, raising concerns about its practicality.

**Methods And Evaluation Criteria:**

- I am unsure how Algorithm 1 differs from the existing algorithms. Additionally, is it correct that Algorithm 2 differs from Algorithm 1 only in terms of batch consideration?

**Other Comments Or Suggestions:**

- While the theoretical aspects are important, I prioritize the practical aspects more. I wanted to clarify that this evaluation was made with that perspective in mind.

**Other Strengths And Weaknesses:**

None

**Questions For Authors:**

None

**Relation To Broader Scientific Literature:**

This study aims to theoretically analyze the trade-off between communication and accuracy, which is the fundamental aspect of FL.

**Theoretical Claims:**

Yes.

---

> ### Author Rebuttal · Authors · 2025-04-01
>
> We thank the reviewer for the insightful and constructive comments. Below, we provide our detailed responses to each point. We hope these clarifications help address your concerns.
>
> **The summarized key takeaway as our main claim**
>
> From a statistical perspective (Section 4.1), we provide a tight generalization bound for Problem 2 under standard assumptions. Our analysis quantifies how the degree of personalization $\lambda$ affects the statistical accuracy. We show that when collaborative learning is beneficial, increasing personalization reduces collaboration across clients,  which may degrade the statistical accuracy of the problem solution.
>
> From an algorithmic perspective (Section 4.2), we analyze the convergence of FedCLUP. Our results characterize how $\lambda$ influences the number of communication rounds required for convergence. Increasing personalization implies less dependence on server-client communication, which reduces communication overhead and thus improves communication efficiency.
>
> By combining these two parts, summarized in Section 5—demonstrate a fundamental trade-off: higher personalization may improve communication efficiency but at the cost of statistical accuracy. This claim leads to a key practical insight: to achieve a specific target error, there exists an optimal choice of personalization degree that minimizes the total communication cost required to reach the desired performance. The claim is verified in Figure 1. (See [Figure](https://postimg.cc/Dm6w5vdw) and the response to reviewer Krd5 for a detailed description).
>
>
> **Unsure how Algorithm 1 differs from the existing algorithms**
>
> The existing algorithms solving Problem 2 are listed in Table 1, where most of the existing works either don't explicitly analyze the local convergence cost, like FedProx and pFedMe, or fail to demonstrate such a trade-off. For the designed algorithm, we quantitatively characterize how changing the personalization degree leads to the trade-off between communication cost and statistical accuracy. In contrast, most prior studies either do not explicitly analyze statistical convergence or fail to provide a fine-grained analysis linking optimization and statistical error, therefore lacking a theoretical guarantee for practical guidance.
>
>
> **The Difference between Algorithm 1 and Algorithm 2**
>
> Algorithm 2 is the stochastic version of Algorithm 1, accounting for the scenarios where the sample size is large and computing the full gradient is impractical. We also extended our analysis to this setting, accounting for the injection of stochastic noise into the algorithm.
>
>
> **The models and datasets are all too outdated**
>
> Thank you for the suggestion. We have extended our experiments to include more recent datasets and models such as CelebA (image classification) and Sent140 (NLP). We also experiment with more diverse models from CNN to LSTM. Some preliminary results are available [here](https://postimg.cc/H8r5drTd), showing similar trends as in Figure 1 and Table 1. We will add more details on the experiments, elaborate on these findings, and include them in the final version of the paper.
>
> **The performance is lower compared to other algorithms used for comparison, raising concerns about its practicality**
>
> We would like to clarify that the primary goal of our experiments is not to demonstrate that the proposed algorithm statistically outperforms existing methods. Instead, our experiments are designed to validate the theoretical insights and provide practical tuning guidance. To this end, the experiments mainly focus on evaluating one algorithm across different levels of personalization. This design enables a clean and focused investigation of the trade-offs described in Corollary 3.
>
> Beyond the validation of trade-off, our analysis does provide practical guidance. First, based on the trade-off, there exists an optimal $\lambda$ for reaching the target error with the minimal communication cost (See Figure 1 or detailed description in Left of [Figure](https://postimg.cc/Dm6w5vdw)). For practicality, when the optimal $\lambda$ is unknown, we provide tuning guidance (See Right of [Figure](https://postimg.cc/Dm6w5vdw)): as soon as the validation performance plateaus or the statistical error stops improving significantly, we gradually increase $\lambda$. The dynamic strategy approximates the optimal $\lambda$, achieving target error with minimal communication cost (See the response under reviewer Krd5 for details).

---

### Official Review · Reviewer_XVuq · 2025-03-16

**Overall Recommendation:** 2

**Summary:**

This paper studies personalized federated learning, i.e., where data owners (clients) have their own distribution.
The paper studies in particular the trade off between exploiting more shared knowledge (increasing the communication cost) and relying mire on local data.

**Claims And Evidence:**

The theorems are supported by proofs in the appendix.
The proofs are rather hard to follow

The paper contains experiments, which are illustrative and compare different levels of personalization but dont really support the claims and dont compare to existing methods.  Table 2 does some comparison with pFedMe but does seem to show significant differences.

**Essential References Not Discussed:**

--

**Ethical Review Concerns:**

--

**Experimental Designs Or Analyses:**

The experiments dont indicate which conclusions are statistically significant.

The experiments are illustrative, not aimed at proving claims, hence it is not critical that their design is soubd.

**Methods And Evaluation Criteria:**

The authors claim to investigate the communication cost, but in reality only study the number of iterations.  The communication cost is assumed to be linear in the number of rounds, but this ignores approaches such as fedavg and approaches which send only compressed gradients in every round.

The theoretical results are compared to other work in Table 1 but as the several results use different variables the comparison is difficult and it is unclear what we can conclude.

It would be interesting to see experiments that show how and when the results of the current paper are significantly better than existing work.

**Other Comments Or Suggestions:**

* 115R : solution of i-th local model -> solution of the i-th local model
* Eq (5) the variable i in $\sum_{i=1}^m$ shadows the different variable i used among others in the preceding term $w_*^{(i)}$. Please use another index, e.g., j, instead
* 282R : a higher degree of collaboration will also results -> result

**Other Strengths And Weaknesses:**

My main concern is the significance of the work.

**Questions For Authors:**

--

**Relation To Broader Scientific Literature:**

The work ignores alternative optimization algori5hms and other strategies which may affect the relation between the communication cost per round. The paper analyzes a single (simple) algorithm.

The paper compares with earlier work analyzing convergence speedn but as said Table 1 is not easy to interpret.

**Theoretical Claims:**

I went through most proofs but they are hard to follow and confirming their correctness would require a huge amount of time

---

> ### Author Rebuttal · Authors · 2025-04-01
>
> We thank the reviewer for the insightful and constructive comments. Below, we provide our detailed responses to each point.
>
> **The Structure of the Theoretical Proof**
>
> We establish the statistical convergence rate (Theorem 1) in Section A.3. Specifically, Section A.3.1 derives the statistical rate for the global model $\tilde{w}^{(g)}$, and Section A.3.2 builds upon this result to establish the statistical rate for each local model $\tilde{w}^{(i)}$. We then derive the optimization convergence rate (Theorem 2) in Section B.2. The analysis proceeds in stages: we first establish the convergence rate for the global model in Section B.3, followed by the local convergence rate in Section B.5. We will further refine and reorganize the proof structure later.
>
> **Significance of Experiment, Validation of Theoretical Claims, and Comparison with Existing Algorithms**
>
> The primary goal of our experiments is *not* to demonstrate that the proposed algorithm statistically outperforms existing methods. Instead, our experiments are designed to validate the theoretical insights. To this end, the experiments focus on evaluating *one algorithm across different levels of personalization*, rather than making direct comparisons across algorithms. This design enables a clean and focused investigation of the trade-offs described in Theorems 1 and 2 and Corollary 3:
>
> $\bullet$ Statistical Accuracy: As shown in Table 2 and Table 4 (Appendix), when collaborative learning is beneficial, decreasing the personalization degree improves generalization performance. This observation directly supports the theoretical result in Theorem 1.
>
> $\bullet$ Communication Efficiency: Figure~1 and Figure 3,4,5 (Appendix) show that decreasing the personalization degree leads to slower convergence in terms of communication rounds. More communication is required to achieve the same level of optimization error, which is consistent with the theoretical analysis in Theorem 2 and Corollary 2.
>
> $\bullet$ Trade-off: the trade-off characterized in Corollary 3 is verified: increasing the personalization degree reduces communication cost but may hurt statistical accuracy.
>
> For practical implications, there exists an optimal choice of $\lambda$ that uses minimal communication rounds to achieve a given target total error (See Figure 1 or Left of [Figure 3](https://postimg.cc/Dm6w5vdw). This provides a tuning guidance for personalization to utilize the tradeoff. As Right of [Figure 3](https://postimg.cc/Dm6w5vdw) shows, as soon as the validation performance plateaus or the statistical error stops improving significantly, we gradually increase $\lambda$. The dynamic strategy approximates the optimal  $\lambda$ and meets a desired error threshold with minimal communication rounds. Due to space constraints, we kindly refer the reviewer to our response to Reviewer Krd5 for a more detailed discussion.
>
> **The communication cost and approaches with compressed gradients**
>
> We define the communication cost as the total number of communication rounds. The vanilla FedAvg method, as far as we know, also measures the communication cost using this metric[1]. Compression schemes such as top-K pruning and quantization, on the other hand, are extra techniques one can integrate and apply to either the model or the gradient transmitted. This line of work is orthogonal to our study.
>
> **Compare to other work in Table 1, but use different variables. So the comparison is difficult, and it's unclear for the conclusion**
>
> The key takeaway from Table 1 is that most prior works omit statistical analysis, whereas we fill this gap by deriving a minimax-optimal statistical rate. Our communication cost is also nearly optimal (up to logarithmic factors [2]) and has a clear, interpretable dependence on the personalization parameter. This makes our work the first to quantitatively characterize the communication–accuracy trade-off. Indeed, different works adopt different notations and assumptions, making direct alignment difficult—we will add clarifications later.
>
> **Focuses on a basic algorithm, omitting others that might influence communication costs per round**
>
> The primary goal of this work is to provably show the trade-off and provide tuning guidance. The algorithm we propose is intentionally designed as a clean, simple, and principled instantiation for solving Problem 2. We have derived a minimax optimal statistical and a sharp linear optimization convergence bounds to clearly disentangle the role of the personalization degree in the trade-off.
>
> **Typo error**
>
> Thank you for pointing this out. We will carefully proofread the manuscript and correct all typos in the revised version.
>
> We hope our responses address your concerns.
>
> [1] Koloskova A, Loizou N, Boreiri S, et al. A Unified Theory of Decentralized SGD with Changing Topology and Local Updates[J].
>
> [2] Hanzely F, Hanzely S, Horvath S, et al. Lower Bounds and Optimal Algorithms for Personalized Federated Learning[J]. 2020.

---

### Official Review · Reviewer_XV8t · 2025-03-19

**Overall Recommendation:** 3

**Summary:**

This paper proposed a personalized federated learning algorithm that captures the relationship between communication cost and the degree of personalization. Convergence theories are derived, showing that the total required gradient steps are irrelevant to the personalization degree $\lambda$, while the scaling of the communication cost w.r.t  $\lambda$ is related to the conditional number $\kappa $. Numerical results further showed the convergence speed of their method under different personalization degree.

**Claims And Evidence:**

Yes, the claims and results are all backup with proof or evidence.

**Essential References Not Discussed:**

Based on my knowledge, the paper provides a reasonable overview of prior work relevant to its key contributions. I am not currently aware of any essential references that are missing.

**Experimental Designs Or Analyses:**

The theoretical result in this work is under strongly-convex settings, hence some experiments under strongly convex settings is expected with discussion related to the conditional number $\kappa$ . However, it seems the experiments are all under weakly-convex/non-convex, hence it would be nice if additional discussion on strongly-convex settings (synthetic data should be enough) can be added.

**Methods And Evaluation Criteria:**

Yes, the choice of the datasets makes sense, and the additional data heterogeneity experiments in the appendix showed the difference between various personalization settings.

**Other Comments Or Suggestions:**

I am not currently aware of any typos or minor mistakes.

**Other Strengths And Weaknesses:**

The rate they derived is based on strongly-convex. However, it unclear when under weakly-convex/non-convex cases, where possibily non-unique set of optimal/stationary solutions exists, whether if the accuracy-communication trade-off they discover can still hold true. Further analysis should provide a better insight for more real-world optimization problems.

**Questions For Authors:**

There are a few points I wish the authors to address:
1.	The theoretical results are derived under strongly convex settings, but the experiments appear to be conducted in weakly convex or non-convex settings. It would be beneficial to include experiments under strongly convex settings, possibly with synthetic data, and discuss their relation to the condition number $\kappa$.
2.	Shouldn't the definition of $w_*$ 's parameter space be centered around the global ground truth model instead? A clearer justification for this choice would strengthen the argument.
3.	The derived rate assumes strong convexity, but it is unclear whether the accuracy-communication trade-off still holds in weakly convex or non-convex settings, where multiple optimal or stationary solutions may exist. Further analysis is needed to understand how these findings extend to real-world optimization problems.

**Relation To Broader Scientific Literature:**

The work provides a new algorithm and theoretical insight for personalized FL to trade-off between communication cost and personalization level.

**Theoretical Claims:**

I didn't find the definition of 's parameter space quite convincing. The parameter space is a ball build around the weighted average of all local ground truth models. However, if goal this space is to capture the heterogeneity, shouldn't the ball measure the Euclidean distance between the global ground truth (the optimal model of the sum of all local models) instead of the weighted average of the local ground truth? A further explanation of why such measure is chosen to evaluate the statistical error would make the result more persuasive.

---

> ### Author Rebuttal · Authors · 2025-04-01
>
> We thank the reviewer for the insightful and constructive comments. Below, we provide our detailed responses to each point. We hope these clarifications help address your concerns.
>
> **The concern raises for the definition of parameter space**
>
> A natural measure of local model's heterogeneity could be the average of their mutual distance given by $\frac{1}{m^2}\sum_{i = 1}^m \sum_{j = 1}^m \\|w_\star^{(i)} - w_\star^{(j)}\\|^2$, which is equivalent to $\frac{2}{m} \cdot \sum_{i = 1}^m \\| w_\star^{(i)} - w_\star^{(g)}\\|^2$ with $w_\star^{(g)} := \frac{1}{m} \sum_{i=1}^m w_\star^{(i)}$. Therefore, we define the data heterogeneity over the parameter space as $\frac{1}{m} \sum_{i = 1}^m \\| w_\star^{(i)} - w_\star^{(g)}\\|^2\leq R^2$. The definition is also used in prior works [1][2].
>
>
>
> **Experiment with strongly convex and discuss the condition number -- least square problem, vary the choice of condition number, examine both communication and computation**
>
>
> We thank the reviewer for this valuable suggestion. In response, we conducted additional experiments on a strongly convex problem: an overdetermined linear regression task. We strictly follow the choices of local step size, local computation rounds, and global step size as specified in Corollary 2. Please refer to [Figure1](https://postimg.cc/BLJmCB4b) and [Figure2](https://postimg.cc/zbRG0K7m) for detailed experimental results.
>
> As suggested, we study the effect of the condition number $\kappa$ on convergence behavior. Specifically, for a fixed personalization parameter $\lambda$, we observe that larger values of $\kappa$ result in slower convergence rates with respect to the number of communication rounds. This empirical trend is consistent with our theoretical prediction in Corollary 2, where the number of communication rounds required to achieve a given target error $\epsilon$ scales with $\mathcal{O}(\kappa \frac{\lambda + L}{\lambda + \mu} \log(1/\varepsilon))$. Moreover, we observe that the impact of increasing $\kappa$ becomes stronger as $\lambda$ increases. This phenomenon aligns with our theoretical analysis, which shows that the sensitivity of communication complexity to $\kappa$ is amplified for larger values of $\lambda$. Similarly, we observe that increasing the condition number $\kappa$ also leads to a higher total number of gradient evaluations required to reach a target error. This observation is in agreement with our theoretical results where the total number of local gradient evaluations scales as $\mathcal{O}(\kappa\log(1/\varepsilon))$.
>
>
>
> **The study focuses on strongly-convex problems, but the accuracy-communication trade-off may differ in weakly-convex or non-convex cases with non-unique solutions. Further analysis is needed**
>
> Establishing the statistical rate of the solution relies on the  strong convexity of the loss. This reason is that for strongly convex problems the minimizer is unique, which enables analyzing its property using first order optimality conditions. However, weakly convex and nonconvex problems can have multiple minimizers and stationary points. Under such cases, first order stationarity can no longer distinguish these points, leave alone characterizing their statistical accuracy. One may resort methods such as consider losses with special landscape (e.g., restricted strong convexity, one-point strong convexity) or algorithmic regularization to tackle some subclasses of nonconvex losses, yet such extensions are
> highly nontrivial and would require a different set of techniques. We acknowledge that analyzing the convergence of the algorithm to a stationary point for weakly convex objectives could be more tractable, but doing so alone without having its statistical accuracy cannot reveal the communication-accuracy tradeoff.
>
>
> [1] Chen, S., Zheng, Q., Long, Q., and Su, W. J. (2023). Minimax Estimation for Personalized Federated Learning: An Alternative between FedAvg and Local Training? Journal of Machine Learning Research, 24(262), 1-59.
>
> [2] Duan Y, Wang K. Adaptive and robust multi-task learning. The Annals of Statistics, 2023, 51(5): 2015-2039.

---

### Decision · Program_Chairs · 2025-05-01

**Decision:**

Accept (poster)

**Comment:**

This paper makes a very important theoretical contribution to personalized federated learning (PFL), by analyzing the statistical accuracy and its relation to the personalization coefficient and communication efficiency. Because PFL is essentially about generalization performance, standard approaches that analyze the algorithmic convergence using optimization theory are insufficient to show the gains in PFL. This paper takes the first step to analyze the statistical accuracy, i.e., the gap between the empirical optimal model and the statistical optimal model (the latter assumes knowledge of the full data distribution), for an actual implementable algorithm. The results can enrich the fundamental understanding of PFL.

I feel that the paper would reach an audience with more relevant interest if the title could include keywords like statistical accuracy, theoretical analysis, etc. The current title is too generic. Based on the reviewers' comments, it would be helpful to include a more detailed comparison with related works (parts of such discussions could go into the appendix if the main paper's space is limited) and experiments with more sophisticated datasets and models. Although such additions may not be closely related to the core contribution of this work, they could help readers in the general area of FL and PFL understand this work's contributions better.